

# Non-invertible symmetries and LSM-type constraints on a tensor product Hilbert space

Nathan Seiberg[1], Sahand Seifnashri[1] and Shu-Heng Shao[2]

**1** School of Natural Sciences, Institute for Advanced Study, Princeton, NJ
**2** C. N. Yang Institute for Theoretical Physics, Stony Brook University, Stony Brook, NY

## Abstract

We discuss the exact non-invertible Kramers-Wannier symmetry of 1+1d lattice models on a tensor product Hilbert space of qubits. This symmetry is associated with a topological defect and a conserved operator, and the latter can be presented as a matrix product operator. Importantly, unlike its continuum counterpart, the symmetry algebra involves lattice translations. Consequently, it is not described by a fusion category. In the presence of this defect, the symmetry algebra involving parity/time-reversal is realized projectively, which is reminiscent of an anomaly. Different Hamiltonians with the same lattice non-invertible symmetry can flow in their continuum limits to infinitely many different fusion categories (with different Frobenius-Schur indicators), including, as a special case, the Ising CFT. The non-invertible symmetry leads to a constraint similar to that of Lieb-Schultz-Mattis, implying that the system cannot have a unique gapped ground state. It is either in a gapless phase or in a gapped phase with three (or a multiple of three) ground states, associated with the spontaneous breaking of the lattice non-invertible symmetry.

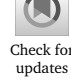

# 1  Introduction

Symmetry plays a central role in our understanding of nature. In particular, it serves as a powerful tool in analyzing strongly coupled quantum systems. The notion of global symmetry has recently been generalized in several directions, leading to exciting results and developments.

In continuum quantum field theory, generalized global symmetries are defined by topological operators/defects [1]. (See Section 1.5.) This definition has led to the notion of higher group and non-invertible symmetries, together with many generalizations. See [2–8] for recent reviews.

## 1.1  The lattice and the continuum symmetries

Given the rapid development of generalized symmetries in continuum field theories, it is natural to ask under what conditions they can also exist as exact symmetries on the lattice. In particular, can models based on a tensor product Hilbert space have such symmetries? What is the relation between these lattice symmetries and their continuum counterparts?

In this work, we focus on arguably the simplest possible non-invertible symmetry, often known as the Kramers-Wannier duality symmetry in 1+1d [9–12]. Specifically, we consider the lattice realization of this symmetry in quantum spin chains with $L$ sites. We focus on 1+1d lattice Hamiltonian systems, where the Hilbert space is a tensor product of two-dimensional Hilbert spaces for each site of the chain.

Let $X_j$ and $Z_j$ denote the Pauli operators acting on the $j$-th site of the chain. (See Appendix A, for our conventions.) The Hamiltonian is such that the theory is invariant under translation $T$ acting on local operators $O_j$ as

$$TO_j T^{-1} = O_{j+1}, \qquad j \sim j + L. \tag{1}$$

We also impose that the Hamiltonian is invariant under an ordinary $\mathbb{Z}_2$ global symmetry generated by

$$\eta = \prod_{j=1}^{L} X_j, \tag{2}$$

Table 1: The Kramers-Wannier symmetry operator algebra on the lattice and in the continuum. The first column is the lattice operator algebra on a periodic chain with $L$ sites. The second column is the symmetry algebra in the continuum. $\mathcal{N}$ is the non-invertible continuum symmetry and $P$ is the momentum of the continuum theory. The lattice operators commutes with the transverse-field Ising Hamiltonian at the critical coupling and the continuum algebra is realized by the Ising CFT. As we discuss in section 2.2.2, there is a similar algebra of defects both on the lattice and in the continuum.

| lattice operators | continuum operators |
|---|---|
| $\eta^2 = 1$, $\quad \eta D = D\eta = D$ | $\eta^2 = 1$, $\quad \eta\mathcal{N} = \mathcal{N}\eta = \mathcal{N}$ |
| $D^2 = (1+\eta)T^{-1}$ | $\mathcal{N}^2 = 1 + \eta$ |
| $TD = DT = D^\dagger$ | $\mathcal{N} = \mathcal{N}^\dagger$ |
| $T\eta = \eta T$, $\quad T^L = 1$ | $e^{2\pi i P} = 1$ |

with

$$\eta^2 = 1\,,$$
$$\eta X_j = X_j \eta\,, \qquad \eta Z_j = -Z_j \eta\,. \tag{3}$$

Finally, we come to the non-invertible symmetry. It is implemented by a non-invertible operator D that satisfies the algebra [13][1]

$$\text{lattice}: \quad D^2 = (1+\eta)T^{-1}\,,$$
$$\eta D = D\eta = D\,. \tag{4}$$

The algebra (4) (see also Table 1) implies that the operator D is not invertible (i.e., it has a nontrivial kernel). Also, it is clear that D projects onto the $\mathbb{Z}_2$ even states. Its action on $\mathbb{Z}_2$ invariant operators is the standard Kramers-Wannier transformation

$$DX_j = Z_{j-1}Z_j D\,,$$
$$DZ_{j-1}Z_j = X_{j-1}D\,. \tag{5}$$

(The action on $\mathbb{Z}_2$ odd operators, like $Z_j$ is more complicated.)

The transverse-field Ising Hamiltonian (10) is the prototypical example of a Hamiltonian invariant under this symmetry. But we will also consider more general systems invariant under this symmetry. We sometimes refer to this lattice realization of the Kramers-Wannier symmetry as a *non-invertible lattice translation* since the transformation in (5) squares to lattice translation by one site $T^{-1}$ on the $\mathbb{Z}_2$-even sector.

Comparing with the literature, there are at least two general approaches to construct the non-invertible lattice operator D and the corresponding defect on the lattice:

- Kramers-Wannier duality, which is implemented by gauging the $\mathbb{Z}_2$ global symmetry [14–18] (see Appendix B). The resulting non-invertible operator admits a presentation in terms of a matrix product operator (MPO) [19,20] (see Section 2.3.2). See, for example, [21,22] for reviews of MPOs.

- Bosonizing the Majorana chain [13] (see Appendix C). It is also related to a sequential quantum circuit [23–25] (see Appendix D).

---

[1]Comparing with [13], our D corresponds to $\sqrt{2}D^\dagger$, where $D$ is the non-invertible symmetry operator there (see the discussion around equation (C.17)).

Unlike some other references (such as [14–18, 26–28]), we will insist that the operator D acts as an operator on the Hilbert space of the theory, rather than being a map from one Hilbert space to another. This will allow us to examine its algebra, and in particular, to compute $D^2$, as in (4).

We emphasize that the non-invertible lattice translation symmetry D forms a different algebra than its continuum counterpart. The latter symmetry $\mathcal{N}$ satisfies [10–12]:

$$\text{continuum}: \quad \begin{aligned} \mathcal{N}^2 &= 1 + \eta\,, \\ \eta\mathcal{N} &= \mathcal{N}\eta = \mathcal{N}\,. \end{aligned} \tag{6}$$

See Table 1. Crucially, the algebra generated by D mixes with lattice translation and depends on the number of lattice sites $L$, becoming infinite-dimensional on an infinite chain.

## 1.2 The lattice symmetry is not a fusion category

So far, we discussed the symmetry operators. Importantly, the symmetries are also related to topological defects.

In relativistic continuum field theories, (non-invertible) global symmetries are defined by topological operators and defects.[2] They are topological in the sense that physical answers do not depend on small changes of their locations. Symmetry operators act on the Hilbert space at a given time. They are maps from the Hilbert space to itself. Symmetry defects are stretched along the time direction and correspond to changes in the system. In Euclidean signature, there is no distinction between operators and defects.

In Hamiltonian lattice models, we should distinguish between the symmetry operators and the symmetry defects. A symmetry operator is associated with a conserved operator that commutes with the Hamiltonian and acts within the same Hilbert space. However, not every conserved operator qualifies as a global symmetry. The crucial property is locality. More specifically, we focus on the symmetry operator that is associated with a defect, which is represented by a localized modification of the original Hamiltonian. Below, we will discuss the precise relation between them. In particular, in Section 2, we will use a symmetry defect to derive the corresponding symmetry operator.

Invertible internal symmetries are described by symmetry groups and their 't Hooft anomalies. This information is characterized by the symmetry operators, the defects, and their interactions, which capture the anomalies.

Finite internal non-invertible symmetries are not captured by groups. In 1+1d, their symmetry operators and defects are described by fusion categories [29, 30].[3] (In the special case of finite invertible symmetries in 1+1d, the description in terms of fusion categories is also valid and it describes the symmetry group and its anomalies.) A typical example is the internal non-invertible symmetry of (6), which is described by the Tambara-Yamagami (TY) fusion category [43].

However, all this does not apply to the translation symmetry. Although $T$ generates a symmetry of the problem, since it does not act internally, the corresponding defect is quite subtle. As we review in Appendix E, there are two kinds of translation defects $\mathcal{T}^+$ and $\mathcal{T}^-$. $\mathcal{T}^+$ adds a site to our chain and $\mathcal{T}^-$ removes a site from our chain [44–53]. Consequently, as emphasized in [53], the width of the defect $\mathcal{T}^{-n} = (\mathcal{T}^-)^{\otimes n}$, which removes $n$ sites, is proportional to $n$. Therefore, for large $n$ ($n \sim L$), such defects are nonlocal and hence they cannot be described by a fusion category. Related to that, while we can add an arbitrary

---

[2]In some circles, the term "topological defect" refers to a defect that is associated with the topology of field space. This is not the definition we will use here.

[3]This fact was first mentioned in the context of continuum field theory in [12, 31, 32]. See [10, 11, 33–42] for earlier related works in the context of rational CFTs.

number of $\mathcal{T}^+$ defects, we cannot add an arbitrary number of $\mathcal{T}^-$ defects. (See more about this point in [52, 53].) Finally, in the presence of a defect associated with the symmetry operator R, the group relation corresponding to periodic boundary conditions $T^L = 1$ is modified to $T^{-L} = R$. Such modifications in the relations are not incorporated in the fusion category.

Now, our lattice symmetry (4) involves lattice translation $T$ and therefore, it also cannot be described by a standard fusion category. Instead, one should use a more general mathematical setup. Although we do not yet have such a setup, we will present some preliminary step toward finding it. In particular, in Appendix E, we will present a construction of the defects $\mathcal{T}^\pm$. And in section 2 we will discuss the symmetries in the presence of various defects.

Even though the lattice symmetry is not described by a fusion category, it flows in the continuum to a fusion category. In section 3, we will examine what information about the continuum fusion category can be obtained already on the lattice.

## 1.3 On anomalies of non-invertible symmetries

Standard (i.e., invertible, zero-form) internal symmetries are characterized by a group and there is a clear understanding of their possible 't Hooft anomalies [54]. For our purposes, we need to extend this treatment:

- In the continuum, finite non-invertible symmetries in 1+1d are characterized by a fusion category. Just as for ordinary symmetries, there is a notion of gauging the entire fusion category [31,55–64]. The obstruction to that gauging can be interpreted as an anomaly.[4] In particular, the fusion category of the Ising CFT has such an anomaly. It implies that its long-distance behavior cannot be completely trivial even if we deform the system, while preserving this symmetry. One topic we will address is how to treat such non-invertible symmetries on the lattice.

- Spacetime symmetries appear on the lattice as crystalline symmetries. It is interesting when these crystalline symmetries mix with internal symmetries, and in particular, when they lead to new "emanant" internal symmetries in the continuum [52]. Then, anomalies in crystalline symmetries [45, 46, 52, 65–69] are matched in the low-energy theory by anomalies in that emanant internal symmetry.

- In our discussion below, we will face a combination of these issues. We will have a non-invertible symmetry on the lattice, which mixes with the crystalline symmetry and therefore it is not described by a fusion category. Related to that, its anomalies are particularly subtle. Nevertheless, in Section 2.5, we will find that the symmetry algebra in the presence of a non-invertible defect is realized projectively. This is reminiscent of the consequence of an anomaly for ordinary symmetries.

## 1.4 LSM-type constraints

An important consequence of the symmetries of a system is possible Lieb-Schultz-Mattis (LSM) constraints, which forbid a unique gapped ground state [44–46, 52, 70–97]. In that case, the system is either gapless or some of its global symmetries (either internal or crystalline) are spontaneously broken. One of our main results is a similar constraint following from the exact non-invertible lattice translation symmetry (4):

---

[4]In the literature, a fusion category is sometimes referred to as anomaly-free if it admits a fiber functor, i.e., a module category with one simple object. Physically, it means that the fusion category is compatible with a trivially gapped phase. See [60–62] for the relation between this notion of anomalies and the obstruction to gauging. The fusion category of the Ising CFT is anomalous in both senses.

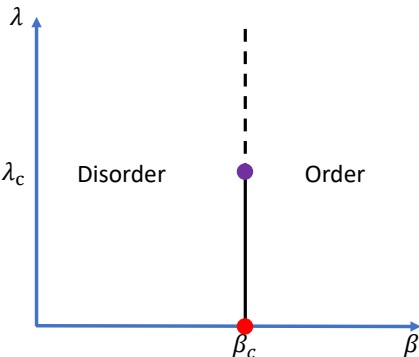

Figure 1: The phase diagram at the vicinity of the tricritical Ising CFT point (purple point). The continuous black line is a second order line ending at the critical Ising lattice model (red point). The black solid line flows to the Ising CFT. The black dashed line is a first order transition. Along that line, the theory is gapped and has three low-lying states.

*Any system with a finite-range Hamiltonian preserving the non-invertible lattice translation symmetry* D *must either be gapless or gapped with its symmetry being spontaneously broken. In the latter case, the number of superselection sectors must be a multiple of 3.*

Unlike other LSM-type constraints, the way we will argue for that conclusion in Section 4 (which follows the continuum discussion in [98–100] closely) will be quite elementary and will not use more abstract notions involving anomalies.

A characteristic example, which demonstrates this constraint is the phase diagram of the tricritical Ising model. (See [101–104] for recent related studies.) Its phase diagram is presented in Figure 1. At $\beta_c$ the model is invariant under Kramers-Wannier duality and therefore has the non-invertible symmetry D. For other values of $\beta$ the symmetry is not present. For $\beta > \beta_c$ the global $\mathbb{Z}_2$ symmetry is spontaneously broken and the model is ordered. In finite volume, the system has two nearly degenerate ground states with a gap above them. For $\beta < \beta_c$ the global $\mathbb{Z}_2$ symmetry is unbroken and the model is disordered. The system has a unique gapped ground state. The vertical line at $\beta = \beta_c$ corresponds to a phase transition between these two phases. The solid line is a second order transition where the theory flows from the tricritical Ising model to the critical Ising model. The dashed line is a first order line. In finite volume, the theory along the dashed line has three low-lying states with a gap above them. Two of them are the low-lying states for $\beta > \beta_c$ and the third is the ground state for $\beta < \beta_c$. In infinite volume, the theory has two superselection sectors for $\beta > \beta_c$, one superselection sector for $\beta < \beta_c$, and three superselection sectors for $\beta = \beta_c$. In the latter case, the non-invertible symmetry D and the invertible $\mathbb{Z}_2$ symmetry $\eta$ both act non-trivially on the three superselection sectors, and we interpret it as the spontaneous breaking of D and $\eta$.

## 1.5 Topological defects

Let us elaborate more on the defect in a Hamiltonian lattice model. Consider the system on a periodic chain of size $L$ with Hamiltonian $H$. The insertion of a defect $\mathcal{A}$ in the system is represented by modifying some terms in the Hamiltonian $H$ near link $(j - 1, j)$.[5] We denote

---

[5]There are two equivalent ways to represent a defect for an internal invertible symmetry on the lattice. First, we use the same Hamiltonian, but impose twisted boundary conditions on the operators. Second, we keep the periodic boundary conditions on the operators, but modify the Hamiltonian locally in some neighborhood. Throughout this paper, we use the latter perspective to represent a (possibly non-invertible) defect. See [52, 53] for more discussions.

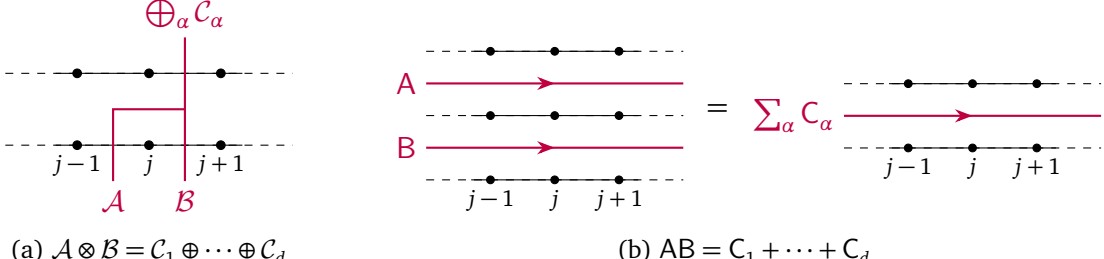

(a) $\mathcal{A} \otimes \mathcal{B} = \mathcal{C}_1 \oplus \cdots \oplus \mathcal{C}_d$        (b) $AB = C_1 + \cdots + C_d$

Figure 2: Fusion of topological defects vs. the algebra of conserved operators. In these spacetime diagrams, time runs upward. Figure 2a on the left denotes a local unitary operator $\lambda_{\mathcal{A} \otimes \mathcal{B}}$ implementing the fusion of $\mathcal{A}$ with $\mathcal{B}$ – it conjugates the defect Hamiltonian $H_{\mathcal{A};\mathcal{B}}$ to $H_{\mathcal{C}_1 \oplus \cdots \oplus \mathcal{C}_d} = H_{\mathcal{C}_1} \otimes |1\rangle\langle 1| + \cdots + H_{\mathcal{C}_d} \otimes |d\rangle\langle d|$. In the special case when $\mathcal{B}$ is the trivial defect, this fusion operation reduces to the movement operator for $\mathcal{A}$ in (8). Figure 2b on the right indicates the fusion of the symmetry operators A and B.

the modified Hamiltonian by $H_{\mathcal{A}}^{(j-1,j)}$ which represents a $\mathcal{A}$ defect on link $(j-1,j)$, and refer to it as the defect Hamiltonian. We will also often use the phrase "twisted theory" for the system with a defect.[6]

The topological property of the defect means that the location of the defect is arbitrary and can be changed by conjugating the defect Hamiltonian with local unitary operators. Specifically, the defect Hamiltonians $H_{\mathcal{A}}^{(j-1,j)}$ and $H_{\mathcal{A}}^{(j,j+1)}$ are related by conjugation with a local unitary operator:

$$H_{\mathcal{A}}^{(j,j+1)} = U_{\mathcal{A}}^j H_{\mathcal{A}}^{(j-1,j)} (U_{\mathcal{A}}^j)^{-1} . \tag{7}$$

We refer to such local unitary operators $U_{\mathcal{A}}^j$ as the *movement operator*. Pictorially, it is represented as

$$U_{\mathcal{A}}^j = \qquad\qquad\qquad\qquad . \tag{8}$$

Using the local unitary operators, we define the fusion of two topological defects $\mathcal{A}$ and $\mathcal{B}$. The fusion rule takes the form $\mathcal{A} \otimes \mathcal{B} = \bigoplus_\alpha \mathcal{C}_\alpha$, where '$\otimes$' represents fusion and '$\oplus$' represents direct sum operation. They are defined as follows:

- **Fusion:** First, $\mathcal{A} \otimes \mathcal{B}$ represents the topological defect obtained by putting an $\mathcal{A}$ defect and a $\mathcal{B}$ defect next to each other; say, respectively, on links $(j-1,j)$ and $(j,j+1)$ as in Figure 2a. We denote the corresponding defect Hamiltonian by $H_{\mathcal{A};\mathcal{B}}$.

- **Direct sum:** The defect $\bigoplus_\alpha \mathcal{C}_\alpha = \mathcal{C}_1 \oplus \cdots \oplus \mathcal{C}_d$ corresponds to taking the direct sum of systems with $\mathcal{C}_1, \mathcal{C}_2, \cdots, \mathcal{C}_d$ defects. A defect Hamiltonian for $\mathcal{C}_1 \oplus \cdots \oplus \mathcal{C}_d$ is given by $H_{\mathcal{C}_1 \oplus \cdots \oplus \mathcal{C}_d} = H_{\mathcal{C}_1} \otimes |1\rangle\langle 1| + \cdots + H_{\mathcal{C}_d} \otimes |d\rangle\langle d|$, where we have added an extra qudit, associated with $|1\rangle, |2\rangle, \cdots, |d\rangle$, to the Hilbert space.

---

[6]More precisely, a theory describes a system as a function of its background fields. In particular, different defect configurations should be viewed as part of the same theory even though their Hamiltonians have different spectra. Nevertheless, we will occasionally follow standard imprecise language and refer to the system with defects as twisted theory.

- **Fusion operator:** The fusion relation 2a is equivalent to a local unitary operator $\lambda_{\mathcal{A}\otimes\mathcal{B}}$, which we refer to as the *fusion operator*, satisfying[7]

$$\lambda_{\mathcal{A}\otimes\mathcal{B}} H_{\mathcal{A};\mathcal{B}} (\lambda_{\mathcal{A}\otimes\mathcal{B}})^{-1} = H_{\mathcal{C}_1\oplus\cdots\oplus\mathcal{C}_d}. \tag{9}$$

In this paper we will focus on an example of such a fusion rule involving the non-invertible Kramers-Wannier symmetry. From the topological defects $\mathcal{A}, \mathcal{B}, \mathcal{C}_\alpha$, we will construct the corresponding conserved operators[8] $\mathsf{A}, \mathsf{B}, \mathsf{C}_\alpha$ satisfying the operator algebra $\mathsf{A}\mathsf{B} = \sum_\alpha \mathsf{C}_\alpha$.[9] See Figure 2.

## 1.6 Outline

The paper is organized as follows.

Section 2 is devoted to the symmetries and defects of the lattice system with emphasis on the Kramers-Wannier non-invertible symmetry. In particular, we derive the non-invertible operator D from the corresponding defect $\mathcal{D}$. Section 2.4 presents the algebra of symmetry operators in the presence of various defects and Section 2.5 discusses a projective phase in the symmetry algebra involving parity/time-reversal in the presence of the non-invertible defect.

Section 3 explores the relation between the lattice non-invertible symmetry and its continuum counterpart. Section 3.1 reviews the fusion category symmetry of the continuum Ising CFT, which emanates from the lattice non-invertible symmetry of the transverse-field Ising model, and compares it with the lattice symmetry. The rest of Section 3 addresses more details of this comparison.

Section 4 discusses more general lattice models with the same symmetry including the non-invertible symmetry D. These deformed models are presented in Section 4.1. Then, Section 4.2 argues for an LSM-type constraint that holds in all such D-preserving systems.

A series of appendices presents reviews of useful background material, more technical details, and various extensions of our discussion.

Appendix A outlines our notations and conventions. Appendix B reviews the construction of the lattice non-invertible symmetry operator D and the corresponding defect $\mathcal{D}$ via gauging the $\mathbb{Z}_2$ symmetry generated by $\eta$. Appendices C and D review the construction of the non-invertible symmetry from the Majorana chain and the sequential quantum circuit perspectives, respectively. Appendix E provides an in-depth discussion of translation defects. Appendix F contains detailed calculations of the fusion algebra involving the lattice non-invertible symmetry operators. Appendix G reviews aspects of the Tambara-Yamagami fusion category describing the continuum non-invertible symmetry of the Ising CFT. Appendix H demonstrates that the lattice non-invertible symmetry can lead to two different fusion category symmetries with different Frobenius-Schur indicators in the continuum. Appendix I discusses the non-invertible symmetry in the continuum and its spontaneous breaking in infinite volume. Examples in 1+1d supersymmetric theories are discussed. Finally, Appendix J reviews the constraints due to non-invertible symmetries on renormalization group flows of continuum theories.

---

[7]Here, we have not specified the location of the defects for simplicity. Later, we will incorporate the location of the defects into our notations, and (9) will become $\lambda^j_{\mathcal{A}\otimes\mathcal{B}} H^{(j-1,j);(j,j+1)}_{\mathcal{A};\mathcal{B}} (\lambda^j_{\mathcal{A}\otimes\mathcal{B}})^{-1} = H^{(j,j+1)}_{\mathcal{C}_1\oplus\cdots\oplus\mathcal{C}_d}$.

[8]Unless otherwise stated, on the lattice, we will use different fonts for the operator and the corresponding defect. For example, the defect $\mathcal{A}$ is associated with the symmetry operator $\mathsf{A}$. Also, in the presence of the defect $\mathcal{A}$, the symmetry algebra could differ from the algebra without defects. First, some symmetry operators are no longer conserved. Second, other symmetry operators $\mathsf{B}$ are not conserved but can be deformed to be conserved. We denote the deformed operator by $\mathsf{B}_{\mathcal{A}}$, which generally obeys a different fusion relation.

[9]While the coefficients in the fusion of defects must be non-negative integers, the coefficients in the operator algebra can take arbitrary values. Nevertheless, we normalize our conserved operators such that the fusion coefficients for operators match with those of the defects.

## 2 Non-invertible Kramers-Wannier symmetry on a tensor product Hilbert space

Here we will study the global symmetries of the critical transverse-field Ising model. In particular, we will study its non-invertible symmetry operator and defect in detail. The Hamiltonian of the theory on a finite periodic chain with $L$ sites is given by

$$H = -\sum_{j=1}^{L} \left( Z_{j-1} Z_j + X_j \right), \tag{10}$$

where $X_j = X_{j+L}$ and $Z_j = Z_{j+L}$. The Hilbert space $\mathcal{H} = \mathcal{H}_1 \otimes \cdots \otimes \mathcal{H}_L$ is a tensor product of two dimensional Hilbert spaces $\mathcal{H}_j = \mathbb{C}^2$ for each site $j = 1, \cdots, L$.

While we will mostly focus on the Ising Hamiltonian (10), most of our conclusions apply to more general Hamiltonians, gapped or gapless, with the non-invertible symmetry. In Section 4, we will consider deformations away from the critical Ising Hamiltonian (10) while preserving all the symmetries

Many of the results in this section were known in the literature, such as in [13, 14, 105]. Here we emphasize the role of the lattice translation, and provide a streamlined discussion combining the operators and the defects in the setup of a Hamiltonian lattice model.

### 2.1 The invertible symmetries

Before discussing the non-invertible symmetry, we discuss some of the ordinary invertible symmetries of the model.

#### 2.1.1 $\mathbb{Z}_2$ symmetry

The most obvious symmetry of the Ising model is its internal, on-site $\mathbb{Z}_2$ spin flip symmetry. It is generated by

$$\eta = \prod_{j=1}^{L} X_j, \tag{11}$$

which acts on the local operators as $\eta : X_j \mapsto X_j, Z_j \mapsto -Z_j$.[10]

Associated with the $\mathbb{Z}_2$ symmetry operator (11) is the defect Hamiltonian

$$H_\eta^{(L,1)} = -(-Z_L Z_1 + X_1) - \sum_{j=2}^{L} (Z_{j-1} Z_j + X_j), \tag{12}$$

where the symmetry defect is at link $(L, 1)$.

As in Section 1.5, this defect is *topological* in the sense that we can move it by conjugating the Hamiltonian with a local unitary operator, the *movement operator*, $U_\eta^j = X_j$. For instance,

$$H_\eta^{(1,2)} = U_\eta^1 H_\eta^{(L,1)} (U_\eta^1)^{-1}, \quad \text{where} \quad H_\eta^{(1,2)} = -(-Z_1 Z_2 + X_2) - \sum_{\substack{j=1 \\ j\neq 2}}^{L} (Z_{j-1} Z_j + X_j). \tag{13}$$

---

[10]In this paper, given an *invertible* operator $U$, '$\mapsto$' stands for conjugation, i.e. $U : O \mapsto U O U^{-1}$.

More generally, we diagrammatically represent the movement of the defect as

$$U_\eta^j = X_j =$$

(14)

where the second equality also indicates $H_\eta^{(j,j+1)} = U_\eta^j H_\eta^{(j,j-1)} (U_\eta^j)^{-1}$.

We note that the symmetry operator (11) is constructed as a product of the movement operators $U_\eta^j$ that move the defect around the chain. This is a general feature reflecting a one-to-one correspondence between topological defects and symmetry operators; see [53] for a general discussion.

As in the discussion around Figure 2, we now define the fusion of two $\mathbb{Z}_2$ defects $\eta$. We start with the defect Hamiltonian with one $\eta$ defect at the link $(J-1,J)$ and another one at $(J'-1,J')$ with $J < J'$. To fuse these two defects, we first apply a sequence of movement operators $U_\eta^j$ to move the right defect to $(J, J+1)$ to find the following defect Hamiltonian:

$$H_{\eta;\eta}^{(J-1,J);(J,J+1)} = -(-Z_{J-1}Z_J + X_J) - (-Z_J Z_{J+1} + X_{J+1}) - \sum_{\substack{j=1 \\ j \neq J, J+1}}^{L} (Z_{j-1}Z_j + X_j).$$

(15)

We follow the notation above where the subscripts denote the kind of defects and the superscripts denote their location on the lattice. Next, we apply a *fusion operator* $\lambda_{\eta \otimes \eta}^J = X_J$ to pair annihilate these two adjacent defects:[11]

$$\lambda_{\eta \otimes \eta}^J H_{\eta;\eta}^{(J-1,J);(J,J+1)} (\lambda_{\eta \otimes \eta}^J)^{-1} = H.$$

(16)

We interpret this unitary transformation as the fusion between two $\mathbb{Z}_2$ defects:

$$\eta \otimes \eta = 1.$$

(17)

The fusion operator in (16) can be diagrammatically represented as

$$\lambda_{\eta \otimes \eta}^J = X_J =$$

.

(18)

More generally, the fusion operation '$\otimes$' for two topological lattice defects $\mathcal{A} \otimes \mathcal{B}$ is defined as follows. We first insert $\mathcal{A}$ and $\mathcal{B}$ away from each other, such that the corresponding deformations of the Hamiltonian do not overlap. Next, we apply a sequence of movement operators to bring them adjacent to each other. Finally, we apply a fusion operator to simplify the defect Hamiltonian in terms of (simple) defects. Since each step is implemented by a local unitary operator, this establishes the equivalence between the initial Hamiltonian of two separated defects and the final Hamiltonian of defects at a single location.

---

[11]Note that even though the fusion operator $\lambda_{\eta \otimes \eta}^j$ coincides with the movement operator $U_\eta^j$ for the $\eta$ defect, these two unitary operators are generally different for other defects.

### 2.1.2 Lattice translation symmetry

Another important symmetry is the lattice translation $T$ acting as

$$TX_jT^{-1} = X_{j+1}, \qquad TZ_jT^{-1} = Z_{j+1}. \tag{19}$$

We can write the translation symmetry operator $T$ on a finite periodic chain as a product of local *swap* operators. Namely,

$$T^{-1} = \mathsf{S}_{L,L-1}\,\mathsf{S}_{L-1,L-2}\cdots\mathsf{S}_{4,3}\,\mathsf{S}_{3,2}\,\mathsf{S}_{2,1}, \tag{20}$$

where $\mathsf{S}_{j,k} = \frac{1}{2}(1 + X_jX_k + Y_jY_k + Z_jZ_k)$ is the swap operator that exchanges the $j$-th and $k$-th qubits.

In Appendix E, we discuss topological defects for lattice translation symmetry and construct the translation symmetry operator from those defects.

## 2.2 The non-invertible symmetry defects

Having discussed the operators and defects associated with the invertible symmetries, we now move on to the non-invertible symmetry. To motivate this novel symmetry, we note that the Ising Hamiltonian (10) is invariant under the Kramers-Wannier transformation[12]

$$\begin{aligned}
X_j &\rightsquigarrow Z_{j-1}Z_j, \\
Z_{j-1}Z_j &\rightsquigarrow X_{j-1}.
\end{aligned} \tag{21}$$

Performing this transformation twice shifts these operators by one site to the left – it acts on them as $T^{-1}$. The above transformation defines an automorphism of the algebra of $\mathbb{Z}_2$ invariant operators.

However, (21) cannot possibly be implemented by a unitary operator on a finite periodic chain (and hence the notation '$\rightsquigarrow$'). To see that, suppose it were implemented by a unitary operator $U$, then $U\eta U^{-1} = U\prod_{j=1}^{L}X_jU^{-1} = \prod_{j=1}^{L}(Z_{j-1}Z_j) = 1$, which leads to the contradiction $\eta = 1$. Therefore, (21) cannot be an automorphism implemented by an invertible operator on the entire algebra on a periodic chain.

As we will see later in this section, any Hamiltonian invariant under the $\mathbb{Z}_2$ symmetry $\eta$ and the transformation above enjoys a *non-invertible* symmetry. Similar to its invertible cousins, the non-invertible symmetry is also associated with a conserved operator and a topological defect. In this subsection we will first discuss the defect associated with this symmetry and show that it obeys a non-invertible fusion rule. For this reason we refer to it as a non-invertible defect. Later, we derive the corresponding conserved operator from fusion and movement of the defects.

### 2.2.1 The non-invertible defect $\mathcal{D}$

The non-invertible topological defect $\mathcal{D}$ corresponds to the Kramers-Wannier self-duality of the theory at the critical temperature. The defect Hamiltonian is given by [9, 13, 14, 105–107][13]

$$H_{\mathcal{D}}^{(L,1)} = -\sum_{j=2}^{L}\left(Z_{j-1}Z_j + X_j\right) - Z_LX_1, \tag{22}$$

---

[12]The transformation $X_j \rightsquigarrow Z_jZ_{j+1}, Z_jZ_{j+1} \rightsquigarrow X_{j+1}$ can be obtained from composing (21) with the lattice translation by one-site to the right $T$.

[13]The defect Hamiltonian in (22) is related to the Hamiltonian $H_D$ of [13, (5.43)] as follows. The local change of variable $X_1 \mapsto Z_1$ and $Z_1 \mapsto -Y_1$ maps the Hamiltonian $H_{\mathcal{D}}^{(L,1)}$ into $Y_1Z_2 - \sum_{j=2}^{L}(X_j + Z_jZ_{j+1})$. The latter Hamiltonian is related to $2H_D$ (with $L = N + 1$) by a lattice translation and changing $X_j \leftrightarrow Z_j$.

which describes the insertion of a $\mathcal{D}$ defect on link $(L,1)$. The defect $\mathcal{D}$ is topological since there is a movement operator $U_{\mathcal{D}}^{j}$ that moves the defect from link $(j-1,j)$ to link $(j,j+1)$

$$H_{\mathcal{D}}^{(j,j+1)} = U_{\mathcal{D}}^{j} H_{\mathcal{D}}^{(j-1,j)} (U_{\mathcal{D}}^{j})^{-1}. \tag{23}$$

The movement operator is given by [14]

$$U_{\mathcal{D}}^{j} = \mathsf{CZ}_{j+1,j}\mathsf{H}_{j} \;=\; \qquad\qquad , \tag{24}$$

where $\mathsf{H}_{j} = \frac{X_{j}+Z_{j}}{\sqrt{2}}$ is the Hadamard gate and $\mathsf{CZ}_{j,k} = \frac{1}{2}(1+Z_{j}+Z_{k}-Z_{j}Z_{k})$ is the controlled-$Z$ gate. See Appendix A for more details. The movement operator acts on the local operators as

$$U_{\mathcal{D}}^{j}: \quad \begin{matrix} X_{j} \mapsto Z_{j}, & Z_{j} \mapsto X_{j}Z_{j+1}, \\ X_{j+1} \mapsto Z_{j}X_{j+1}, & Z_{j+1} \mapsto Z_{j+1}, \end{matrix} \tag{25}$$

which can be used to verify equation (23).

### 2.2.2 Defect fusion rules

In Section 2.1 we defined the fusion $\eta \otimes \eta = 1$ between two invertible, $\mathbb{Z}_2$ defects $\eta$. We now move on to the fusion rules involving the non-invertible defect $\mathcal{D}$.

We start with the fusion between $\eta$ and $\mathcal{D}$. Consider the defect Hamiltonian of $\eta$ at link $(L,1)$ and $\mathcal{D}$ at link $(1,2)$ (generalizations to other locations are straightforward):

$$H_{\eta;\mathcal{D}}^{(L,1);(1,2)} = -(-Z_{L}Z_{1}+X_{1})-Z_{1}X_{2}-\sum_{j=3}^{L}(Z_{j-1}Z_{j}+X_{j}). \tag{26}$$

Next, we apply the fusion operator $\lambda_{\eta\otimes\mathcal{D}}^{1} = X_{1}Z_{2}$ to annihilate the $\eta$ defect:

$$\lambda_{\eta\otimes\mathcal{D}}^{1} H_{\eta;\mathcal{D}}^{(L,1);(1,2)} (\lambda_{\eta\otimes\mathcal{D}}^{1})^{-1} = H_{\mathcal{D}}^{(1,2)}. \tag{27}$$

We diagrammatically denote it by

$$\lambda_{\eta\otimes\mathcal{D}}^{1} = X_{1}Z_{2} \;=\; \qquad\qquad . \tag{28}$$

Similarly, we can start with the defect Hamiltonian of $\mathcal{D}$ at $(L,1)$ and $\eta$ at $(1,2)$:

$$H_{\mathcal{D};\eta}^{(L,1);(1,2)} = -Z_{L}X_{1}-(-Z_{1}Z_{2}+X_{2})-\sum_{j=3}^{L}(Z_{j-1}Z_{j}+X_{j}). \tag{29}$$

The fusion operator is then $\lambda_{\mathcal{D}\otimes\eta}^{1} = U_{\mathcal{D}}^{1}X_{1}$:

$$\lambda_{\mathcal{D}\otimes\eta}^{1} H_{\mathcal{D};\eta}^{(L,1);(1,2)} (\lambda_{\mathcal{D}\otimes\eta}^{1})^{-1} = H_{\mathcal{D}}^{(1,2)}, \tag{30}$$

which is diagrammatically represented as

$$\lambda^1_{\mathcal{D}\otimes\eta} = U^1_{\mathcal{D}} X_1 \;= \tag{31}$$

As in Section 2.1, we interpret the above two unitary operations as the fusion

$$\eta \otimes \mathcal{D} = \mathcal{D} \otimes \eta = \mathcal{D}. \tag{32}$$

The notation '$\otimes$' corresponds to the fusion operation. Namely, we start with a defect Hamiltonian with the insertion of the $\eta$ and $\mathcal{D}$ defects at two separate locations, and then apply a unitary transformation, the fusion operator, to simplify it in terms of other defects.

Next, we move on to the more complicated fusion between two non-invertible defects $\mathcal{D}$, which was discussed in [105]. Consider the Hamiltonian for two $\mathcal{D}$ defects on links $(L,1)$ and $(1,2)$, which we denote by

$$H^{(L,1);(1,2)}_{\mathcal{D};\mathcal{D}} = -\sum_{j=3}^{L}\left(Z_{j-1}Z_j + X_j\right) - Z_L X_1 - Z_1 X_2. \tag{33}$$

Conjugating this defect Hamiltonian with the fusion operator $\lambda^1_{\mathcal{D}\otimes\mathcal{D}} = (U^1_{\mathcal{D}})^{-1}$ we find

$$\lambda^1_{\mathcal{D}\otimes\mathcal{D}} H^{(L,1);(1,2)}_{\mathcal{D};\mathcal{D}} (\lambda^1_{\mathcal{D}\otimes\mathcal{D}})^{-1} = -\sum_{j=3}^{L}\left(Z_{j-1}Z_j + X_j\right) - (Z_1)Z_L Z_2 - X_2. \tag{34}$$

We note that this defect Hamiltonian commutes with $Z_1$ and the latter can be diagonalized.[14] As we explain below, different eigenspaces of $Z_1$ correspond to two different fusion channels. Using the projection operators $(1 \pm Z_1)/2$, we rewrite the above defect Hamiltonian as a sum of two terms

$$\lambda^1_{\mathcal{D}\otimes\mathcal{D}} H^{(L,1);(1,2)}_{\mathcal{D};\mathcal{D}} (\lambda^1_{\mathcal{D}\otimes\mathcal{D}})^{-1} = H^1_{\mathcal{T}^-} \otimes |0\rangle\langle 0|_1 + H^1_{\mathcal{T}^-\eta} \otimes |1\rangle\langle 1|_1, \tag{35}$$

where

$$H^1_{\mathcal{T}^-} = -(Z_L Z_2 + X_2) - \sum_{j=3}^{L}(Z_{j-1}Z_j + X_j),$$

$$H^1_{\mathcal{T}^-\eta} = -(-Z_L Z_2 + X_2) - \sum_{j=3}^{L}(Z_{j-1}Z_j + X_j), \tag{36}$$

act on the $2^{L-1}$-dimensional Hilbert space $\mathcal{H}_2 \otimes \cdots \otimes \mathcal{H}_L$. The defect Hamiltonian $H^1_{\mathcal{T}^-}$ corresponds to removing site 1 from the chain and considering sites $L$ and 2 to be nearest neighbors. The defect Hamiltonian $H^1_{\mathcal{T}^-\eta}$ corresponds to removing site 1 and also inserting an $\eta$ defect on link $(L,2)$. (See Appendix E, for a more detailed discussion of these defect Hamiltonians.)

We interpret equation (35) as the *non-invertible* fusion rule

$$\mathcal{D} \otimes \mathcal{D} = \mathcal{T}^- \oplus \mathcal{T}^-\eta. \tag{37}$$

---

[14]This means that this system has another $\mathbb{Z}_2$ symmetry generated by $Z_1$. See the more general discussion about it in footnote 25.

The notation '$\oplus$' denotes a *direct sum* operation. This terminology reflects the fact that the Hilbert space of the defect $\mathcal{A} \oplus \mathcal{B}$ is the direct sum of the Hilbert space with the defect $\mathcal{A}$ and the Hilbert space with the defect $\mathcal{B}$, i.e., $\mathcal{H}_{\mathcal{A} \oplus \mathcal{B}} = \mathcal{H}_{\mathcal{A}} \oplus \mathcal{H}_{\mathcal{B}}$.

The fusion operator $\lambda^1_{\mathcal{D} \otimes \mathcal{D}}$, used in (35), is a unitary operator that implements the fusion $\mathcal{D} \otimes \mathcal{D} = \mathcal{T}^- \oplus \mathcal{T}^- \eta$. We diagrammatically denote it by

$$\lambda^1_{\mathcal{D} \otimes \mathcal{D}} = \mathsf{H}_1 \mathsf{CZ}_{1,2} = \qquad\qquad . \tag{38}$$

Note that this fusion operator acts on the original Hilbert space $\mathcal{H}_1 \otimes \cdots \otimes \mathcal{H}_L$.[15]

We note that the topological defect $\mathcal{T}^-$, associated with removing a site, corresponds to the lattice translation symmetry. We can see this in several ways. First, recall that on a periodic chain with $L$ sites, imposing a symmetry twist $g$ corresponds to modifying the relation $T^L = 1$ to $T^L_g = U^{-1}_g$ where $T_g$ is the translation symmetry of the system with a $g$-defect and $U_g$ is the symmetry operator. Inserting a $\mathcal{T}^-$ defect corresponds to a system with $L-1$ sites which indeed have a translation symmetry $T$ satisfying $T^L = T$. The latter equation can be interpreted as the analog of $T^L_g = U^{-1}_g$, for $U_g = T^{-1}$.

Another way to see this is to note that moving the translation defect around the chain generates the translation symmetry operator. More precisely, we will construct the lattice translation symmetry operator $T$ in Appendix E as the unitary operator that implements the following sequence of moves: We start with the untwisted Hamiltonian and pair create translation defects $\mathcal{T}^-$ and its dual $\mathcal{T}^+$, then we move the $\mathcal{T}^-$ defect around the chain and bring it next to $\mathcal{T}^+$ and fuse them together to get back to the untwisted Hamiltonian.

In summary we find the fusion rule $\mathcal{D} \otimes \mathcal{D} = \mathcal{T}^- \oplus \mathcal{T}^- \eta = (1 \oplus \eta) \otimes \mathcal{T}^-$ where $\mathcal{T}^-$ is a lattice translation symmetry defect associated with removing a site and

$$\mathcal{T}^- \eta = \mathcal{T}^- \otimes \eta = \eta \otimes \mathcal{T}^-, \tag{40}$$

is a simple defect obtained by the fusion of $\mathcal{T}^-$ with the $\mathbb{Z}_2$ defect $\eta$. We call a defect *simple* (or irreducible) if it cannot be written as a direct sum of two topological defects. Equivalently, a defect is simple if there is no non-trivial *local* operator that commutes with the defect Hamiltonian. For instance, the defect $\mathcal{T}^- \oplus \mathcal{T}^- \eta$ on the righthand side of (34) is not simple because the local operator $Z_1$ commutes with the defect Hamiltonian. We will explain the fusion (40) in Appendix E in details.

The list of all simple defects obtained by fusing these defects are

$$\mathcal{T}^n, \qquad \mathcal{T}^n \eta = \mathcal{T}^n \otimes \eta = \eta \otimes \mathcal{T}^n, \qquad \mathcal{T}^n \mathcal{D} = \mathcal{T}^n \otimes \mathcal{D} = \mathcal{D} \otimes \mathcal{T}^n. \tag{41}$$

Here, $\mathcal{T}^0 = 1$ is the trivial defect, $\mathcal{T}^{|n|} = \mathcal{T}^+ \otimes \cdots \otimes \mathcal{T}^+$ (with $|n|$ number of $\mathcal{T}^+$) is the defect associated with adding $|n|$ sites, and $\mathcal{T}^{-|n|} = \mathcal{T}^- \otimes \cdots \otimes \mathcal{T}^-$ is associated with removing $|n|$ sites for any integer $n$. The minimal list of fusion rules are:

$$\eta \otimes \eta = 1, \quad \eta \otimes \mathcal{D} = \mathcal{D} \otimes \eta = \mathcal{D}, \quad \mathcal{D} \otimes \mathcal{D} = \mathcal{T}^- \oplus \mathcal{T}^- \eta, \quad \mathcal{T}^n \otimes \mathcal{T}^m = \mathcal{T}^{n+m}. \tag{42}$$

---

[15]Alternatively, we can interpret the fusion operator $\lambda^1_{\mathcal{D} \otimes \mathcal{D}}$ as a map

$$\lambda^1_{\mathcal{D} \otimes \mathcal{D}} \quad : \quad \mathcal{H}_1 \otimes \cdots \otimes \mathcal{H}_L \to (\mathcal{H}_2 \otimes \cdots \otimes \mathcal{H}_L) \oplus (\mathcal{H}_2 \otimes \cdots \otimes \mathcal{H}_L), \tag{39}$$

reflecting the fusion relation $\mathcal{D} \otimes \mathcal{D} = \mathcal{T}^- \oplus \mathcal{T}^- \eta$. More generally, we denote the fusion operator $\lambda_{\mathcal{A} \otimes \mathcal{B}}$ that implements the fusion $\mathcal{A} \otimes \mathcal{B} = \mathcal{C}_1 \oplus \cdots \oplus \mathcal{C}_d$, as a map from $\mathcal{H}_{\mathcal{A};\mathcal{B}}$ to $\mathcal{H}_{\mathcal{C}_1} \oplus \cdots \oplus \mathcal{H}_{\mathcal{C}_d}$.

Note that in a system with $L$ sites, $\mathcal{T}^n$ exist only for $n > -L$. Therefore, the last fusion relation in (42) is meaningful only when $n, m, n + m > -L$.

Finally, we define the *dual* defect of $\mathcal{D}$,

$$\mathcal{D}^* = \mathcal{T}^+ \otimes \mathcal{D}. \tag{43}$$

It is the dual of $\mathcal{D}$ in the sense that

$$\mathcal{D} \otimes \mathcal{D}^* = \mathcal{D}^* \otimes \mathcal{D} = 1 \oplus \eta, \tag{44}$$

contains the identity defect on the righthand side. While the defect $\mathcal{D}$ does not change the Hilbert space, its dual defect $\mathcal{D}^*$ adds one qubit to the Hilbert space since it involves the translation defect $\mathcal{T}^+$. See Appendix E.3 for more discussions on $\mathcal{D}^*$.

## 2.3 The non-invertible symmetry operators

In this section, we will present several different expressions for the non-invertible operator D. The most elementary expression is [13]

$$\mathsf{D} = e^{2\pi i \frac{L}{8}} \frac{1 + \eta}{\sqrt{2}} \frac{1 - iX_L}{\sqrt{2}} \frac{1 - iZ_L Z_{L-1}}{\sqrt{2}} \frac{1 - iX_{L-1}}{\sqrt{2}} \cdots \frac{1 - iZ_2 Z_1}{\sqrt{2}} \frac{1 - iX_1}{\sqrt{2}}. \tag{45}$$

(The phase $e^{2\pi i L/8}$ was added for convenience.) This operator does not appear to be translation invariant. Also, it is not manifest how it acts on local operators. This issue is related to the projection operator on the left and the fact that the local unitary operators in this expression, $\frac{1 - iX_j}{\sqrt{2}}$ and $\frac{1 - iZ_j Z_{j-1}}{\sqrt{2}}$, do not commute with each other. Below, we will present equivalent expressions for D that make it clear that it is translation invariant and its locality properties will also be clarified.

In Section 2.3.1, we will derive an expression for D by manipulating the defect $\mathcal{D}$.[16] Later, we will relate it to other perspectives. Specifically, in Section 2.3.2, we will provide a matrix product operator expression for D. In Appendix B.3, it will be presented as implementing $\mathbb{Z}_2$ gauging in the future (or in the past). Appendix C will discuss its relation to the Majorana lattice translation, and Appendix D will present its relation to the sequential quantum circuit of [25].

### 2.3.1 Non-invertible operator D from the defect $\mathcal{D}$

Here we construct the non-invertible conserved operator D from the symmetry defect $\mathcal{D}$. To construct the symmetry operator, we first construct a *unitary operator* $\mathsf{D}_{1 \oplus \eta}$ that acts on the extended Hilbert space

$$\mathcal{H}_{1 \oplus \eta} = \mathcal{H} \oplus \mathcal{H}, \tag{46}$$

where $\mathcal{H}$ is the Hilbert space of the chain with $L$ sites. The first and second copy of $\mathcal{H}$, respectively, represent the problem without and with a $\mathbb{Z}_2$ defect. We restrict the action of $\mathsf{D}_{1 \oplus \eta}$ to the first (or second) copy of $\mathcal{H}$ to find the non-invertible symmetry operator D (or $\mathsf{D}_\eta$) that commutes with the original Hamiltonian $H$ (or $H_\eta^{(L,1)}$). (Recall the discussion in Section 1.5 about our notation of operators in the presence of defects.)

The idea to construct the unitary operator $\mathsf{D}_{1 \oplus \eta}$ is as follows. We start from a $1 \oplus \eta$ defect and, using the fusion rule, we split it into a pair of $\mathcal{D}$ and $\mathcal{D}^*$ defects, where $\mathcal{D}^* = \mathcal{T}^+ \otimes \mathcal{D}$.

---

[16]As always, we use different fonts for the non-invertible operator D and its corresponding defect $\mathcal{D}$. Similar distinctions are made for the lattice translation operator $T^n$ and its defect $\mathcal{T}^n$, and their counterparts with the various defects. However, for the invertible $\mathbb{Z}_2$ symmetry, we use the same symbol $\eta$ for both the operator and the defect, since this is the least subtle symmetry of all.

We then move the defect $\mathcal{D}$ around the chain and bring it near the other defect. Next, we fuse these two defects to a $1 \oplus \eta$ defect at its initial position. These moves are implemented by conjugating the defect Hamiltonian with a series of local unitary operators. Since the initial and final configurations are the same, the product of all these unitary operators define $\mathsf{D}_{1\oplus\eta}$ which commutes with the defect Hamiltonian of $1 \oplus \eta$. We will now go through this procedure in detail.

To model the direct sum of the systems with and without the $\eta$ defect, we add a qubit on link $(L, 1)$, between sites $L$ and $1$, and denote its Hilbert space by $\mathcal{H}_{1\oplus\eta}$. Then, we consider the Hamiltonian

$$H_{1\oplus\eta}^{(L,1)} = H \otimes |0\rangle\langle0|_{(L,1)} + H_{\eta}^{(L,1)} \otimes |1\rangle\langle1|_{(L,1)} = -\sum_{j=2}^{L}\big(Z_{j-1}Z_j + X_j\big) - (Z_{(L,1)})Z_L Z_1 - X_1 \,. \quad (47)$$

It acts on the Hilbert space $\mathcal{H}_{1\oplus\eta} = \mathcal{H}_{(L,1)} \otimes \mathcal{H}_1 \otimes \cdots \otimes \mathcal{H}_L = \mathcal{H} \oplus \mathcal{H}$, where $Z_{(L,1)}$ is the Pauli-$Z$ operator acting on $\mathcal{H}_{1\oplus\eta}$. The defect Hamiltonian $H_{1\oplus\eta}^{(L,1)}$ describes a (non-simple) $1 \oplus \eta$ defect on link $(L, 1)$.

Using the fusion rule $1 \oplus \eta = \mathcal{D} \otimes \mathcal{T}^+ \otimes \mathcal{D}$, we can split the defect $1 \oplus \eta$ into a pair of $\mathcal{D}$ defects in a system with $L + 1$ sites. To do that, we first relabel the link $(L, 1)$ as site number $L + 1$ and make the identifications $X_{L+1} = X_{(L,1)}$ and $Z_{L+1} = Z_{(L,1)}$. Using the inverse of (34), we find

$$(\lambda_{\mathcal{D}\otimes\mathcal{D}}^{L+1})^{-1} H_{1\oplus\eta}^{(L,1)} \lambda_{\mathcal{D}\otimes\mathcal{D}}^{L+1} = -\sum_{j=2}^{L}(Z_{j-1}Z_j + X_j) - Z_L X_{L+1} - Z_{L+1}X_1 \,. \quad (48)$$

This defect Hamiltonian describes a pair of $\mathcal{D}$ defects on links $(L, L+1)$ and $(L+1, 1)$. Using the unitary operator $U_{\mathcal{D}}^{L-1} \cdots U_{\mathcal{D}}^{2} U_{\mathcal{D}}^{1}$, we move the defect on link $(L+1, 1)$ around the chain and bring it to the left of the other defect on link $(L, L+1)$. Then, we use $\lambda_{\mathcal{D}\otimes\mathcal{D}}^{L+1} U_{\mathcal{D}}^{L} U_{\mathcal{D}}^{L+1}$ to move both of the defects one site to the right and fuse them to a $1 \oplus \eta$ defect on link $(L, 1)$, which is the initial configuration we started with. In the end, we find the unitary operator[17]

$$\mathsf{D}_{1\oplus\eta} = \lambda_{\mathcal{D}\otimes\mathcal{D}}^{L+1} U_{\mathcal{D}}^{L} U_{\mathcal{D}}^{L+1} U_{\mathcal{D}}^{L-1} \cdots U_{\mathcal{D}}^{2} U_{\mathcal{D}}^{1} (\lambda_{\mathcal{D}\otimes\mathcal{D}}^{L+1})^{-1} \,, \quad (49)$$

that commutes with the defect Hamiltonian $H_{1\oplus\eta}^{(L,1)}$. See Figure 3 for a diagrammatic expression of the operator.

The operator $\mathsf{D}_{1\oplus\eta}$ in terms of Hadamard and controlled gates is given by

$$\begin{aligned}\mathsf{D}_{1\oplus\eta} &= \mathsf{H}_{L+1}\big(\mathsf{CZ}_{L+1,L}\mathsf{H}_L\big)\big(\mathsf{CZ}_{L,L-1}\mathsf{H}_{L-1}\big)\cdots\big(\mathsf{CZ}_{2,1}\mathsf{H}_1\big)\mathsf{CNOT}_{1,L+1} \\ &= \big(\mathsf{H}_{L+1}\mathsf{H}_L\cdots\mathsf{H}_1\big)\big(\mathsf{CNOT}_{L+1,L}\mathsf{CNOT}_{L,L-1}\cdots\mathsf{CNOT}_{2,1}\mathsf{CNOT}_{1,L+1}\big)\,,\end{aligned} \quad (50)$$

where we have used $\lambda_{\mathcal{D}\otimes\mathcal{D}}^{j} = (U_{\mathcal{D}}^{j})^{-1} = (\mathsf{CZ}_{j+1,j}\mathsf{H}_j)^{-1}$ and $\mathsf{CNOT}_{j+1,j} = \mathsf{H}_j\mathsf{CZ}_{j+1,j}\mathsf{H}_j$. See Appendix A for details. The operator $\mathsf{D}_{1\oplus\eta}$ is a *unitary* operator acting on the extended Hilbert space (46) and commutes with the Hamiltonian $H_{1\oplus\eta}^{(L,1)}$. It acts on local operators as

$$\mathsf{D}_{1\oplus\eta}: \quad \begin{aligned} X_j &\mapsto Z_{j-1}Z_j & &\text{(for } j \neq 1\text{)}, \\ X_1 &\mapsto Z_L Z_{L+1} Z_1 \,, \\ Z_j &\mapsto X_j X_{j+1}\cdots X_{L+1} & &\text{(for } j \neq L+1\text{)}, \\ Z_{L+1} &\mapsto X_1 \cdots X_L \,. \end{aligned} \quad (51)$$

To find the symmetry operator that commutes with the untwisted Hamiltonian, we need to project onto the original Hilbert space given by the eigenspace $Z_{L+1} = 1$. Taking the $|0\rangle\langle0|_{L+1}$

---

[17]This unitary operator $\mathsf{D}_{1\oplus\eta}$ is the counterpart of $U$ of [108], but on a tensor product Hilbert space.

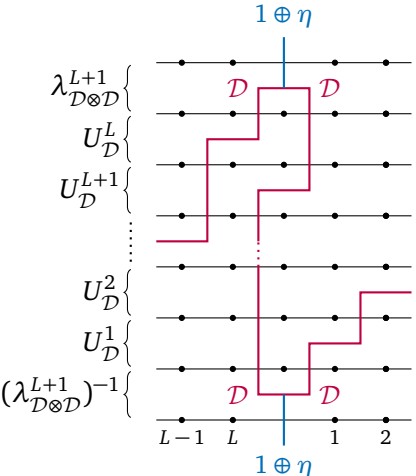

Figure 3: Construction of symmetry operator $D_{1\oplus\eta}$ from the symmetry defect $\mathcal{D}$. Recall from (38) that $\mathcal{D}\otimes\mathcal{D} = \mathcal{T}^- \oplus \mathcal{T}^-\eta$. In the first and the last row, we represent $\mathcal{T}^- \oplus \mathcal{T}^-\eta$ by $1 \oplus \eta$, but with one site less compared to the other rows.

matrix element of the equation $D_{1\oplus\eta}H_{1\oplus\eta}^{(L,1)} = H_{1\oplus\eta}^{(L,1)}D_{1\oplus\eta}$, we find

$$DH = HD, \quad \text{where} \quad D = \sqrt{2}\,_{L+1}\langle 0|\, D_{1\oplus\eta}\, |0\rangle_{L+1}. \tag{52}$$

Because of the projection, the symmetry operator $D$ is not unitary. In fact it is not even invertible. As a result, its normalization is not fixed. We added a factor of $\sqrt{2}$ in the definition of $D$ so that the formula for the fusion of operators matches with that of the defects. Furthermore, this normalization is also natural from the point of the matrix product operator presentation as we discuss below.

### 2.3.2 Matrix product operator expression for D

This non-invertible operator $D$ admits a natural presentation in terms of a Matrix Product Operator (MPO).[18] The operator $D$ is explicitly given by

$$D = H_L \left( U_{\mathcal{D}}^{L-1} \cdots U_{\mathcal{D}}^2 U_{\mathcal{D}}^1 \right) \frac{1+Z_1}{2} + Z_L H_L \left( U_{\mathcal{D}}^{L-1} \cdots U_{\mathcal{D}}^2 U_{\mathcal{D}}^1 \right) \frac{1-Z_1}{2}. \tag{53}$$

Recall that $H_j = \frac{X_j + Z_j}{\sqrt{2}}$ is the Hadamard gate. To proceed, we rewrite the movement operator (23) as

$$U_{\mathcal{D}}^j = \left( \frac{1+Z_{j+1}}{2} \quad \frac{1-Z_{j+1}}{2} \right) \begin{pmatrix} H_j \\ Z_j H_j \end{pmatrix} = \frac{1+Z_{j+1}}{2} H_j + \frac{1-Z_{j+1}}{2} Z_j H_j. \tag{54}$$

---

[18] We thank Nathanan Tantivasadakarn for extensive discussions on this point.

We can associate a $2 \times 2$ operator-valued matrix to this movement operator [15]:[19]

$$\mathbb{U}_{\mathcal{D}}^{j} = \begin{pmatrix} \mathsf{H}_j \\ Z_j \mathsf{H}_j \end{pmatrix} \begin{pmatrix} \frac{1+Z_j}{2} & \frac{1-Z_j}{2} \end{pmatrix} = \begin{pmatrix} \mathsf{H}_j \frac{1+Z_j}{2} & \mathsf{H}_j \frac{1-Z_j}{2} \\ Z_j \mathsf{H}_j \frac{1+Z_j}{2} & Z_j \mathsf{H}_j \frac{1-Z_j}{2} \end{pmatrix} = \begin{pmatrix} |+\rangle\langle 0|_j & |-\rangle\langle 1|_j \\ |-\rangle\langle 0|_j & |+\rangle\langle 1|_j \end{pmatrix}. \quad (55)$$

We can then write D as

$$\mathsf{D} = \mathrm{Tr}\left( \mathbb{U}_{\mathcal{D}}^{L} \mathbb{U}_{\mathcal{D}}^{L-1} \cdots \mathbb{U}_{\mathcal{D}}^{1} \right), \quad (56)$$

where the trace is taken over the auxiliary, or virtual, degrees of freedom associated with indices of the $2 \times 2$ matrix $\mathbb{U}_{\mathcal{D}}^{j}$. (We use *blackboard-bold* symbols for all quantities that involve auxiliary/virtual degrees of freedom. In particular, $\mathbb{X}$, $\mathbb{Y}$, and $\mathbb{Z}$ are Pauli matrices acting on the virtual degrees of freedom.) Since the auxiliary degrees of freedom are inside a two dimensional space, the MPO is said to have bond dimension 2.

Using (53) we can also write the non-invertible operator as

$$\mathsf{D} = \frac{1}{2}(1 + \eta)\mathsf{H}_L U_{\mathcal{D}}^{L-1} \cdots U_{\mathcal{D}}^{2} U_{\mathcal{D}}^{1}(1 + \eta), \quad (57)$$

which is closer to the original expression (45).

Let us compare the two expressions for the operator D, (56) and (57) (along with its close cousin (45)) with each other. The MPO presentation (56) makes the locality of D manifest. Furthermore, the cyclic property of the trace also makes the translation invariance manifest, i.e., $T\mathsf{D} = \mathsf{D}T$. On the other hand, the expression (57) makes it clear that D annihilates all the $\eta$-odd states and is therefore non-invertible. Its close cousin (45) also makes the connection to the sequential quantum circuit (reviewed in Appendix D) manifest.

Using $\eta = \prod_{j=1}^{L} X_j$ and our conventions in Appendix A, we find how $\eta$ acts on $\mathbb{U}_{\mathcal{D}}^{j}$ of (55)

$$\eta: \qquad \mathbb{U}_{\mathcal{D}}^{j} = \begin{pmatrix} |+\rangle\langle 0|_j & |-\rangle\langle 1|_j \\ |-\rangle\langle 0|_j & |+\rangle\langle 1|_j \end{pmatrix} \mapsto \begin{pmatrix} |+\rangle\langle 1|_j & -|-\rangle\langle 0|_j \\ -|-\rangle\langle 1|_j & |+\rangle\langle 0|_j \end{pmatrix} = \mathbb{Y}\mathbb{U}_{\mathcal{D}}^{j}\mathbb{Y}. \quad (58)$$

More generally, we will consider operators made out of a string of $\mathbb{U}_{\mathcal{D}}^{j}$'s as in (56) and insert at various places the bond operators $\mathbb{X}$, $\mathbb{Y}$, and $\mathbb{Z}$. In this context, we see that $\eta$ acts on the bond degrees of freedom like $\mathbb{Y}$. Therefore, the bond operators $\mathbb{X}$ and $\mathbb{Z}$ are odd under the global $\mathbb{Z}_2$ symmetry generated by $\eta$.[20]

---

[19]Our MPO differs slightly from the gauging map in [15]. The authors of that paper have a map from the Hilbert space on the sites to the Hilbert space on the links. More specifically, their tensor, up to adjoint, is $\begin{pmatrix} |+\rangle_{j-\frac{1}{2}}\langle 0|_j & |-\rangle_{j-\frac{1}{2}}\langle 1|_j \\ |-\rangle_{j-\frac{1}{2}}\langle 0|_j & |+\rangle_{j-\frac{1}{2}}\langle 1|_j \end{pmatrix}$. See also [16,17,109]. In contrast, our MPO acts in the same $2^L$-dimensional Hilbert space and every bra and ket in (55) is on the same site $j$. Here and below, whenever the bra and ket are on the same site, we will write the subscript $j$ only once. See also the discussion in Appendix B.3.

[20]Using conjugation, we can make the bond operators $\mathbb{X}$, $\mathbb{Y}$, and $\mathbb{Z}$ transform under $\eta$ like the physical operators $X$, $Y$, and $Z$. Specifically, conjugate (55) by the unitary matrix

$$\mathbb{V} = \begin{pmatrix} 0 & e^{i\pi/4} \\ e^{3i\pi/4} & 0 \end{pmatrix}, \quad (59)$$

to find

$$\mathbb{V}\mathbb{X}\mathbb{V}^{-1} = \mathbb{Y}, \qquad \mathbb{V}\mathbb{Y}\mathbb{V}^{-1} = \mathbb{X}, \qquad \mathbb{V}\mathbb{Z}\mathbb{V}^{-1} = -\mathbb{Z}. \quad (60)$$

In this basis, $\eta$ acts as $\mathbb{X}$, and therefore $\mathbb{Y}$ and $\mathbb{Z}$ are $\mathbb{Z}_2$ odd.

**Action of** D **on operators**

Since the operator D is not invertible, it cannot act on operators by conjugation. Instead, we can have expressions like $DO = O'D$ for some operators $O$ and $O'$. Here we will refer to that expression as the action of D on $O$ or $O'$. It is important to stress that such a relation does not exist for every operator $O$ and $O'$. In particular, it only exists for $\mathbb{Z}_2$-even operators.

Using the MPO presentation of the non-invertible lattice translation symmetry D, we now find its action on local operators. In particular, we will verify the transformation (21), by computing the commutation relation between D and $\mathbb{Z}_2$ invariant local operators.

The essential relations that we need are the following properties, which determine the tensor $\mathbb{U}_{\mathcal{D}}^{j}$ uniquely up to an overall normalization,

$$
\begin{aligned}
X_j \mathbb{U}_{\mathcal{D}}^{j} &= \mathbb{Z}\, \mathbb{U}_{\mathcal{D}}^{j}\, \mathbb{Z}, & \mathbb{U}_{\mathcal{D}}^{j} X_j &= \mathbb{X}\, \mathbb{U}_{\mathcal{D}}^{j}\, \mathbb{X}, \\
Z_j \mathbb{U}_{\mathcal{D}}^{j} &= \mathbb{X}\, \mathbb{U}_{\mathcal{D}}^{j}, & \mathbb{U}_{\mathcal{D}}^{j} Z_j &= \mathbb{U}_{\mathcal{D}}^{j}\, \mathbb{Z}.
\end{aligned}
\tag{61}
$$

As a check, these relations are compatible with the global $\mathbb{Z}_2$ symmetry under which $Z$, $Y$, $\mathbb{X}$, and $\mathbb{Z}$ are odd, while $X$ and $\mathbb{Y}$ are even. Using these relations we find

$$
\begin{aligned}
X_j\, \mathbb{U}_{\mathcal{D}}^{j+1}\mathbb{U}_{\mathcal{D}}^{j} &= \mathbb{U}_{\mathcal{D}}^{j+1} X_j \mathbb{U}_{\mathcal{D}}^{j} = \mathbb{U}_{\mathcal{D}}^{j+1}\mathbb{Z}\mathbb{U}_{\mathcal{D}}^{j}\mathbb{Z} = \mathbb{U}_{\mathcal{D}}^{j+1} Z_{j+1}\mathbb{U}_{\mathcal{D}}^{j} Z_j = \mathbb{U}_{\mathcal{D}}^{j+1}\mathbb{U}_{\mathcal{D}}^{j} Z_j Z_{j+1}, \\
Z_j Z_{j-1}\mathbb{U}_{\mathcal{D}}^{j}\mathbb{U}_{\mathcal{D}}^{j-1} &= Z_j \mathbb{U}_{\mathcal{D}}^{j} Z_{j-1}\mathbb{U}_{\mathcal{D}}^{j-1} = \mathbb{X}\mathbb{U}_{\mathcal{D}}^{j}\mathbb{X}\mathbb{U}_{\mathcal{D}}^{j-1} = \mathbb{U}_{\mathcal{D}}^{j} X_j \mathbb{U}_{\mathcal{D}}^{j-1} = \mathbb{U}_{\mathcal{D}}^{j}\mathbb{U}_{\mathcal{D}}^{j-1} X_j,
\end{aligned}
\tag{62}
$$

which leads to the following commutation relations, implying the transformations (21),

$$
X_j \mathsf{D} = \mathsf{D} Z_j Z_{j+1}, \quad \text{and} \quad Z_{j-1} Z_j \mathsf{D} = \mathsf{D} X_j.
\tag{63}
$$

The non-invertible symmetry operator D relates the correlation functions of the order operator to those of the disorder operator. Suppose that we have a state $|\Omega\rangle$ preserving the non-invertible symmetry, say $\mathsf{D}|\Omega\rangle = \sqrt{2}|\Omega\rangle$, and therefore $\eta|\Omega\rangle = |\Omega\rangle$ and $T|\Omega\rangle = |\Omega\rangle$. Then the two-point function of the order operator $Z_j$ equals that of the disorder operator:

$$
\langle\Omega|Z_{j_1} Z_{j_2}|\Omega\rangle = \langle\Omega|X_{j_1+1} X_{j_1+2}\cdots X_{j_2}|\Omega\rangle,
\tag{64}
$$

and depends only on $j_2 - j_1$ and not on $j_1$ and $j_2$ separately. In deriving this, use $\langle\Omega|\mathsf{D} = (\mathsf{D}^{\dagger}|\Omega\rangle)^{\dagger} = (T\mathsf{D}|\Omega\rangle)^{\dagger} = \sqrt{2}\langle\Omega|$. (Note that since $\eta|\Omega\rangle = |\Omega\rangle$, the string of $X'$s can also run through the other direction of the chain.)

### 2.3.3 The duality operator $\mathsf{D}_{\eta}$ of the $\mathbb{Z}_2$-twisted Hamiltonian

In the previous section, we found the non-invertible symmetry D that acts on the periodic chain. Here, we construct the non-invertible symmetry $\mathsf{D}_{\eta}$ that commutes with the $\mathbb{Z}_2$-twisted Hamiltonian $H_{\eta}^{(L,1)}$, by taking the $|1\rangle\langle 1|_{L+1}$ matrix element of $\mathsf{D}_{1\oplus\eta}$.

Recall that $\mathsf{D}_{1\oplus\eta}$ commutes with the defect Hamiltonian $H_{1\oplus\eta}^{(L,1)}$ of equation (47). In the $2 \times 2$ matrix presentation of the Pauli operator $Z_{L+1} = Z_{(L,1)}$, we have

$$
H_{1\oplus\eta}^{(L,1)} = \begin{pmatrix} H & 0 \\ 0 & H_{\eta}^{(L,1)} \end{pmatrix}, \quad \text{and} \quad \mathsf{D}_{1\oplus\eta} = \frac{1}{\sqrt{2}}\begin{pmatrix} \mathsf{D} & \mathsf{D}_{\eta\to 1} \\ \mathsf{D}_{1\to\eta} & \mathsf{D}_{\eta} \end{pmatrix}.
\tag{65}
$$

See (F.21) for the definition of the off-diagonal elements $\mathsf{D}_{\eta\to 1}$ and $\mathsf{D}_{1\to\eta}$, which intertwine $H$ with $H_{\eta}^{(L,1)}$. Taking the $|1\rangle\langle 1|_{L+1}$ matrix element of $\mathsf{D}_{1\oplus\eta} H_{1\oplus\eta}^{(L,1)} = H_{1\oplus\eta}^{(L,1)} \mathsf{D}_{1\oplus\eta}$ leads to

$$
\mathsf{D}_{\eta} H_{\eta}^{(L,1)} = H_{\eta}^{(L,1)} \mathsf{D}_{\eta}, \quad \text{where} \quad \mathsf{D}_{\eta} = \sqrt{2}\langle 1|\mathsf{D}_{1\oplus\eta}|1\rangle_{L+1}.
\tag{66}
$$

Using equation (50) and the results in Appendix F.3, the MPO presentation of the unitary operator $D_{1\oplus\eta}$ is given by

$$D_{1\oplus\eta} = \mathrm{Tr}\left(\mathbb{D}_{1\oplus\eta}^{L+1} \mathbb{U}_{\mathcal{D}}^{L} \cdots \mathbb{U}_{\mathcal{D}}^{1}\right), \quad \text{where} \quad \mathbb{D}_{1\oplus\eta}^{L+1} = \begin{pmatrix} |+\rangle\langle 0|_{L+1} & |-\rangle\langle 1|_{L+1} \\ |+\rangle\langle 1|_{L+1} & |-\rangle\langle 0|_{L+1} \end{pmatrix}. \tag{67}$$

To find the MPO presentation of $D_\eta$, we take the $|1\rangle\langle 1|_{L+1}$ component of the tensor $\mathbb{D}_{1\oplus\eta}^{L+1}$, multiplied by $\sqrt{2}$, to find

$$D_\eta = \mathrm{Tr}\left(\mathbb{X}\mathbb{Z}\, \mathbb{U}_{\mathcal{D}}^{L} \cdots \mathbb{U}_{\mathcal{D}}^{1}\right), \quad \text{where} \quad \mathbb{X}\mathbb{Z} = \begin{pmatrix} 0 & -1 \\ 1 & 0 \end{pmatrix}. \tag{68}$$

The insertion of $\mathbb{X}\mathbb{Z}$ can be interpreted as due to the action of the symmetry operator $\eta$ on the bond variables (as in (58)).

In this presentation, it is straightforward to find the relation between $D_\eta$ and $D$. Using the relations in the second line of (61) and equation (57), we find

$$D_\eta = -Z_L D Z_1 = \frac{1}{2}(1-\eta)\, H_L U_{\mathcal{D}}^{L-1} \cdots U_{\mathcal{D}}^{2} U_{\mathcal{D}}^{1}(1-\eta). \tag{69}$$

### 2.3.4 The duality operator $D_{\mathcal{D}}$ of the duality-twisted Hamiltonian

Here, we consider the non-invertible symmetry operator of the $\mathcal{D}$-twisted Hamiltonian $H_{\mathcal{D}}^{(L,1)}$ of equation (22). This case is simpler than the other cases discussed above. The duality operator of the defect Hamiltonian $H_{\mathcal{D}}^{(L,1)}$ is simply given by the product of the movement operators

$$D_{\mathcal{D}} = U_{\mathcal{D}}^{L} U_{\mathcal{D}}^{L-1} \cdots U_{\mathcal{D}}^{2} U_{\mathcal{D}}^{1}, \tag{70}$$

which moves the defect around the periodic chain. Importantly, the duality operator $D_{\mathcal{D}}$ in the theory with a duality defect is unitary, and in particular, invertible. This is unlike its counterparts $D$ and $D_\eta$ in the untwisted and $\mathbb{Z}_2$-twisted problems.

It will be useful, especially for the computation of the operator fusion algebra, to find the MPO presentation of $D_{\mathcal{D}}$. Using equations (54) and (55), we rewrite (70) as

$$D_{\mathcal{D}} = \begin{pmatrix} \frac{1+Z_1}{2} & \frac{1-Z_1}{2} \end{pmatrix} \mathbb{U}_{\mathcal{D}}^{L} \mathbb{U}_{\mathcal{D}}^{L-1} \cdots \mathbb{U}_{\mathcal{D}}^{2} \begin{pmatrix} H_1 \\ Z_1 H_1 \end{pmatrix}$$

$$= \mathrm{Tr}\left(\mathbb{U}_{\mathcal{D}}^{L} \mathbb{U}_{\mathcal{D}}^{L-1} \cdots \mathbb{U}_{\mathcal{D}}^{2} \mathbb{D}_{\mathcal{D}}^{1}\right), \tag{71}$$

where

$$\mathbb{D}_{\mathcal{D}}^{1} = \begin{pmatrix} \frac{1+Z_1}{2} H_1 & \frac{1-Z_1}{2} H_1 \\ \frac{1+Z_1}{2} Z_1 H_1 & \frac{1-Z_1}{2} Z_1 H_1 \end{pmatrix} = \begin{pmatrix} |0\rangle\langle +|_1 & |1\rangle\langle -|_1 \\ |0\rangle\langle +|_1 & -|1\rangle\langle -|_1 \end{pmatrix}. \tag{72}$$

## 2.4 The operator algebra

Here we present the operator fusion algebra of the invertible and non-invertible operators on a closed chain with no defect, with a $\mathbb{Z}_2$ defect, and with a duality defect. We leave some of the derivations to Appendix F.

### 2.4.1 No defect

We begin with the symmetry operators that commute with the untwisted Hamiltonian $H$, given in (10), defined on the periodic chain with $L$ sites. These symmetries are generated by

$$\eta = X_L \cdots X_1, \quad T^{-1} = \text{Tr}\left(\mathbb{U}^L_{\mathcal{T}^-}\mathbb{U}^{L-1}_{\mathcal{T}^-}\cdots\mathbb{U}^1_{\mathcal{T}^-}\right), \quad \text{D} = \text{Tr}\left(\mathbb{U}^L_{\mathcal{D}}\mathbb{U}^{L-1}_{\mathcal{D}}\cdots\mathbb{U}^1_{\mathcal{D}}\right), \tag{73}$$

where the tensors $\mathbb{U}^j_{\mathcal{T}^-}$ and $\mathbb{U}^j_{\mathcal{D}}$ are given by (E.18) and (55). Alternative presentations of $T^{-1}$ are given in equations (20) and (E.16).

These operators satisfy the algebra [13]

$$\begin{aligned}
\eta^2 &= 1, & T\eta &= \eta T, & T^L &= 1, \\
\text{D}\eta &= \eta\text{D} = \text{D}, & T\text{D} &= \text{D}T = \text{D}^\dagger, & \text{D}^2 &= (1+\eta)T^{-1}.
\end{aligned} \tag{74}$$

Here, $\text{D}^\dagger = \text{D}T$ acts on our Hilbert space as the Hermitian conjugate of $\text{D}$.[21] For the defects, our notation is such that the dual of the defect $\mathcal{D}$ is $\mathcal{D}^*$. The relations in the first line of (74) are standard. The relation $\eta\text{D} = \text{D}\eta = \text{D}$ follows from (45). The relation $\text{D}T = T\text{D}$ follows from the cyclic property of the trace and the fact that $T\mathbb{U}^j_{\mathcal{D}}T^{-1} = \mathbb{U}^{j+1}_{\mathcal{D}}$. Finally, see Appendices F.2 and F.3 for the relations involving $\text{D}^2$ and $\text{D}^\dagger$.[22]

Let us compare (74) with the continuum fusion algebra (6) reviewed in Appendix G. In the continuum, the non-invertible Kramers-Wannier topological operator $\mathcal{N}$ is internal and obeys $\mathcal{N}^2 = 1 + \eta$. In contrast, the lattice non-invertible symmetry D mixes with the lattice translation. For this reason, D is referred to as the non-invertible lattice translation in [13]. See Section 3.1, for more comparisons between the lattice and the continuum.

Finally, we can also consider the action of parity P and time-reversal K, which acts in our basis as complex conjugation combined with reversing the time. As always

$$\begin{aligned}
\text{K}T &= T\text{K}, & \text{K}\eta &= \eta\text{K}, \\
\text{P}T &= T^{-1}\text{P}, & \text{P}\eta &= \eta\text{P}.
\end{aligned} \tag{75}$$

Using the MPO expression (56) and (F.13), we find

$$\begin{aligned}
\text{K}\text{D} &= \text{D}\text{K}, \\
\text{P}\text{D} &= \text{D}^\dagger\text{P}.
\end{aligned} \tag{76}$$

### 2.4.2 With a $\mathbb{Z}_2$ defect

Here we discuss the symmetry operators of the problem with a $\mathbb{Z}_2$ defect, which is described by the defect Hamiltonian $H^{(L,1)}_\eta$ of equation (12). The generating symmetry operators are

$$\eta_\eta = X_L \cdots X_1, \quad T^{-1}_\eta = \text{Tr}\left(\mathbb{X}\,\mathbb{U}^L_{\mathcal{T}^-}\cdots\mathbb{U}^1_{\mathcal{T}^-}\right), \quad \text{D}_\eta = \text{Tr}\left(\mathbb{X}\mathbb{Z}\,\mathbb{U}^L_{\mathcal{D}}\cdots\mathbb{U}^1_{\mathcal{D}}\right). \tag{77}$$

---

[21]For unitary symmetry operators, such a Hermitian conjugate is the inverse of the operator. But recall that D is not unitary.

[22]As far as the fusion algebra (74) is concerned, it is a subalgebra of the Ising algebra and a $\mathbb{Z}_{2L}$ algebra. But as we stressed in Section 1.2, it does not have the full-fledged structure of a fusion category because of the mixing with the lattice translation. If we denote the non-trivial invertible and non-invertible elements of the Ising algebra by $\eta$ and $\text{D}_0$, and the generator of $\mathbb{Z}_{2L}$ by $T_0$, then the relation is given by $\text{D} = \text{D}_0 T_0^{-1}$ and $T = T_0^2$. Intuitively, even though $T_0$ is not a symmetry of our problem, the relation $T = T_0^2$ identifies it as "half-translation", in the spirit of [110]. Then, the relation $\text{D} = \text{D}_0 T_0^{-1}$ clarifies in what sense D is associated with "half-translation."

In the special case of odd $L$, we can use the fact that $\mathbb{Z}_{2L} = \mathbb{Z}_L \times \mathbb{Z}_2$ to find a stronger statement. Instead of writing $\text{D} = \text{D}_0 T_0^{-1}$, we write $\text{D} = \text{D}_0 T_0^{L-1} = \text{D}_0 T^{\frac{L-1}{2}}$. Then, the elements in (74) are expressed in terms of the generators $\eta, \text{D}_0$ and $T = T_0^2$ of the Ising algebra and a $\mathbb{Z}_L$ algebra. However, these relations obscure the spatial locality of the problem and we will not pursue them. We thank Eric Rowell and Zhenghan Wang for a useful discussion about these facts.

The relations between the symmetry operators with or without a $\mathbb{Z}_2$ defect are[23]

$$\eta_\eta = \eta, \qquad T_\eta^{-1} = T^{-1}X_1, \qquad \mathsf{D}_\eta = -Z_L \mathsf{D} Z_1. \tag{78}$$

The operator fusion algebra is given by

$$\begin{aligned}
\eta^2 &= 1, & T_\eta\, \eta &= \eta\, T_\eta, & T_\eta^L &= \eta, \\
\mathsf{D}_\eta\, \eta &= \eta\, \mathsf{D}_\eta = -\mathsf{D}_\eta, & \mathsf{D}_\eta T_\eta = T_\eta \mathsf{D}_\eta &= -\mathsf{D}_\eta^\dagger, & \mathsf{D}_\eta^2 &= -(1-\eta)T_\eta^{-1}.
\end{aligned} \tag{79}$$

The continuum counterpart of this operator algebra is Table 2. See Section 3.1 for more comparisons between the lattice and the continuum.

The relation $T_\eta^L = \eta$ is generally expected to hold [14, 52].[24] Using $\mathsf{D}_\eta = -Z_L \mathsf{D} Z_1$ and (74), we find $\eta \mathsf{D}_\eta = \mathsf{D}_\eta \eta = -\mathsf{D}_\eta$. Moreover, using equation (63) one can easily compute $\mathsf{D}_\eta^2$. Finally, the fusion relation $\mathsf{D}_\eta T_\eta = T_\eta \mathsf{D}_\eta = -\mathsf{D}_\eta^\dagger$ is derived in Appendix F.3.

Parity P and time-reversal K are as in the untwisted theory and act as in (75) and (76)

$$\begin{aligned}
\mathsf{K}T_\eta &= T_\eta \mathsf{K}, & \mathsf{K}\eta &= \eta \mathsf{K}, & \mathsf{K}\mathsf{D}_\eta &= \mathsf{D}_\eta \mathsf{K}, \\
\mathsf{P}T_\eta &= T_\eta^{-1}\mathsf{P}, & \mathsf{P}\eta &= \eta \mathsf{P}, & \mathsf{P}\mathsf{D}_\eta &= \mathsf{D}_\eta^\dagger \mathsf{P}.
\end{aligned} \tag{81}$$

(As in footnote 23, we suppress a subscript $\eta$ on $\eta$, K and P.)

### 2.4.3 With a duality defect

Finally, we consider the algebra involving the symmetry operators of the duality-twisted Hamiltonian $H_\mathcal{D}^{(L,1)}$. The operator algebra in the presence of a duality defect was derived in [14]. Below we reproduce the same algebra from our MPO presentation of the operators.

The symmetries are generated by[25]

$$\eta_\mathcal{D} = X_L \cdots X_2 (Z_1 X_1), \quad T_\mathcal{D}^{-1} = \mathrm{Tr}\Big(\mathbb{U}_{\mathcal{T}^-}^L \cdots \mathbb{U}_{\mathcal{T}^-}^2 (\mathbb{T}^{-1})_\mathcal{D}^1\Big), \quad \mathsf{D}_\mathcal{D} = \mathrm{Tr}\Big(\mathbb{U}_\mathcal{D}^L \cdots \mathbb{U}_\mathcal{D}^2 \mathbb{D}_\mathcal{D}^1\Big), \tag{85}$$

---

[23]Since $\eta_\eta = \eta = \prod_{j=1}^L X_j$, in the following we will sometime use $\eta$ instead of $\eta_\eta$ on the lattice to avoid cluttering.

[24]Given a topological defect $\mathcal{A}$, the lattice translation in the presence of $\mathcal{A}$, $T_\mathcal{A}$ is given by $T_\mathcal{A} = (U_\mathcal{A}^1)^{-1}T$, which commutes with the defect Hamiltonian $H_\mathcal{A}^{(L,1)}$. This is because $T$ brings the defect to link $(1,2)$ by conjugation and $(U_\mathcal{A}^1)^{-1}$ brings it back on the original link. Using $T^L = 1$, we find the general relation

$$T_\mathcal{A}^{-L} = (T^{-1}U_\mathcal{A}^1 T)(T^{-2}U_\mathcal{A}^1 T^2)\cdots(T^{-L}U_\mathcal{A}^1 T^L) = U_\mathcal{A}^L U_\mathcal{A}^{L-1} \cdots U_\mathcal{A}^1 = \mathsf{A}_\mathcal{A}, \tag{80}$$

where $\mathsf{A}_\mathcal{A}$ is the operator corresponding to the defect $\mathcal{A}$ in the presence of an $\mathcal{A}$ defect. The convention here leads to the relation between the operator/defect algebras $\mathcal{A} \otimes \mathcal{B} \leftrightarrow AB$ as in Figure 2. In contrast, the alternative convention $T_\mathcal{A}^L = \mathsf{A}_\mathcal{A}$, which was used in [52], leads to $\mathcal{A} \otimes \mathcal{B} \leftrightarrow BA$.

[25]The symmetry is larger when there are several D defects. For example, consider a system with two $\mathcal{D}$ defects, located at two different links, say, $(L,1)$ and $(J, J+1)$ (with $J \neq L$)

$$H_{\mathcal{D};\mathcal{D}}^{(L,1);(J,J+1)} = -\sum_{j \neq 1, J+1}\big(Z_{j-1}Z_j + X_j\big) - Z_L X_1 - Z_J X_{J+1}. \tag{82}$$

(The Hamiltonian (33) corresponds to $J = 1$.) Unlike the system with no defect or with a single defect, here we have two $\mathbb{Z}_2$ internal symmetries, generated by

$$\eta_{\mathcal{D};\mathcal{D}}^{1,J} = X_1 X_2 \cdots X_J Z_{J+1}, \quad \text{and} \quad \eta_{\mathcal{D};\mathcal{D}}^{J+1,L} = X_{J+1}X_{J+2}\cdots X_L Z_1. \tag{83}$$

(When we separate the two defects far away from each other, these two operators become the ones discussed in [111].) The two $\mathcal{D}$ defects can be fused to find a direct sum of two systems and then these two $\mathbb{Z}_2$ symmetries act in each of them. More generally, if there are $M > 1$ $\mathcal{D}$ defects, then we have $M$ $\mathbb{Z}_2$ symmetries as in (83). Similar reasoning applies to the symmetry operators of $\mathsf{D}_{\mathcal{D};\mathcal{D}}$ and $T_{\mathcal{D};\mathcal{D}}$.

This is a manifestation of a more general phenomenon. Whenever we have two defects $\mathcal{A}$ and $\mathcal{B}$ such that $\mathcal{A} \otimes \mathcal{B} = \mathcal{B} \oplus \cdots$, then in the system with $M$ $\mathcal{B}$ defects, we will have $M$ conserved symmetry operators (which

where the tensors $(\mathbb{T}^{-1})^1_\mathcal{D}$ and $\mathbb{D}^1_\mathcal{D}$ are given in equations (F.38) and (72). The relations between these symmetry operators with or without a duality defect are

$$\eta_\mathcal{D} = Z_1 \eta, \qquad T^{-1}_\mathcal{D} = T^{-1} U^1_\mathcal{D}, \qquad \mathsf{D}_\mathcal{D} = U^L_\mathcal{D} U^{L-1}_\mathcal{D} \cdots U^2_\mathcal{D} U^1_\mathcal{D}. \tag{86}$$

Note that the duality operator $\mathsf{D}_\mathcal{D}$ in the duality-twisted problem is a product of the unitary movement operators $U^j_\mathcal{D}$. It is obtained from moving the defect $\mathcal{D}$ around the entire spatial circle. Therefore, $\mathsf{D}_\mathcal{D}$ is unitary, i.e., $\mathsf{D}^{-1}_\mathcal{D} = \mathsf{D}^\dagger_\mathcal{D}$, and is, in particular, invertible. This is a general fact: the symmetry operator in the Hamiltonian twisted by the said symmetry is always invertible, even when the symmetry in the untwisted problem is non-invertible.

The operator fusion algebra of these operators is[26]

$$\begin{aligned}
\eta^2_\mathcal{D} &= -1, & T_\mathcal{D}\, \eta_\mathcal{D} &= \eta_\mathcal{D}\, T_\mathcal{D}, & T^L_\mathcal{D} &= \mathsf{D}^{-1}_\mathcal{D} = \mathsf{D}^\dagger_\mathcal{D}, \\
\mathsf{D}_\mathcal{D}\, \eta_\mathcal{D} &= \eta_\mathcal{D}\, \mathsf{D}_\mathcal{D}, & \mathsf{D}_\mathcal{D} T_\mathcal{D} &= T_\mathcal{D} \mathsf{D}_\mathcal{D}, & \mathsf{D}^2_\mathcal{D} &= \frac{1}{\sqrt{2}}(1 + \eta_\mathcal{D}) T^{-1}_\mathcal{D}.
\end{aligned} \tag{87}$$

The continuum counterpart of this operator algebra is in Table 2. See Section 3.1 for more comparisons between the lattice and the continuum.

The relation $T^L_\mathcal{D} = \mathsf{D}^{-1}_\mathcal{D}$ follows from the general relation in (80). All other fusion relations follow from the last one, which is derived in equation (F.43) of Appendix F.4.

The entire operator algebra in the presence of a $\mathcal{D}$ defect is generated by $T_\mathcal{D}$. Indeed, $\eta_\mathcal{D}, \mathsf{D}_\mathcal{D}$ can be expressed in terms of $T_\mathcal{D}$:

$$\mathsf{D}_\mathcal{D} = T^{-L}_\mathcal{D}, \qquad \eta_\mathcal{D} = T^{-2(2L-1)}_\mathcal{D}. \tag{88}$$

The translation operator $T_\mathcal{D}$ obeys a single operator relation [14]:

$$T^{2(2L-1)}_\mathcal{D} = \sqrt{2}\, T^{2L-1}_\mathcal{D} - 1. \tag{89}$$

The operator algebra in (87) follows (88) and (89).

The fact that we have

$$T^{4(2L-1)}_\mathcal{D} = -1, \tag{90}$$

means that the symmetry group is $\mathbb{Z}_{4(2L-1)}$. Interestingly, the relation (89) restricts the operator algebra beyond $\mathbb{Z}_{4(2L-1)}$. It means that the eigenvalues of $T^{2L-1}_\mathcal{D}$ are $e^{\pm \frac{i\pi}{4}}$, but not $e^{\pm \frac{3\pi i}{4}}$. Such a restriction does not lead to a quotient of the symmetry algebra. Indeed, the operators in the theory transform linearly and faithfully under $\mathbb{Z}_{4(2L-1)}$. Instead, the relation (89) means that the operator algebra does not have operators with all possible $\mathbb{Z}_{4(2L-1)}$ representations.

Finally, we turn to the parity and time-reversal symmetries. Even though the Hamiltonian (22) is not manifestly parity invariant, it is easy to check that the parity transformation

$$\mathsf{P}_\mathcal{D} = \frac{X_1 + Z_1}{\sqrt{2}} \mathsf{P} \tag{91}$$

---

locally are identical to the symmetry operator of A), each stretching between two adjacent $\mathcal{B}$ defects:

$$\tag{84}$$



[26]Here we choose a phase for the lattice operator $\eta_\mathcal{D}$ so that $\eta^2_\mathcal{D} = -1$. This convention agrees with the one for the continuum operator $\eta_\mathcal{N}$ in Appendix G.3.

(where P is the naive parity transformation around site number 1), commutes with it. As above, time-reversal K acts simply as complex conjugation. These operators satisfy

$$\mathsf{K}T_{\mathcal{D}} = T_{\mathcal{D}}\mathsf{K}, \qquad \mathsf{K}\eta_{\mathcal{D}} = \eta_{\mathcal{D}}\mathsf{K}, \qquad \mathsf{K}\mathsf{D}_{\mathcal{D}} = \mathsf{D}_{\mathcal{D}}\mathsf{K},$$
$$\mathsf{P}_{\mathcal{D}}T_{\eta} = T_{\eta}^{-1}\mathsf{P}_{\mathcal{D}}, \qquad \mathsf{P}_{\mathcal{D}}\eta_{\mathcal{D}} = -\eta_{\mathcal{D}}\mathsf{P}_{\mathcal{D}}, \qquad \mathsf{P}_{\mathcal{D}}\mathsf{D}_{\mathcal{D}} = \mathsf{D}_{\mathcal{D}}^{\dagger}\mathsf{P}_{\mathcal{D}}. \tag{92}$$

Again, K commutes with all the symmetry operators. The only unusual point here is that $\mathsf{P}_{\mathcal{D}}$ anticommutes with $\eta_{\mathcal{D}}$ (or equivalently, $\mathsf{P}_{\mathcal{D}}\eta_{\mathcal{D}} = \eta_{\mathcal{D}}^{-1}\mathsf{P}_{\mathcal{D}}$). This fact will be important in Section 2.5.

## 2.5 Projective phases in algebras involving parity/time-reversal

As we mentioned above, the notion of anomalies in non-invertible symmetries that mix with lattice translations is subtle, partly because it is not completely clear how to gauge them. Some (or all) anomalies of invertible symmetries are associated with projective phases in the Hilbert space or in the Hilbert space in the presence of defects. In this section, we are going to find some projective phases in the Hilbert space with a non-invertible defect. This is reminiscent of the consequence of an anomaly for ordinary symmetries.

Specifically, we now discuss the projective phases in the symmetry algebras involving parity/time-reversal in the presence of a non-invertible defect in the critical Ising lattice model (10), as well as its deformations preserving these symmetries (see Section 4.1).

We will consider the Hamiltonian twisted by the non-invertible defect $\mathcal{D}$, such as (22). In the $\mathcal{D}$-twisted problem, we focus on the internal, invertible $\mathbb{Z}_2$ symmetry and the parity symmetry. As discussed in Section 2.4.3, the invertible symmetries of the twisted Hamiltonian are modified to $\eta_{\mathcal{D}} = Z_1\eta$ and $\mathsf{P}_{\mathcal{D}} = \frac{X_1+Z_1}{\sqrt{2}}\mathsf{P}$. The crucial point is that these two symmetries anti-commute (92):

$$\eta_{\mathcal{D}}\mathsf{P}_{\mathcal{D}} = -\mathsf{P}_{\mathcal{D}}\eta_{\mathcal{D}}. \tag{93}$$

This is to be contrasted with the corresponding algebra in the untwisted problem (75) where they commute, i.e., $\eta\mathsf{P} = \mathsf{P}\eta$. Note that the sign in (93) cannot be removed by any operator redefinition. This projective sign is similar to the consequence of an anomaly. See Appendix G.4 for the corresponding discussion in the continuum.

There is a similar projective phase involving the non-invertible symmetry and time-reversal. The algebra of $\eta_{\mathcal{D}}$ and time-reversal K is (92)

$$\eta_{\mathcal{D}}^2 = -1, \qquad \mathsf{K}^2 = 1, \qquad \eta_{\mathcal{D}}\mathsf{K} = \mathsf{K}\eta_{\mathcal{D}}. \tag{94}$$

We can try to remove the minus sign in the first equation by redefining $\eta_{\mathcal{D}}' = i\eta_{\mathcal{D}}$, but then we generate a sign in the third equation:

$$(\eta_{\mathcal{D}}')^2 = 1, \qquad \mathsf{K}^2 = 1, \qquad \eta_{\mathcal{D}}'\mathsf{K} = -\mathsf{K}\eta_{\mathcal{D}}'. \tag{95}$$

A related projective phase can be seen by analyzing the entire symmetry generated by $T_{\mathcal{D}}$ and parity in the $\mathcal{D}$-twisted problem. We can try to remove the $-1$ in the symmetry group relation (90) (and relatedly, in $\eta_{\mathcal{D}}^2 = -1$) by redefining $T_{\mathcal{D}}$ by a phase, $T_{\mathcal{D}}' = e^{\frac{i\pi n}{4}}T_{\mathcal{D}}$ (and therefore, $\eta_{\mathcal{D}}' = (T_{\mathcal{D}}')^{-2(2L-1)} = e^{-\frac{i\pi n(2L-1)}{2}}\eta_{\mathcal{D}}$) with odd $n$. However, such a redefinition leads to a phase in the relation between $T_{\mathcal{D}}$ and the parity operator (91). Explicitly, $T_{\mathcal{D}}$ and $\mathsf{P}_{\mathcal{D}}$ satisfy $T_{\mathcal{D}}\mathsf{P}_{\mathcal{D}} = T_{\mathcal{D}}^{-1}\mathsf{P}_{\mathcal{D}}$ and $\mathsf{P}_{\mathcal{D}}^2 = 1$. And after the redefinition, we have $T_{\mathcal{D}}'\mathsf{P}_{\mathcal{D}} = e^{\frac{i\pi n}{2}}(T_{\mathcal{D}}')^{-1}\mathsf{P}_{\mathcal{D}}$. As a result, the symmetry involving parity and translation is realized projectively.[27]

---

[27]There are also related projective phases in the $\eta$-twisted problem. We can redefine $\mathsf{D}_{\eta}' = i\mathsf{D}_{\eta}$ and $\eta' = -\eta$

# 3 The lattice symmetry vs. the continuum symmetry

We stressed in Section 1.2 that the lattice symmetry is not described by a fusion category, while the continuum symmetry is. Therefore, it is natural to ask how they are related and how much of the structure of the fusion category is captured by the lattice symmetry with its tensor product Hilbert space.

## 3.1 Non-invertible emanant symmetries

Let us focus on the special case of the critical Ising Hamiltonian and compare the operator algebras on the lattice (74), (79), (87) with those in the continuum Ising CFT in (G.1), (G.7), (G.11). We have summarized these operator algebras in Table 2.

**Brief review of the continuum symmetry**

In the continuum Ising CFT, the non-invertible Kramers-Wannier duality symmetry, together with the invertible $\mathbb{Z}_2$ symmetry, are described by a special case of the TY fusion category [43]. More generally, a TY fusion category, denoted as $TY(G, \chi, \epsilon)$ depends on a choice of a finite Abelian group $G$, a symmetric non-degenerate bicharacter $\chi$, a choice of the sign $\epsilon = \pm 1$ known as the Frobenius-Schur (FS) indicator.[28]

  When $G = \mathbb{Z}_2$, the TY fusion category has three simple objects: the identity line $1$, the invertible $\mathbb{Z}_2$ line $\eta$, and the non-invertible duality line $\mathcal{N}$. There is a unique symmetric non-degenerate bicharacter $\chi(\eta, \eta) = -1$, $\chi(1,1) = \chi(1,\eta) = \chi(\eta,1) = 1$. There are two TY fusion categories based on $G = \mathbb{Z}_2$ with different FS indicators. We denote them as $TY(\mathbb{Z}_2, \epsilon = \pm)$ suppressing the dependence on the bicharacter $\chi$ since the choice is unique. Both $TY(\mathbb{Z}_2, \epsilon = \pm)$ share the same fusion algebra. The Ising and the tricritical Ising CFTs realize the $\epsilon = +1$ case, while the $SU(2)_2$ WZW model realizes the $\epsilon = -1$ case (see for example [115]). We review the $TY(\mathbb{Z}_2, \epsilon)$ fusion categories in Appendix G, and refer the readers to [12, 31, 32, 62–64, 111, 116] for more discussions in the CFT context.

**No defect**

We start with the problem without a defect, which was already discussed in [13]. At finite $L$, we can identify unambiguously some of the low-lying states on the lattice with the states in the Ising CFT. These states are in Virasoro representations labeled by $(h, \bar{h}) = (0,0), (1/2, 1/2), (1/16, 1/16)$. (See Table 3.) On these states, we can express the lattice operators $T, D$ in terms of the CFT operators:

$$T = e^{\frac{2\pi i P}{L}}, \qquad D = \mathcal{N} e^{-\frac{2\pi i P}{2L}}, \tag{98}$$

in (79) to remove various minus signs in the operator algebra to find

$$
\begin{aligned}
(\eta')^2 &= 1, & T_\eta \, \eta' &= \eta' \, T_\eta, & T_\eta^L &= -\eta', \\
D'_\eta \, \eta' &= \eta' D'_\eta = +D'_\eta, & D'_\eta T_\eta &= T_\eta D'_\eta = +(D'_\eta)^\dagger, & (D'_\eta)^2 &= (1+\eta')T_\eta^{-1}.
\end{aligned}
\tag{96}
$$

This is very similar to the algebra without defects (73), with the only difference being $T_\eta^L = -\eta'$ (which does not matter when we act on local operator). However, now the action of parity and time-reversal (81) become

$$
\begin{aligned}
KD'_\eta &= -D'_\eta K, \\
PD'_\eta &= -(D'_\eta)^\dagger P.
\end{aligned}
\tag{97}
$$

[28]Given a simple object $\mathcal{L}$ in a fusion category, its dual object $\mathcal{L}^*$ is another simple object such that $\mathcal{L}\mathcal{L}^* = \mathcal{L}^*\mathcal{L} = 1 + \cdots$ contains the identity. For a self-dual object, i.e., $\mathcal{L} = \mathcal{L}^*$, the FS indicator $\epsilon$ was first defined in the context of Modular Tensor Categories (MTC) in [34, 112–114]. It can also be defined via a certain topological move such as in Appendix E of [114]. This is analogous to a representation being real or pseudo-real.

Table 2: The lattice and continuum operator algebras with various defects. The lattice operator algebras (74), (79), (87) hold for any Hamiltonian with the non-invertible symmetry, including the critical Ising Hamiltonian as a special case. The continuum operator algebras (G.1), (G.7), (G.11) hold in any CFT with the fusion category symmetry $\mathrm{TY}(\mathbb{Z}_2, \epsilon)$, which is reviewed in Appendix G. The special case of the Ising CFT realizes the $\epsilon = +1$ case. As we discussed around equation (96), we can redefine the phases of $\mathrm{D}_\eta$ and $\eta_\eta$ and change their algebra. Similar redefinitions can be done in the corresponding continuum elements $\mathcal{N}_\eta$ and $\eta_\eta$.

|  | lattice | continuum |
|---|---|---|
| no defect | $\eta^2 = 1,\ \eta\mathrm{D} = \mathrm{D}\eta = \mathrm{D}$ <br> $\mathrm{D}^2 = (1+\eta)T^{-1}$ <br> $T\eta = \eta T,\ T\mathrm{D} = \mathrm{D}T = \mathrm{D}^\dagger$ <br> $T^L = 1$ | $\eta^2 = 1,\ \eta\mathcal{N} = \mathcal{N}\eta = \mathcal{N}$ <br> $\mathcal{N}^2 = 1 + \eta$ <br> $e^{2\pi i P} = 1,\ \mathcal{N} = \mathcal{N}^\dagger$ <br> $e^{2\pi i P} = 1$ |
| $\mathbb{Z}_2$ defect | $\eta_\eta^2 = 1,\ \eta_\eta\,\mathrm{D}_\eta = \mathrm{D}_\eta\,\eta_\eta = -\mathrm{D}_\eta$ <br> $\mathrm{D}_\eta^2 = -(1-\eta_\eta)T_\eta^{-1}$ <br> $T_\eta\,\eta_\eta = \eta_\eta\,T_\eta,\ T_\eta\,\mathrm{D}_\eta = \mathrm{D}_\eta\,T_\eta = -\mathrm{D}_\eta^\dagger$ <br> $T_\eta^L = \eta_\eta$ | $\eta_\eta^2 = 1,\ \eta_\eta\,\mathcal{N}_\eta = \mathcal{N}_\eta\,\eta_\eta = -\mathcal{N}_\eta$ <br> $\mathcal{N}_\eta^2 = -\epsilon(1-\eta_\eta)$ <br> $\mathcal{N}_\eta = -\mathcal{N}_\eta^\dagger$ <br> $e^{2\pi i P} = \eta_\eta$ |
| duality defect | $\eta_{\mathcal{D}}^2 = -1$ <br> $\mathrm{D}_{\mathcal{D}}^2 = \frac{1}{\sqrt{2}}(1+\eta_{\mathcal{D}})T_{\mathcal{D}}^{-1}$ <br> $\mathrm{D}_{\mathcal{D}} = T_{\mathcal{D}}^{-L},\ \eta_{\mathcal{D}} = T_{\mathcal{D}}^{-2(2L-1)}$ <br> $T_{\mathcal{D}}^{2(2L-1)} = \sqrt{2}\,T_{\mathcal{D}}^{2L-1} - 1$ | $\eta_{\mathcal{N}}^2 = -1$ <br> $\mathcal{N}_{\mathcal{N}}^2 = \frac{\epsilon}{\sqrt{2}}(1+\eta_{\mathcal{N}})$ <br> $\mathcal{N}_{\mathcal{N}} = e^{-2\pi i P},\ \eta_{\mathcal{N}} = \mathcal{N}_{\mathcal{N}}^4$ <br> $\mathcal{N}_{\mathcal{N}}^4 = \sqrt{2}\mathcal{N}_{\mathcal{N}}^2 - 1$ |

where $P$ is the momentum operator in the continuum. (In CFT, its eigenvalues $h - \bar{h}$ are known as the conformal spins.) Importantly, the relations between the lattice and the continuum quantities (98) are exact on the low-lying states even for finite $L$ [52] because of the operator algebra $T^L = 1$ and $\mathrm{D}^{2L} = 2^{L-1}(1+\eta)$ [13]. In the thermodynamic limit, $T \to 1$ and $\mathrm{D} \to \mathcal{N}$, and the lattice algebra (74) reduces to the continuum fusion rule in Table 2. The non-invertible Kramers-Wannier symmetry $\mathcal{N}$ of the continuum Ising CFT is not emergent; rather, it emanates from the non-invertible lattice translation $\mathrm{D}$ of the transverse-field Ising lattice model. In particular, it is not violated by any irrelevant operator that preserves the exact lattice symmetry $\mathrm{D}$. In this sense, the continuum $\mathcal{N}$ is a non-invertible emanant symmetry [13, 52].

**With a $\mathbb{Z}_2$ defect**

As in the problem with no defect, on the low-lying states, the lattice operators $T_\eta, \mathrm{D}_\eta$ can be expressed in terms of the CFT operators as:

$$T_\eta = e^{\frac{2\pi i P}{L}}, \quad \mathrm{D}_\eta = \mathcal{N}_\eta\, e^{-\frac{2\pi i P}{2L}}, \tag{99}$$

where $\mathcal{N}_\eta$ is the Kramers-Wannier topological operator in the $\mathbb{Z}_2$-twisted Ising CFT (see Appendix G.2). As in [13, 52], these relations are exact on the low-lying states even for finite $L$

Table 3: The quantum numbers of the primary states of the untwisted Ising CFT.

|  | $(0,0)$ | $(\frac{1}{2}, \frac{1}{2})$ | $(\frac{1}{16}, \frac{1}{16})$ |
|---|---|---|---|
| $\eta$ | 1 | 1 | $-1$ |
| $\mathcal{N}$ | $\sqrt{2}$ | $-\sqrt{2}$ | 0 |

Table 4: The quantum numbers of the primary states in the $\eta$-twisted Ising CFT.

|  | $(\frac{1}{16}, \frac{1}{16})$ | $(0, \frac{1}{2})$ | $(\frac{1}{2}, 0)$ |
|---|---|---|---|
| $\eta_\eta$ | 1 | $-1$ | $-1$ |
| $\mathcal{N}_\eta$ | 0 | $i\sqrt{2}$ | $-i\sqrt{2}$ |

because $T_\eta^L = \eta_\eta$ and $D_\eta^{2L} = (-2)^{L-1}(\eta_\eta - 1)$. In particular, the former relation reproduces the spin selection rule in [12, 14, 117, 118]:

$$h - \bar{h} \in \begin{cases} \mathbb{Z}, & \text{if } \eta = +1, \\ \mathbb{Z} + \frac{1}{2}, & \text{if } \eta = -1. \end{cases} \tag{100}$$

This is indeed consistent with the conformal weights of the Virasoro primaries $(h, \bar{h}) = (1/16, 1/16), (0, 1/2), (1/2, 0)$ of the $\mathbb{Z}_2$-twisted Hilbert space for the Ising CFT. The $(h, \bar{h}) = (1/16, 1/16)$ state is $\mathbb{Z}_2$-even and corresponds to the disorder operator, while the other two states are $\mathbb{Z}_2$-odd and correspond to the right- and left-moving Majorana fermions.

In the thermodynamic limit, $T_\eta \to 1$, $D_\eta \to \mathcal{N}_\eta$, and the lattice algebra (79) reduces to the continuum operator algebra of Table 2 with $\epsilon = +1$.[29] The quantum numbers of the primary states under the symmetry operators in the $\eta$-twisted Ising CFT [119] appear in Table 4.

**With a duality defect**

The symmetry operators $T_{\mathcal{D}}$ and $D_{\mathcal{D}}$ act on the low-lying states as

$$T_{\mathcal{D}} = e^{2\pi i P \frac{2}{2L-1}}, \qquad D_{\mathcal{D}} = \mathcal{N}_{\mathcal{N}} e^{-\frac{2\pi i P}{2L-1}}, \tag{101}$$

where $\mathcal{N}_{\mathcal{N}}$ is the duality operator in the duality-twisted Hilbert space of the Ising CFT (see Appendix G.3). Note that $\mathcal{N}_{\mathcal{N}}$ is invertible. Again, these relations are exact on the low-lying states for finite $L$ because of the operator algebra $T_{\mathcal{D}}^{4(2L-1)} = -1$ and $D_{\mathcal{D}} = T_{\mathcal{D}}^{-L}$. The operator relation (89) implies that the eigenvalues of $T_{\mathcal{D}}^{2L-1}$ are equal to $e^{\pm 2\pi i/8}$, which together with (101) implies the spin selection rule on the low-lying states [12, 14, 115]:

$$h - \bar{h} \in \pm \frac{1}{16} + \frac{\mathbb{Z}}{2}. \tag{102}$$

This is indeed consistent with the conformal weights of the Virasoro primaries $(h, \bar{h}) = (1/16, 0), (1/16, 1/2), (0, 1/16), (1/2, 1/16)$ of the duality-twisted Hilbert space for the Ising CFT.

The effective number of sites in (101) is $L - \frac{1}{2}$ [14]. It is related to the fact that the duality-twisted Hamiltonian on $L$ Ising sites can be obtained from a Jordan-Wigner-like transformation of $2L - 1$ Majorana fermions [13].

In the thermodynamic limit for the critical Ising Hamiltonian, $T_{\mathcal{D}} \to 1$, and the lattice algebra (87) reduces to the continuum operator algebra in Table 2 with $\epsilon = +1$. The quantum numbers of the primary states under the symmetry operators in the $\mathcal{N}$-twisted Ising CFT [119] appear in Table 5. Note that parity $\mathcal{P}_{\mathcal{N}}$, which acts as $\mathcal{P}_{\mathcal{N}} |h, \bar{h}\rangle = |\bar{h}, h\rangle$, obeys the projective algebra

$$\mathcal{P}_{\mathcal{N}} \eta_{\mathcal{N}} = -\eta_{\mathcal{N}} \mathcal{P}_{\mathcal{N}}. \tag{103}$$

---

[29]Here we identify the lattice operator $\eta$ of the $\mathbb{Z}_2$-twisted problem with the continuum operator $\eta_\eta$. On the lattice, the $\mathbb{Z}_2$ operator is $\eta = \prod_{j=1}^{L} X_j$ for both the untwisted and the $\mathbb{Z}_2$-twisted Hamiltonians, so we use the same symbol $\eta$ for both of them. See footnote 23. In the continuum, the untwisted and the $\mathbb{Z}_2$-twisted Hilbert spaces are different and we need to distinguish the $\mathbb{Z}_2$ symmetry operators, denoted as $\eta$ and $\eta_\eta$, on these two Hilbert spaces.

Table 5: The quantum numbers of the primary states in the $\mathcal{N}$-twisted Ising CFT.

| | $(\frac{1}{16},0)$ | $(\frac{1}{16},\frac{1}{2})$ | $(0,\frac{1}{16})$ | $(\frac{1}{2},\frac{1}{16})$ |
|---|---|---|---|---|
| $\eta_{\mathcal{N}}$ | $-i$ | $-i$ | $i$ | $i$ |
| $\mathcal{N}_{\mathcal{N}}$ | $e^{-\frac{2\pi i}{16}}$ | $-e^{-\frac{2\pi i}{16}}$ | $e^{\frac{2\pi i}{16}}$ | $-e^{\frac{2\pi i}{16}}$ |

This matches with the algebra discussed in Section 2.5. See Appendix G.4 for a more general derivation.

We emphasize that most of our discussion in this subsubsection is special to the critical Ising Hamiltonian. In more general Hamiltonians flowing to a CFT, the relation between the lattice and CFT operators might be different. In particular, later in Section 3.2 we will discuss the sign $\epsilon$ in more details.

**Different emanant symmetries**

For more general Hamiltonians, it is possible that in the continuum, the lattice translation symmetry leads to an emanant internal finite symmetry of order $n$. For instance, $T$ can be spontaneously broken in a gapped phase, and acts as an internal symmetry on the nearly degenerate ground states in finite volume. In this case we have $T = g e^{\frac{2\pi i P}{L}}$ with $g^n = 1$. In the $L \to \infty$ limit, for $L$ a multiple of $n$, the lattice algebra (74) then becomes[30]

$$\begin{aligned}
\eta^2 &= 1, & g\eta &= \eta g, & g^n &= 1, \\
\mathcal{N}\eta &= \eta\mathcal{N} = \mathcal{N}, & g\mathcal{N} &= \mathcal{N}g = \mathcal{N}^\dagger, & \mathcal{N}^2 &= (1+\eta)g^{-1}.
\end{aligned} \tag{104}$$

In addition, we have $e^{2\pi i P} = 1$. In this case, the symmetry generated by $g$ is a $\mathbb{Z}_n$ emanant symmetry [52]. (The thermodynamic limit with $L$ not a multiple of $n$ corresponds to the problem with a $g$-symmetry twist.)

We see that the single lattice operator algebra (74) can lead to infinitely many fusion categories in the continuum depending on the choice of the Hamiltonian. When $n = 1$, which is the case for the critical Ising Hamiltonian (10), this reduces to the fusion algebra of $\mathrm{TY}(\mathbb{Z}_2, \epsilon)$.

**Comparison with the anyonic chain and other works**

It is interesting to compare our system with the anyonic chain [120–125]. The main difference between them stems from the fact that unlike our system, the Hilbert space of the anyonic chain is generally not a tensor product of local Hilbert spaces.

Unlike our case, where the lattice symmetry D mixes with lattice translation, in the anyonic chain, the lattice fusion category symmetry operators (also known as the "topological symmetries") are internal and do not mix with the lattice translation $T$. In the special case when the fusion category is $\mathrm{TY}(\mathbb{Z}_2, +)$, the anyonic chain is also not a tensor product Hilbert space, but is the *direct sum* of the Hilbert space on the sites and that on the links, $\mathcal{H}_{\text{site}} \oplus \mathcal{H}_{\text{link}}$.[31]

Comparing these two lattice constructions of the Kramers-Wannier duality symmetry, there appears to be a tension between the tensor product Hilbert space and an internal non-invertible symmetry that does not mix with the lattice translation. In the transverse-field Ising model, the

---

[30]Similar to footnote 22, this algebra can be realized in a subcategory of the fusion category $\mathrm{TY}(\mathbb{Z}_2, \epsilon) \boxtimes \mathrm{Vec}_{\mathbb{Z}_{2n}}$. (Let $\mathrm{TY}(\mathbb{Z}_2, \epsilon) \boxtimes \mathrm{Vec}_{\mathbb{Z}_{2n}}$ be generated by $\mathcal{N}_0$, $\eta$, and $g_0$ and write $\mathcal{N} = \mathcal{N}_0 g_0^{-1}$ and $g = g_0^2$.) Unlike footnote 22, which discusses a lattice symmetry, this comment is about the continuum symmetry. In particular, $n$ is the order of the emanant $\mathbb{Z}_n$ internal symmetry of the continuum theory, a fixed positive integer that does not depend on the lattice size $L$.

[31]Similar to the anyonic chain, in the model of [108], the Kramers-Wannier duality is viewed as an operator acting on the direct sum of the original Hilbert space with another copy of it corresponding to a $\mathbb{Z}_2$ defect.

Hilbert space is a tensor product of local Hilbert spaces but the non-invertible symmetry is not internal. In the anyonic chain, the Hilbert space is not a tensor product, but the non-invertible symmetry is internal.[32]

## 3.2 No Frobenius-Schur indicator on the lattice

In this subsection we will point out that for the lattice symmetry, there is no FS indicator and it arises only in the continuum limit. In contrast, in Section 3.3, we will see that the bicharacter can be defined on the lattice, where it is captured by a certain F-move.

On the lattice, the Kramers-Wannier symmetry element D is not self-dual. (See footnote 28.) We can see it either from the operator or the defect perspectives. As an operator, D is not a self-adjoint operator, as can be seen using

$$\text{lattice operator:} \quad \mathsf{D} \neq \mathsf{D}^{\dagger} = \mathsf{D}T, \tag{105}$$

where the lattice translation operator $T$ is nontrivial. As a defect, $\mathcal{D}$ is also different from its dual:

$$\text{lattice defect:} \quad \mathcal{D} \neq \mathcal{D}^{*} = \mathcal{D} \otimes \mathcal{T}^{+}, \tag{106}$$

where $\mathcal{T}^{+}$ is the lattice translation defect. (See Appendix E.) In particular, $\mathcal{D}^{*}$ involves adding a qubit to the Hilbert space, while $\mathcal{D}$ does not.

Since the lattice non-invertible symmetry is not self-dual, one cannot define a FS indicator. This is to be contrasted with the non-invertible symmetry in the continuum, which is self-dual, i.e. $\mathcal{N} = \mathcal{N}^{*}$.[33]

We conclude that the FS indicator of the continuum symmetry arises only in the continuum limit.[34] Indeed, in Appendix H, we will give an example of a continuous family of lattice Hamiltonians, all with the same lattice non-invertible symmetry. Depending on the parameters in the Hamiltonian, there are several low-energy continuum theories either with symmetry $\text{TY}(\mathbb{Z}_2, +)$ or $\text{TY}(\mathbb{Z}_2, -)$. This demonstrates our conclusion that the FS indicator of the continuum symmetry becomes meaningful only in the limit.

## 3.3 Bicharacter and F-symbols

We will now see that the lattice counterpart of the bicharacter is meaningful and is encoded in the lattice F-symbols.

Here we study the associativity of the movement and fusion operators of the defects. We start with a Hamiltonian with three defect insertions $\mathcal{L}_1, \mathcal{L}_2, \mathcal{L}_3$ and compare two sequences of fusion operations. In the first sequence, we first fuse $\mathcal{L}_1 \otimes \mathcal{L}_2$, and then fuse the composite

---

[32]We stress that this observation is valid only for certain classes of lattice non-invertible symmetries such as the one in the Ising model. It is possible to realize fusion categories with a fiber functor (which are sometimes referred to as anomaly-free fusion categories [32,61,116]) on a tensor product Hilbert space without mixing with the lattice translation [126,127].

[33]We can also see the ambiguity of the FS indicator from the operator algebra. In the continuum, the FS indicator enters the operator algebra as in (G.11), $\mathcal{N}_{\mathcal{N}}^2 = \frac{\epsilon}{\sqrt{2}}(1 + \eta_{\mathcal{N}})$. The lattice counterpart is given in (87):

$$\mathsf{D}_{\mathcal{D}}^2 = \frac{1}{\sqrt{2}}(1 + \eta_{\mathcal{D}})T_{\mathcal{D}}^{-1}. \tag{107}$$

(We identify $\eta_{\mathcal{N}}$ in the continuum with its lattice counterpart $\eta_{\mathcal{D}}$, both squaring to $-1$.) However, the overall sign (which would have been the FS indicator) on the righthand side of (107) can be changed by redefining $U_{\mathcal{D}}^j \to -U_{\mathcal{D}}^j$, which changes $T_{\mathcal{D}} \to -T_{\mathcal{D}}$ and $\mathsf{D}_{\mathcal{D}} \to (-1)^L \mathsf{D}_{\mathcal{D}}$. Clearly, the fact that this redefinition changes the sign is related to the fact that $\mathsf{D}_{\mathcal{D}}$ is not selfdual.

[34]Note that other lattice systems, e.g., the anyonic chain or other non-invertible symmetries on a tensor product Hilbert space, can have non-invertible symmetries that do not involve translation and therefore they can be self-dual. Such symmetries can have meaningful FS indicators.

with $\mathcal{L}_3$. In the second sequence, we first fuse $\mathcal{L}_2 \otimes \mathcal{L}_3$, and then fuse $\mathcal{L}_1$ with the composite. By comparing the two sequences of unitary operators that implement these fusion operations, we can define a lattice counterpart of the F-symbols.

Unlike a fusion category in the continuum, the lattice symmetry mixes with translation. Therefore the F-symbols can be more subtle than in the continuum. In particular, the number of lattice defects can grow with $L$. In this paper we will focus on a particular F-move that captures the bicharacter, but does not involve the lattice translation defect. A closely related lattice F-move has been discussed in [111]. We leave a comprehensive study of the F-symbols on the lattice for the future.

Specifically, consider the defect Hamiltonian with an $\eta$ defect at the link $(L, 1)$, a $\mathcal{D}$ defect at link $(1, 2)$, and another $\eta$ defect at link $(2, 3)$.

$$H_{\eta;\mathcal{D};\eta}^{(L,1);(1,2);(2,3)} = -(-Z_L Z_1 + X_1) - Z_1 X_2 - (-Z_2 Z_3 + X_3) - \sum_{j=4}^{L}(Z_{j-1}Z_j + X_j). \tag{108}$$

We will compare two sequences of fusion operations to bring this configuration to the defect Hamiltonian of $\mathcal{D}$ at link $(1, 2)$. To this end, it will be more convenient to consider a slightly different fusion operator $h_{\mathcal{D}\otimes\eta}^1$ defined by $h_{\mathcal{D}\otimes\eta}^1 H_{\mathcal{D};\eta}^{(L,1);(1,2)} (h_{\mathcal{D}\otimes\eta}^1)^{-1} = H_{\mathcal{D}}^{(L,1)}$. It differs from $\lambda_{\mathcal{D}\otimes\eta}^1$ in (31) by a movement operator $U_{\mathcal{D}}^1$.[35] Diagrammatically, it is

$$h_{\mathcal{D}\otimes\eta}^1 = X_1 = \qquad \tag{109}$$

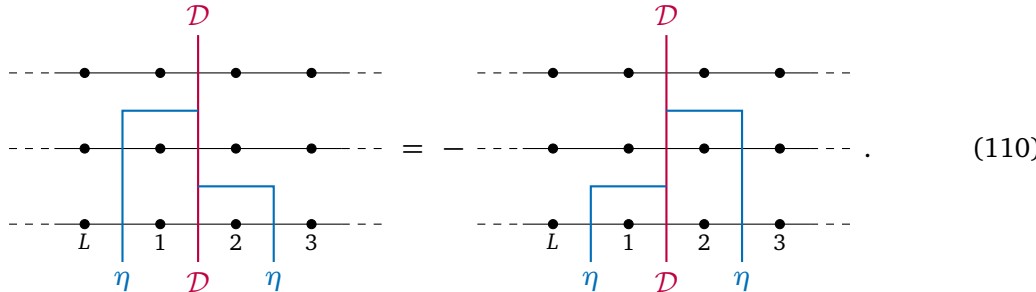

The two sequences that we study are

$$\qquad = - \qquad . \tag{110}$$

The first sequence, shown on the left of (110), is implemented by the unitary operator $\lambda_{\eta\otimes\mathcal{D}}^1 h_{\mathcal{D}\otimes\eta}^2 = (X_1 Z_2) X_2$, while the second sequence, shown on the right, is implemented by the unitary operator $h_{\mathcal{D}\otimes\eta}^2 \lambda_{\eta\otimes\mathcal{D}}^1 = X_2(X_1 Z_2)$.

While both unitary operators map the defect Hamiltonian (108) to $H_{\mathcal{D}}^{(1,2)}$, they differ by a minus sign:

$$\lambda_{\eta\otimes\mathcal{D}}^1 h_{\mathcal{D}\otimes\eta}^2 = -h_{\mathcal{D}\otimes\eta}^2 \lambda_{\eta\otimes\mathcal{D}}^1. \tag{111}$$

Note that this relative sign is independent of the phase redefinition of the fusion operators $\lambda_{\eta\otimes\mathcal{D}}^j$ and $h_{\mathcal{D}\otimes\eta}^j$. This minus sign corresponds to the following F-move in the $\text{TY}(\mathbb{Z}_2, +)$ fusion

---

[35]Note that the symbols "$\lambda$" and "$h$" are chosen to resemble the fusion configurations in (31) and (109), respectively.

category:

$$\text{(figure)} \qquad = \quad - \quad \text{(figure)} \quad , \tag{112}$$

where the blue and red lines stand for the $\mathbb{Z}_2$ line and the non-invertible line, respectively. (See Appendix G for the other F-symbols.) The above minus sign corresponds to $\chi(\eta, \eta) = -1$. We leave the other F-symbols for future investigations.

### 3.4 Lattice quantum dimension

We now define a lattice version of the quantum dimension of a defect, which is generally different from the quantum dimension in fusion categories in the continuum.

We consider a translationally-invariant Hamiltonian $H$ on a one-dimensional closed periodic chain of a tensor product Hilbert space $\mathcal{H} = \bigotimes_{j=1}^{L} \mathcal{H}_j$ with each $\mathcal{H}_j = \mathbb{C}^2$, a qubit. A lattice defect $\mathcal{A}$ is defined in terms of a defect Hamiltonian $H_{\mathcal{A}}$, which differs from $H$ only locally around a particular site. We assume that the defect is topological in the sense that there is a unitary movement operator that changes its location.

We further extend this discussion for the case where $H_{\mathcal{A}}$ involves a translation defect. In that case, we might have more or less qubits around the location of the defect, and the Hilbert space $\mathcal{H}_{\mathcal{A}}$ differs from the original one $\mathcal{H}$ (see Appendix E). This motivates us to define the lattice quantum dimension $\dim \mathcal{A}$ of a defect $\mathcal{A}$ as

$$\dim \mathcal{A} = \frac{\dim \mathcal{H}_{\mathcal{A}}}{\dim \mathcal{H}} . \tag{113}$$

Note that we study a finite lattice with fixed $L$, such that both the numerator and the denominator in (113) are positive integers. Therefore, unlike the quantum dimension in a fusion category, the lattice quantum dimension is always a positive rational number.

Since the fusion operation $\otimes$ between two defects is implemented by the unitary fusion and movement operators, which do not change the dimension of the Hilbert space, we have $\dim(\mathcal{A} \otimes \mathcal{B}) = (\dim \mathcal{A})(\dim \mathcal{B})$. Similarly, since the Hilbert space for the direct sum defect $\mathcal{A} \oplus \mathcal{B}$ is defined as taking the direct sum of the corresponding defect Hilbert spaces, i.e., $\mathcal{H}_{\mathcal{A} \oplus \mathcal{B}} = \mathcal{H}_{\mathcal{A}} \oplus \mathcal{H}_{\mathcal{B}}$, we have $\dim(\mathcal{A} \oplus \mathcal{B}) = \dim \mathcal{A} + \dim \mathcal{B}$. It follows that the lattice quantum dimension gives a positive rational 1-dimensional representation for the lattice defect fusion rule.

We can immediately read off the lattice quantum dimensions of the defects $\mathcal{T}^n, \eta \mathcal{T}^n, \mathcal{D} \mathcal{T}^n$ of the Ising model. Obviously, the trivial defect has unit quantum dimension, $\dim 1 = 1$. The $\mathbb{Z}_2$ defect $\eta$ modifies one term in the Hamiltonian $H_{\eta}^{(L,1)}$ in (12) without changing the Hilbert space, thus $\dim \eta = 1$. The translation defects $\mathcal{T}^n$ adds/removes $|n|$ qubits, hence $\dim \mathcal{T}^n = 2^n$. Since the non-invertible duality defect $\mathcal{D}$ does not change the Hilbert space as in (22), we have $\dim \mathcal{D} = 1$. In contrast, the dual defect $\mathcal{D}^* = \mathcal{D} \otimes \mathcal{T}^+$ adds one more qubit to the Hilbert space (see (E.23)), hence $\dim \mathcal{D}^* = 2$. To summarize, we have

$$\text{lattice :} \quad \begin{aligned} \dim 1 &= 1, \quad \dim \eta = 1, \\ \dim \mathcal{T}^n &= 2^n, \\ \dim \mathcal{D} &= 1, \quad \dim \mathcal{D}^* = 2, \end{aligned} \tag{114}$$

while the rest can be obtained by multiplication. Note that these lattice quantum dimensions are compatible with the fusion rule of the defects $\mathcal{D} \otimes \mathcal{D} = (1 \oplus \eta) \otimes \mathcal{T}^-$ and $\mathcal{D} \otimes \mathcal{D}^* = 1 \oplus \eta$.

Let us compare the lattice quantum dimensions with the continuum quantum dimensions. For the $\mathrm{TY}(\mathbb{Z}_2, \epsilon)$ fusion category in the continuum, the non-invertible defect $\mathcal{N}$ is self-dual and the quantum dimension is

$$\text{continuum}: \quad \dim \mathcal{N} = \dim \mathcal{N}^* = \sqrt{2}. \tag{115}$$

In contrast, on a tensor product lattice $\mathcal{D} \neq \mathcal{D}^*$ and they have different quantum dimensions.

One application of the lattice quantum dimension is that it provides a no-go argument for the realization of certain fusion rules of defects on a tensor product Hilbert space. For instance, the continuum fusion rule $\mathcal{N}^2 = 1 + \eta$ leads to an irrational quantum dimension for $\mathcal{N}$. On the other hand, the lattice quantum dimension is always a positive rational number. Therefore, the fusion rule $\mathcal{N}^2 = 1 + \eta$ cannot be realized for lattice defects on a tensor product Hilbert space.

The above no-go argument applies only to the *defects*. It does not rule out such an algebra for the *operators*. For example, the operator PD realizes this algebra, where P is parity. Indeed, using (76), we find

$$(\mathrm{PD})^2 = \mathrm{D}^\dagger \mathrm{D} = 1 + \eta. \tag{116}$$

However, the operator PD does not act locally on the local operators — it maps a local operator around site $j$ to a local operator around site $L - j$. Therefore, there is no defect (i.e., local modification of the Hamiltonian) associated with PD.

# 4 Deformations and an LSM-type constraint

## 4.1 D-preserving deformations

The lattice non-invertible symmetry is not special to the critical transverse-field Ising model (10). There are infinitely many deformations of (10) preserving the non-invertible operator D. More specifically, any $\mathbb{Z}_2$-invariant and translationally-invariant deformation that is also invariant under (21)

$$\begin{aligned} X_j &\rightsquigarrow Z_{j-1} Z_j, \\ Z_{j-1} Z_j &\rightsquigarrow X_{j-1}, \end{aligned} \tag{117}$$

preserves the non-invertible symmetry.

Another perspective of the non-invertible symmetry D comes from the related Majorana chain. Any D-preserving deformation of the Ising chain is mapped locally to a deformation of the Majorana chain that preserves the translation by one Majorana site. Hence, imposing this non-invertible symmetry locally is as natural as imposing an ordinary invertible symmetry. Globally, the bosonic spin model and the Majorana fermion model are different. In particular, they generally have a different number of ground states on a closed chain. See [13, 128] for recent discussions on the lattice bosonization and Appendix C for a review.

For instance, one D-preserving deformation of (10) is:

$$\lambda_1 \sum_{j=1}^{L} \left( X_j X_{j+1} + Z_j Z_{j+2} \right). \tag{118}$$

This deformation of the critical Ising model was briefly discussed in [101, 102], where the emphasis was on the phase diagram of the fermionic model. Locally, this deformation is mapped to $\sum_\ell \chi_\ell \chi_{\ell+1} \chi_{\ell+2} \chi_{\ell+3}$ of the Majorana fermion under the Jordan-Wigner transformation (C.6).

Another interesting D-preserving deformation is [103]

$$\lambda_2 \sum_{j=1}^{L} \left( X_j Z_{j+1} Z_{j+2} + Z_j Z_{j+1} X_{j+2} \right). \tag{119}$$

Locally, it is mapped under the Jordan-Wigner transformation (C.6) to $\sum_\ell \chi_{\ell-2}\chi_{\ell-1}\chi_{\ell+1}\chi_{\ell+2}$. (In Appendix B.2, we will present the non-invertible defects $\mathcal{D}$ for these deformed Hamiltonians.)

Although we will not discuss it in detail here, we can also consider deformations that preserve all our symmetries except parity and time-reversal. For example, we can have

$$
i \sum_j \left( X_j X_{j+1} Z_{j+1} Z_{j+2} + Z_j Z_{j+2} X_{j+2} \right) = \sum_j \left( X_j Y_{j+1} Z_{j+2} - Z_j Y_{j+2} \right), \tag{120}
$$

which translates in the fermionic theory to $\sum_\ell \chi_\ell \chi_{\ell+1} \chi_{\ell+2} \chi_{\ell+4}$.

See [104, 129, 130] for more examples of Hamiltonians with the non-invertible symmetry D.

We comment that the continuum Ising CFT does not have any relevant deformation that preserves the non-invertible symmetry $\mathcal{N}$. The $(h, \bar{h}) = (1/2, 1/2)$ and $(h, \bar{h}) = (1/16, 1/16)$ Virasoro primary operators transform under the non-invertible symmetry. The lowest dimension $\mathcal{N}$-preserving operator is the $T\bar{T}$ deformation [131]. Therefore, the above D-preserving lattice deformations are irrelevant around the Ising CFT fixed point and there is a finite gapless region in the space of D-preserving deformation corresponding to the Ising CFT, such as in Figure 1. As we increase these coupling constants, the deformations can become important and can change the phase [101–103], again, as in Figure 1.

## 4.2  LSM-type constraint

The existence of the non-invertible lattice symmetry has consequences on the phase diagram. We will argue that:

*Any system with a finite-range Hamiltonian preserving* D *must either be gapless or gapped with its symmetry being spontaneously broken. In the latter case, the number of superselection sectors must be a multiple of 3.*

Our argument follows closely the continuum discussion in [98–100]. As there, it is elementary and does not rely on intricacies of category theory or anomalies. We remind the readers that any D-preserving Hamiltonian is necessarily invariant under the translation symmetry $T$ and the on-site $\mathbb{Z}_2$ symmetry $\eta = \prod_{j=1}^{L} X_j$.

**Well-known comments about the low-energy theory**

In preparation for the discussion of the LSM-type constraint, we would like to make some comments about the effective low-energy theory in a gapped phase. When discussing this topic, we have in mind three distinct situations.

1. We consider the system with large but finite $L$. In a gapped phase, there are $N_{states}$ low-lying states $|I\rangle$. Without loss of generality, we set the energy of the lowest energy state $|I = 1\rangle$ to zero and then the other states $|I\rangle$ with $I = 2, \cdots, N_{states}$ have energy of or order $e^{-a_I L}$ with some positive constants $a_I$. The other states in the spectrum have energy of order one. (In a gapless phase, there are also states with energy of order $\frac{1}{L}$.) The low-energy theory focuses on the $N_{states}$ low-lying states and their effective dynamics is obtained by integrating out the higher energy states.

2. We keep $L$ large but finite and study the same $N_{states}$ states $|I\rangle$. However, now since $L$ is large, we neglect their exponentially small energies. As a result, we have $N_{states}$ zero energy states. These states are described by a 1+1d TQFT.[36] Since all the states

---

[36]In the Condensed Matter literature, it is common to define a TQFT as a theory where there are no local operators acting in the space of ground states. This guarantees that the TQFT is robust against perturbation by

are degenerate, the basis $|I\rangle$ is no longer preferred. Instead, there is another preferred basis of states $|i\rangle$ with $i = 1, \cdots, N_{states}$ in which all the local operators of the theory are diagonal. See Appendix I.3, for additional discussion of this topic and specifically for the Ising TQFT $TY(\mathbb{Z}_2, +)$.

3. In the infinite volume limit, the full Hilbert space of the problem is split into $N_{states}$ distinct superselection sectors, which are labeled by $i = 1, \cdots, N_{states}$. The ground state in each of them are the states $|i\rangle$ mentioned above.

Often, people use imprecise language and say that the infinite volume theory has $N_{states}$ ground states. More precisely, this statement applies to the second case above, or alternatively, it means that the infinite volume theory has $N_{states}$ superselection sectors. Below, we will sometime use this imprecise language.

**The argument**

We start by reviewing some basic facts about generic 1+1d gapped systems with a $\mathbb{Z}_2$ global symmetry generated by $\eta$. If in the infinite volume system (third situation above) the $\mathbb{Z}_2$ symmetry is spontaneously broken (ordered phase), then in finite volume (first situation above), the system has two low-lying states with $\eta = \pm 1$. The energy splitting between them is $e^{-aL}$ (with $a$ an order 1 positive number) and there is a finite gap above these two states. In the picture of the second situation above, it is better to consider another basis with the two states $|i\rangle$ that are exchanged by $\eta$. These two states lead to two distinct superselection sectors in the infinite-volume theory (third situation above). If on the other hand the symmetry is unbroken (disordered phase), then the finite-volume theory has a unique $\mathbb{Z}_2$-invariant ground state with an order 1 gap above it.[37] This state leads to a unique $\mathbb{Z}_2$ preserving superselection sector in the infinite-volume theory.

Next, we use the non-invertible symmetry D. We show in Appendix B that a finite-range Hamiltonian commutes with D if, and only if, it is invariant under gauging the on-site $\mathbb{Z}_2$ symmetry. It is well-known that gauging the $\mathbb{Z}_2$ symmetry exchanges the ordered and the disordered phases. Therefore, a single ordered phase cannot be compatible with the non-invertible symmetry D, nor is a single disordered phase. Instead, the minimal situation corresponds to two ordered states (with $\eta = \pm 1$, or equivalently, two states $|i\rangle$ that are exchanged by $\eta$) and a single disordered state.

Let us make some comments about this statement.

- Unlike the generic situation with a $\mathbb{Z}_2$ symmetry discussed above, here we also have another symmetry, D and therefore we have a more special situation.

- If we slightly break the D symmetry, but preserve the $\mathbb{Z}_2$ symmetry, the ordered and the disordered states are no longer degenerate. Then, it is clear that the D invariant theory corresponds to a first order transition between an ordered and a disordered phases. And the degeneracy that follows from D is the standard degeneracy of first order transitions. (See Figure 1.) In other words, the D symmetry forces the co-existence of order and disorder [103].

- In more special situations, we can have $2m$ ordered states (that are paired by $\eta$) and $m$ disordered states, such that the total number of low-lying states is a multiple of 3.

---

local operators. Following this definition, there are no TQFTs in 1+1d. Instead, in the mathematics and the quantum field theory literature, it is common not to impose this additional requirement and then it is possible to have TQFTs in quantum mechanics and in 1+1d. We will adopt this second definition.

[37]The classification of SPT phases in [132] further implies that such a $\mathbb{Z}_2$-preserving gapped phase in 1+1d is unique since $H^2(\mathbb{Z}_2, U(1))$ is trivial.

- To avoid confusion, this discussion of the low-lying degenerate states corresponds to the second situation above. The states are degenerate because we neglect the exponentially small splitting, but we do not have the separation of the infinite-volume theory into superselection sectors.

We conclude that the number of degenerate ground states in the large volume limit (the second situation above) should be a multiple of 3 . This completes the argument for the LSM-type constraint.[38]

**Further comments**

One crucial fact we used in this argument is that there is no $\mathbb{Z}_2$ SPT phase that is invariant under gauging a $\mathbb{Z}_2$ global symmetry. When we generalize this argument to non-invertible duality symmetries associated with gauging a more general finite Abelian group $G$, it is possible that there exists a $G$-SPT phase that is invariant under gauging $G$ [32, 98]. In this case, the above LSM obstruction disappears. Indeed, this is the case for $G = \mathbb{Z}_2 \times \mathbb{Z}_2$, and some of the associated $\mathbb{Z}_2 \times \mathbb{Z}_2$ TY fusion categories are compatible with a trivially gapped phase [133]. See [134] for the corresponding lattice examples.

Another fact we used is that the $\mathbb{Z}_2$ gauging exchanges order and disorder. Let us argue for this point by tracking states with various boundary conditions, and show algebraically how the non-invertible symmetry exchanges order and disorder. We study the untwisted system with Hamiltonian $H$ and compare it with the twisted system, which is the same system with an $\eta$ defect $H_\eta^{(L,1)}$. It is simple to compare these problems by remembering that the system with $H_\eta^{(L,1)}$ is the same as the system with $H$, but with twisted boundary conditions. Consider first the ordered states of the untwisted problem and use the basis $|i\rangle$ (that leads to superselection sectors in the infinite volume theory). They are characterized by the expectation value of some order parameter. Going to the twisted problem, this order parameter should be antiperiodic as we go around the space and therefore there must be a domain wall. Since the domain wall is associated with the higher energy states, the domain wall must have energy of order one. We conclude that ordered states of the untwisted problem do not lead to low-energy states in the twisted problem.

The situation is different for disordered states. Here, the twisted boundary conditions do not imply the existence of a domain wall and therefore these states do lead to low-lying states in the twisted theory. We conclude that every disordered low-lying state of the untwisted problem leads to a single low-lying disordered state in the twisted theory.

Next, we follow Section 2.3.1 and consider a direct sum of two copies of the system, one with $H$ and the other with $H_\eta^{(L,1)}$. We denote the Hamiltonian on this $2^{L+1}$-dimensional Hilbert space by $H_{1 \oplus \eta}^{(L,1)}$. We assign $\tilde{\eta} = +1$ to the states in the untwisted problem and $\tilde{\eta} = -1$ to the states in the twisted problem. This enlarged system has a $\mathbb{Z}_2 \times \mathbb{Z}_2$ symmetry, which is generated by $\eta = \prod_{j=1}^{L} X_j$ and $\tilde{\eta}$, where $\tilde{\eta}$ acts on the extra qubit that implements the direct sum.[39] This $\mathbb{Z}_2 \times \mathbb{Z}_2$ symmetry has been discussed in various places including [135].

The picture above about the ordered and the disordered states implies that in this larger Hilbert space, the ordered phase corresponds to two ground states of $H_{1 \oplus \eta}^{(L,1)}$ with eigenvalues $\eta = \pm 1$ and $\tilde{\eta} = 1$, while a disordered phase leads to two ground states of $H_{1 \oplus \eta}^{(L,1)}$ with eigen-

---

[38]We stress that our LSM-type constraint, as stated, applies to the bosonic lattice model of Ising spins with periodic boundary conditions. It does not apply to the Majorana chain with fixed boundary conditions, as bosonization changes the number of ground states globally. For instance, the authors of [102] found 2 or 4 states in gapped phases of the corresponding Majorana model of the deformation (118). See also the related discussion in Appendix I.

[39]In the particular case of the Ising Hamiltonian, it is given in (47), in which case $\tilde{\eta} = Z_{(L,1)} = Z_{L+1}$. But here we discussed a gapped system.

Table 6: The quantum numbers for the ground states of the Hamiltonian $H_{1\oplus\eta}^{(L,1)}$ on the $2^{L+1}$-dimensional Hilbert space for the direct sum of the untwisted ($\tilde\eta = +1$) and the $\mathbb{Z}_2$-twisted ($\tilde\eta = -1$) problems. Left: $\mathbb{Z}_2$-symmetry breaking phase (order). Right: $\mathbb{Z}_2$-preserving phase (disorder). These tables reflect the fact that the $\mathbb{Z}_2$-gauging (or equivalently, the operator D) exchanges order and disorder and $\eta \leftrightarrow \tilde\eta$.

| order | $\eta = +1$ | $\eta = -1$ |
|---|---|---|
| $\tilde\eta = +1$ | 1 | 1 |
| $\tilde\eta = -1$ | 0 | 0 |

| disorder | $\eta = +1$ | $\eta = -1$ |
|---|---|---|
| $\tilde\eta = +1$ | 1 | 0 |
| $\tilde\eta = -1$ | 1 | 0 |

values $\eta = 1$ and $\tilde\eta = \pm 1$. See Table 6. (In this discussion, we refer to these states as ground states because we have in mind the picture of the finite-volume problem where we neglect the exponentially small energy splitting.)

Now, we assume that there is also a noninvertible symmetry D and consider the operator $D_{1\oplus\eta}$. It commutes with the Hamiltonian $H_{1\oplus\eta}^{(L,1)}$ and acts as

$$D_{1\oplus\eta}\,\eta\,(D_{1\oplus\eta})^{-1} = \tilde\eta, \quad \text{and} \quad D_{1\oplus\eta}\,\tilde\eta\,(D_{1\oplus\eta})^{-1} = \eta, \tag{121}$$

where we used (51). (Note that as we said around (49), $D_{1\oplus\eta}$ is invertible. This is also manifest in the fermionic presentation (C.22)). This shows explicitly that D swaps the disorder ($\eta = 1, \tilde\eta = \pm 1$) and the order ($\eta = \pm 1, \tilde\eta = 1$) phases.

**Relation to prior works**

Reference [103] discusses a lattice application of this LSM-type constraint. They consider a one-parameter deformation (119) $\lambda_2 \sum_{j=1}^{L} \left( X_j Z_{j+1} Z_{j+2} + Z_j Z_{j+1} X_{j+2} \right)$ of the critical Ising Hamiltonian preserving the non-invertible symmetry D. Since this operator is irrelevant around the Ising CFT fixed point, there is an open neighborhood around (10) that flows to the Ising CFT. As we increase the coupling constant, there is a tricritical Ising CFT point, beyond which the phase is gapped with three nearly degenerate ground states, a consequence of the non-invertible symmetry. (See Figure 1.) Similar results were proven on the anyonic chain in [124].

In [136], it was proven rigorously that any Hamiltonian invariant under (117) must be either gapless or have more than one superselection sector in the thermodynamic limit. Our argument further implies that in the latter case the number of superselection sectors has to be a multiple of 3.

There is a special point $\lambda_2 = 1/2$ on the phase diagram in [103], where the three ground states are exactly degenerate even in finite volume. The three ground states $|I\rangle$ that diagonalize $\eta, D$ are

$$\frac{1}{\sqrt{2}}(|++...+\rangle \pm |\text{GHZ}\rangle), \qquad D = \pm\sqrt{2}, \quad \eta = +1,$$
$$\frac{1}{\sqrt{2}}(|00...0\rangle - |11...1\rangle), \qquad D = 0, \ \eta = -1, \tag{122}$$

where $|\text{GHZ}\rangle = \frac{1}{\sqrt{2}}|00...0\rangle + \frac{1}{\sqrt{2}}|11...1\rangle$. The inner products between the product states $|00...0\rangle$, $|11...1\rangle$, $|++...+\rangle$ are nonzero, but they vanish in the large $L$ limit. This is consistent with the fact that for infinite volume, these three states belong to three different superselection sectors.

The non-invertible symmetry D and the $\mathbb{Z}_2$ symmetry act on them as:

$$\eta|++...+\rangle = |++...+\rangle, \qquad \eta|00...0\rangle = |11...1\rangle, \qquad \eta|11...1\rangle = |00...0\rangle,$$
$$D|++...+\rangle = |00...0\rangle + |11...1\rangle, \qquad D|00...0\rangle = D|11...1\rangle = |++...+\rangle. \tag{123}$$

The invertible $\mathbb{Z}_2$ symmetry $\eta$ is spontaneously broken in the first two sectors and unbroken in the last. The non-invertible symmetry D does not leave any of the sector invariant, and we interpret that D is spontaneously broken. Finally the lattice translation $T$ is preserved in all three superselection sectors.[40]

This LSM-type constraint is the lattice counterpart of Section 7.2.3 of [12] in the continuum, where it was proved that any renormalization group flows preserving the TY($\mathbb{Z}_2$, +) fusion category symmetry must either flow to a CFT, or a 1+1d TQFT with at least 3 states. This constraint was interpreted as a generalized 't Hooft anomaly for the non-invertible global symmetry. As an example, one can start with the tricritical Ising CFT of $c = 7/10$ and turn on a relevant deformation $\phi_{1,3} = \varepsilon'$ with $(h, \bar{h}) = (3/5, 3/5)$. This subleading thermal deformation $\varepsilon'$ preserves the TY($\mathbb{Z}_2$, +) fusion category symmetry.[41] For one sign of the relevant deformation, it flows to the Ising CFT (which is the famous Zamolodchikov flow [137]), while for the other sign, it flows to a 1+1d TQFT with 3 ground states [138]. This is precisely the situation in Figure 1. The deformation of $\lambda$ from the critical point at $(\beta_c, \lambda_c)$ is the deformation by the operator $\varepsilon'$. See Appendix I for more discussions about this flow.

The LSM-type constraint also has a similar flavor as the more general statement proven in Section 7.1 of [12]: Any renormalization group flow preserving a non-invertible topological line whose quantum dimension is not a non-negative integer cannot be gapped with a non-degenerate ground state. This was interpreted as a generalized 't Hooft anomaly in the non-invertible symmetry. We review this proof in the continuum in Appendix J.

# 5 Conclusions and outlook

We analyzed one of the simplest examples of non-invertible global symmetry associated with the Kramers-Wannier transformation (21) on a tensor product Hilbert space of $L$ qubits on a periodic one-dimensional chain. The critical Ising Hamiltonian (10) serves as a prototypical example. The non-invertible Kramers-Wannier symmetry of the continuum Ising CFT is not emergent; rather, it is exact and emanates from this lattice symmetry [13]. We also discussed deformations of the critical Ising Hamiltonian preserving this non-invertible symmetry (Section 4.1). The non-invertible symmetry manifests itself in two different ways, as an operator and as a defect.

First, a symmetry leads to a conserved *operator* D, which commutes with the Hamiltonian (Section 2.3). This operator has a kernel, and is therefore non-invertible. This operator admits an MPO presentation (56), which makes it manifestly translation invariant and local.

The operator algebra (74) of D involves both the invertible $\mathbb{Z}_2$ spin-flip symmetry $\eta$ and the lattice translation operator $T$, i.e. $D^2 = (1 + \eta)T^{-1}$. This is to be contrasted with the non-invertible symmetry on the anyonic chain [14, 16, 120–124], which does not mix with the lattice translation, $\mathcal{N}^2 = 1 + \eta$. This suggests a tension between a tensor product Hilbert space, and an internal non-invertible Kramers-Wannier symmetry that does not mix with the lattice translation.

---

[40]For more general models, $T$ might also be spontaneously broken. When this is the case, the number of nearly degenerate ground states may depend on the number theoretic property of $L$, but it is always a multiple of 3.

[41]In contrast, the leading thermal operator $\varepsilon$ with $(h, \bar{h}) = (1/10, 1/10)$ in the tricritical Ising CFT preserves the $\mathbb{Z}_2$ symmetry $\eta$, but breaks the non-invertible Kramers-Wannier symmetry $\mathcal{N}$. The deformation of $\beta$ from the critical point at $(\beta_c, \lambda_c)$ in Figure 1 is the deformation by the operator $\varepsilon$.

Not every operator that commutes with the Hamiltonian qualifies as a symmetry; a symmetry has to respect certain notions of locality. In particular, it has to lead to a *defect*, which is localized in space and extends in the time direction. In Section 2.2, we discussed the non-invertible defect $\mathcal{D}$ associated with the non-invertible symmetry D and represented it as a modification of the Hamiltonian in a local region in space. Using the local unitary operator that moves the position of the defect, we defined a lattice fusion rule for the defects, which includes $\mathcal{D} \otimes \mathcal{D} = (1 \oplus \eta) \otimes \mathcal{T}^-$, where $\mathcal{T}^-$ is a defect for the lattice translation that removes a qubit (Appendix E). In Section 2.3.1, we related these two different aspects of the symmetry by providing a derivation of the operator D from the defect $\mathcal{D}$.

In Section 2.4, we studied the symmetry operators in the presence of various defects and analyzed their algebras. Some of these algebras involving parity/time-reversal are realized projectively (Section 2.5.)

In Section 3, we compared the lattice symmetry and the continuum symmetry described by a fusion category. Since the lattice symmetry mixes with the lattice translation, it does not form a fusion category (see Section 1.2). In fact, depending on the choice of the Hamiltonian, the same lattice operator D (56) can lead to distinct fusion categories in the continuum limit (Section 3.1). Also, the Frobenius-Schur indicator $\epsilon$ of the continuum fusion category is not meaningful on the lattice (Section 3.2). We further discussed certain lattice F-moves (Section 3.3) and defined the lattice quantum dimensions (Section 3.4) and compared them with the analogous continuum quantities.

In Section 4.1, we discussed more general Hamiltonians invariant under the non-invertible symmetry. And Section 4.2 discussed an LSM-type constraint based on the non-invertible lattice symmetry D. We argued that in the thermodynamic limit, a D-invariant Hamiltonian is either gapless or gapped with the number of degenerate ground states being a multiple of 3. In the latter scenario, we interpreted the lattice non-invertible symmetry D as being spontaneously broken.

This work suggests a number of interesting questions for further study:

- What is the mathematical structure of these lattice defects that mix with the lattice translation? While we have performed some preliminary analyses of the lattice F-moves, we leave a more systematic investigation (such as the modified pentagon identity) for the future.

- Which fusion category symmetries in the continuum can be realized exactly on a tensor product Hilbert space, possibly at the price of mixing with the lattice translation?

- Is there a deeper relation between MPOs and non-invertible symmetries on a one-dimensional chain? There has been a lot of studies on the fusion category structure of MPOs in [16, 17, 127, 139–143]. In contrast, as emphasized in Section 3, our MPO (56) does not form a fusion category because of the mixing with the lattice translation.

- What are the more general LSM-type constraints?

- Are there interesting generalizations to higher spacetime dimensions?

## Acknowledgments

We are grateful to Tom Banks, Maissam Barkeshli, Yichul Choi, Paul Fendley, Tarun Grover, Pranay Gorantla, Zohar Komargodski, Justin Kulp, Max Metlitski, Greg Moore, Brandon Rayhaun, Eric Rowell, Subir Sachdev, Nikita Sopenko, Nathanan Tantivasadakarn, Senthil Todadri, Frank Verstraete, Zhenghan Wang, Tzu-Chieh Wei, Xiao-Gang Wen, Zeqi Zhang, Yunqin Zheng for interesting discussions. The authors of this paper were ordered alphabetically.

**Funding information**    The work of NS was supported in part by DOE grant DE-SC0009988. This work was also supported by the Simons Collaboration on Ultra-Quantum Matter, which is a grant from the Simons Foundation (651444, NS, SHS). SS gratefully acknowledges support from the U.S. Department of Energy grant DE-SC0009988, the Sivian Fund, and the Paul Dirac Fund at the Institute for Advanced Study. This work was performed in part at Aspen Center for Physics during the workshop "Traversing the Particle Physics Peaks: Phenomenology to Formal," which is supported by National Science Foundation grant PHY-2210452.

# A    Notations and conventions

**Quantum gates**

We use the following standard conventions for Pauli matrices

$$Z_j = |0\rangle\langle 0|_j - |1\rangle\langle 1|_j, \quad X_j = |1\rangle\langle 0|_j + |0\rangle\langle 1|_j, \quad Y_j = i|1\rangle\langle 0|_j - i|0\rangle\langle 1|_j, \tag{A.1}$$

and

$$|\pm\rangle_j = \frac{1}{\sqrt{2}}\left(|0\rangle_j \pm |1\rangle_j\right). \tag{A.2}$$

We also use the following gates commonly used in the quantum information literature:

$$
\begin{aligned}
\mathsf{S}_{j,k} &= \frac{1 + X_j X_k + Y_j Y_k + Z_j Z_k}{2}, \\
\mathsf{H}_j &= \frac{X_j + Z_j}{\sqrt{2}} = |+\rangle\langle 0|_j + |-\rangle\langle 1|_j = |0\rangle\langle +|_j + |1\rangle\langle -|_j, \\
\mathsf{CZ}_{j,k} &= \frac{1 + Z_j + Z_k - Z_j Z_k}{2} = |0\rangle\langle 0|_j + |1\rangle\langle 1|_j \otimes Z_k, \\
\mathsf{CNOT}_{j,k} &= \mathsf{CX}_{j,k} = \mathsf{H}_k \mathsf{CZ}_{j,k} \mathsf{H}_k = \frac{1 + Z_j + X_k - Z_j X_k}{2} = |0\rangle\langle 0|_j + |1\rangle\langle 1|_j \otimes X_k.
\end{aligned}
\tag{A.3}
$$

The swap operator acts on the local operators as

$$\mathsf{S}_{j,k}: \begin{array}{ll} X_j \mapsto X_k, & X_k \mapsto X_j, \\ Z_j \mapsto Z_k, & Z_k \mapsto Z_j. \end{array} \tag{A.4}$$

This determines $\mathsf{S}_{j,k}$ up to a phase. The phase choice in $\mathsf{S}_{j,k}$ above is natural because our Hilbert space is a tensor product Hilbert space $\mathcal{H} = \mathcal{H}_1 \otimes \cdots \otimes \mathcal{H}_L$ and the map $\mathsf{S}_{j,k}$ implements a canonical isomorphism between the factors.

The Hadamard gate acts as

$$\mathsf{H}_j: X_j \mapsto Z_j, \ Z_j \mapsto X_j. \tag{A.5}$$

Finally, CZ and CNOT act on the local operators as:

$$
\begin{aligned}
\mathsf{CZ}_{j,k}: & \begin{array}{ll} X_j \mapsto X_j Z_k, & Z_j \mapsto Z_j, \\ X_k \mapsto X_k Z_j, & Z_k \mapsto Z_k, \end{array} \\[1em]
\mathsf{CNOT}_{j,k}: & \begin{array}{ll} X_j \mapsto X_j X_k, & Z_j \mapsto Z_j, \\ X_k \mapsto X_k, & Z_k \mapsto Z_k Z_j. \end{array}
\end{aligned}
\tag{A.6}
$$

# B   Gauging and non-invertible symmetries

In this appendix, we discuss the relation between gauging the $\mathbb{Z}_2$ symmetry generated by

$$\eta = \prod_{j=1}^{L} X_j, \tag{B.1}$$

and the non-invertible symmetry.

After reviewing the gauging procedure in Appendix B.1, in Appendices B.2 and B.3 we relate the defect $\mathcal{D}$ and the symmetry operator D to gauging in half of space or half of time respectively. Then, in Appendix B.4, we argue that on a closed periodic chain, a translation-invariant, finite-range Hamiltonian $H$ commutes with the non-invertible operator D if, and only if, the system based on $H$ is invariant under gauging its global $\mathbb{Z}_2$ symmetry generated by $\eta$.

## B.1   Review of gauging the $\mathbb{Z}_2$ symmetry

In order to set the notation, we start by reviewing the gauging of the spin system.

Our space is a 1d closed, periodic chain of $L$ qubits. The Hilbert space $\mathcal{H}$ is $2^L$-dimensional. We consider the most general finite-range Hamiltonian, which is invariant under the on-site $\mathbb{Z}_2$ global symmetry $\eta = \prod_{j=1}^{L} X_j$ and the lattice translation symmetry $T : j \to j + 1$.

A local term in the Hamiltonian takes the form

$$\left(X_{j_1} X_{j_2} \cdots X_{j_n}\right)\left(Z_{k_1} Z_{k'_1}\right)\left(Z_{k_2} Z_{k'_2}\right)\cdots\left(Z_{k_m} Z_{k'_m}\right), \tag{B.2}$$

where all the sites $j_1 < j_2 < \cdots < j_n$ and $k_1 < k'_1 < k_2 < k'_2 \cdots < k_m < k'_m$ are within some finite region much smaller than $L$. (We allow some of the site indices $j_\ell, k_\ell, k'_\ell$ to be negative, so that the interaction covers a range around the link $(L, 1)$.) Note that the $\mathbb{Z}_2$ symmetry constrains the number of $Z_j$'s to be even. Invariance under $T$ implies that we have to sum terms like (B.2) over the value of one integer, say $j_1$ or $k_1$ keeping the differences between the indices fixed.

A simple example is the transverse-field Ising model at a generic coupling $g$:

$$H(g) = -\sum_{j=1}^{L}(g^{-1} Z_{j-1} Z_j + g X_j), \tag{B.3}$$

which is critical for $g = 1$.

We will refer to the theory based on the Hamiltonian $H$ as a "matter theory."

Next, we would like to gauge the global $\mathbb{Z}_2$ symmetry generated by $\eta = \prod_{j=1}^{L} X_j$.

The first step involves adding a new qubit on every link $(j - 1, j)$, corresponding to the $\mathbb{Z}_2$ gauge field. We denote the corresponding Pauli matrices on the links as $\tilde{X}_{j-\frac{1}{2}}, \tilde{Z}_{j-\frac{1}{2}}$. We couple every local term of the form (B.2) in the original Hamiltonian to the $\mathbb{Z}_2$ gauge field as follows:

$$\left(X_{j_1} X_{j_2} \cdots X_{j_n}\right)\left(Z_{k_1} \tilde{X}_{k_1+\frac{1}{2}} \cdots \tilde{X}_{k'_1-\frac{1}{2}} Z_{k'_1}\right)\cdots\left(Z_{k_m} \tilde{X}_{k_m+\frac{1}{2}} \cdots \tilde{X}_{k'_m-\frac{1}{2}} Z_{k'_m}\right). \tag{B.4}$$

At this point, we obtain a gauged Hamiltonian $H_{\text{gauged}}$ acting on a $2^{2L}$-dimensional Hilbert space $\mathcal{H}_{\text{gauge}}$, with a qubit at every site and link.

In the special case of the Ising Hamiltonian (B.3), we find

$$H_{\text{gauged}}(g) = -\sum_{j=1}^{L}\left(g^{-1} Z_{j-1} \tilde{X}_{j-\frac{1}{2}} Z_j + g X_j\right). \tag{B.5}$$

This system with its $2^{2L}$-dimensional Hilbert space $\mathcal{H}_{\text{gauge}}$ and Hamiltonian $H_{\text{gauged}}$ has a large $\mathbb{Z}_2^L$ symmetry generated by

$$G_j = \tilde{Z}_{j-\frac{1}{2}} X_j \tilde{Z}_{j+\frac{1}{2}}, \quad j = 1, \cdots, L. \tag{B.6}$$

One combination of these, $\prod_j G_j = \prod_j X_j = \eta$ is the original $\mathbb{Z}_2$ global symmetry of the matter theory.

In the second step of the gauging, we project the Hilbert space on states invariant under this large $\mathbb{Z}_2^L$ symmetry by imposing the Gauss law constraint[42]

$$G_j = 1, \quad j = 1, \cdots, L. \tag{B.7}$$

We end up with a physical Hilbert space of dimension $2^L$.

In this discussion, we consider what can be called "minimal coupling." This means that in the first step of the gauging, we do not add to the Hamiltonian of the gauged system additional terms that depend on the gauge fields and commute with Gauss law (B.7). A subset of such terms, like arbitrary local translation invariant terms that depend only on $\tilde{Z}_{j+\frac{1}{2}}$, e.g., $\sum_j \tilde{Z}_{j+\frac{1}{2}}$ can be excluded by imposing a "quantum $\mathbb{Z}_2^{\tilde{\eta}}$ symmetry," generated by $\tilde{\eta} = \prod_j \tilde{X}_{j+\frac{1}{2}}$. This symmetry will be important below.

**An effective description of the gauged system**

The operators that commute with the Gauss law act within this new $2^L$-dimensional Hilbert space. These gauge-invariant operators are generated by:

$$\begin{aligned}
\widehat{Z}_{j-\frac{1}{2}} &= \tilde{Z}_{j-\frac{1}{2}}, \\
\widehat{X}_{j-\frac{1}{2}} &= Z_{j-1} \tilde{X}_{j-\frac{1}{2}} Z_j,
\end{aligned} \tag{B.8}$$

with $j = 1, \cdots, L$. Using the Gauss law, the local Hamiltonian term (B.4) is now written in terms of these gauge-invariant operators as

$$\left(\widehat{Z}_{j_1-\frac{1}{2}} \widehat{Z}_{j_1+\frac{1}{2}}\right) \cdots \left(\widehat{Z}_{j_n-\frac{1}{2}} \widehat{Z}_{j_n+\frac{1}{2}}\right) \left(\widehat{X}_{k_1+\frac{1}{2}} \widehat{X}_{k_1+\frac{3}{2}} \cdots \widehat{X}_{k_1'-\frac{1}{2}}\right) \cdots \left(\widehat{X}_{k_m+\frac{1}{2}} \widehat{X}_{k_m+\frac{3}{2}} \cdots \widehat{X}_{k_m'-\frac{1}{2}}\right). \tag{B.9}$$

In the special case of the Ising Hamiltonian, the gauged Hamiltonian is

$$H_{\text{gauged}}(g) = -\sum_{j=1}^{L} \left( g \widehat{Z}_{j-\frac{1}{2}} \widehat{Z}_{j+\frac{1}{2}} + g^{-1} \widehat{X}_{j-\frac{1}{2}} \right). \tag{B.10}$$

By comparing the terms (B.9) in the gauged Hamiltonian with the terms (B.2) in the original matter Hamiltonian, we see the gauging maps

$$\begin{aligned}
X_j &\rightsquigarrow \widehat{Z}_{j-\frac{1}{2}} \widehat{Z}_{j+\frac{1}{2}}, \\
Z_{j-1} Z_j &\rightsquigarrow \widehat{X}_{j-\frac{1}{2}}.
\end{aligned} \tag{B.11}$$

If we rename

$$\widehat{Z}_{j-\frac{1}{2}} \rightarrow Z_{j-1}, \qquad \widehat{X}_{j-\frac{1}{2}} \rightarrow X_{j-1}, \tag{B.12}$$

then (B.11) coincides with the action of D on the $\mathbb{Z}_2$-even operators (63). The half translation (B.12) provides an isomorphism between the initial and final $2^L$-dimensional Hilbert spaces. We denote both of them as $\mathcal{H}$.

---

[42]It is common to implement Gauss law energetically by adding to the Hamiltonian $-\Lambda \sum_j G_j$ with large positive $\Lambda$, such that the low-lying states satisfy Gauss law, but higher energy states do not. Instead, we will be studying the case $\Lambda \to \infty$, where Gauss law is satisfied on all the states in the Hilbert space.

We conclude that *every* D-*invariant Hamiltonian is invariant under gauging the* $\mathbb{Z}_2$ *symmetry*.

Note that invariance under D implies the invariance under the $\mathbb{Z}_2$ symmetry $\eta$ and lattice translation $T$ since $D^2$ involves both $\eta$ and $T$.[43] In the particular case of the Ising Hamiltonian $H(g)$, we have

$$DH(g) = H(1/g)D. \tag{B.14}$$

Hence D is a symmetry only at $g = 1$. This is indeed the value that the Hamiltonian is invariant under gauging. See Section 4.1 and references therein for more general Hamiltonians that commute with D.

**Another effective description**

Instead of using the variables $\widehat{Z}_{j-\frac{1}{2}}$ and $\widehat{X}_{j-\frac{1}{2}}$ (B.8), we can consider

$$\begin{aligned}
Z'_j &= \tilde{X}_{\frac{1}{2}} \tilde{X}_{\frac{3}{2}} \cdots \tilde{X}_{j-\frac{1}{2}} Z_j\,, \\
X'_j &= X_j\,.
\end{aligned} \tag{B.15}$$

In terms of these, the gauged Hamiltonian (B.5) becomes

$$\begin{aligned}
H_{\text{gauged}}(g) &= -\sum_{j=2}^{L} \left( g^{-1} Z'_{j-1} Z'_j + g X'_j \right) - g^{-1} \tilde{\eta} Z'_L Z'_1 - g X'_1\,, \\
\tilde{\eta} &= \tilde{X}_{\frac{1}{2}} \tilde{X}_{\frac{3}{2}} \cdots \tilde{X}_{L-\frac{1}{2}}\,.
\end{aligned} \tag{B.16}$$

Just like the original "matter variables" $X$ and $Z$, the new variables $X'$ and $Z'$ satisfy standard commutation relations and we can think of them as new "matter variables."

Next, we discuss the gauge fields $\tilde{X}$ and $\tilde{Z}$. First, it seems that we need to tensor a $2^L$ dimensional Hilbert space for them, leading to a $2^{2L}$-dimensional Hilbert space. However, since they appear in the Hamiltonian (B.16) only through the dependence on $\tilde{\eta}$, it is enough to tensor only a single qubit on which $\tilde{\eta} = \pm 1$. At this stage, we can restrict our Hilbert space to be $2^{L+1}$-dimensional.[44] We should also account for Gauss law (B.7) $G_j = \tilde{Z}_{j-\frac{1}{2}} X_j \tilde{Z}_{j+\frac{1}{2}} = 1$. It is easy to check that the new matter variables $X'$ and $Z'$ commute with all the Gauss law constraints, except for $G_L = \tilde{Z}_{L-\frac{1}{2}} X_L \tilde{Z}_{\frac{1}{2}}$. Equivalently, the $L - 1$ Gauss law operators $G_j$ with $j = 1, \cdots, L-1$ act as 1 in the $2^{L+1}$-dimensional Hilbert space, and we only need to impose one Gauss law constraint

$$G_1 G_2 \cdots G_L = X_1 X_2 \cdots X_L = X'_1 X'_2 \cdots X'_L = \eta = 1\,. \tag{B.17}$$

After this final Gauss law constraint is imposed, we end up with a $2^L$-dimensional Hilbert space.

---

[43]In general, if a finite-range Hamiltonian is invariant under $(1+h)g$, for invertible symmetries $g$ and $h$, it must be invariant under $g$ and $h$ separately. To see this, we write $H = \sum_j H_j$, where $H_j$ is localized around site $j$. Then, the invariance under $(1+h)g$ implies

$$(1+h) \sum_j g H_j g^{-1} = \sum_j H_j (1+h)\,. \tag{B.13}$$

Now separating the sums on both sides into local terms and non-local terms acting on the whole chain we find that the equation above implies $gHg^{-1} = H$ and $hHg^{-1} = Hh$ separately.

For our discussion here, the D-invariance implies that the Hamiltonian is also invariant under $D^2 = T^{-1}(1+\eta)$. Setting $g = T^{-1}$ and $h = \eta$ in the general argument above then implies the Hamiltonian is also invariant under $T, \eta$.

[44]At the critical point $g = 1$, the Hamiltonian on this $2^{L+1}$-dimensional Hilbert space is (47), with $Z, X$ there replaced by $Z', X'$ here, and $Z_{(L,1)} \to \tilde{\eta}, X_{(L,1)} \to \tilde{Z}_{L-\frac{1}{2}} X_L$.

In summary, the gauged Ising model can be written as the Hamiltonian (B.16), with $\tilde{\eta} = \pm 1$ and the Hilbert space should be subject to a single Gauss law constraint $\eta = 1$. In more detail, this means that we have a direct sum of two copies of the original, matter Ising model, one corresponding to $\tilde{\eta} = +1$ and the other to $\tilde{\eta} = -1$. The Hamiltonian in the first one, $\tilde{\eta} = +1$, is the standard Ising Hamiltonian. And the Hamiltonian acting on the second one, $\tilde{\eta} = -1$, is the Hamiltonian of the $\mathbb{Z}_2$ twisted Ising model. And in addition, we should impose that in each of them we project on the $\eta = +1$ states.

This picture is similar to the way the gauging is described in the continuum field theory. In that context, the two Hilbert spaces are referred to as the original and the twisted Hilbert spaces and the projection on $\eta = +1$ is the projection on gauge invariant states. (See, for example, [7, 144] for reviews of orbifold in 1+1d continuum CFT.)

## B.2 The non-invertible defect $\mathcal{D}$ from gauging in half of space

In Section B.1 we show that $H\mathrm{D} = \mathrm{D}H$ implies that the Hamiltonian is invariant under $\mathbb{Z}_2$ gauging. Below, we discuss the converse. We will show that the invariance under gauging the $\mathbb{Z}_2$ symmetry leads to the non-invertible defect $\mathcal{D}$, which in turn gives the non-invertible operator D by the construction in Section 2.3.1.

Non-invertible duality symmetries typically exist when the system is invariant under gauging a discrete (possibly higher-form) global symmetry. When this is the case, one can gauge the global symmetry in half of the spacetime, and impose a topological Dirichlet boundary condition for the discrete gauge fields at the interface. This generally gives rise to a non-invertible topological defect. This procedure is known as *half-gauging* [98, 99] and has been used to construct a large class of non-invertible symmetries in quantum systems in diverse spacetime dimensions, including the 1+1d Ising CFT [98], the 3+1d Maxwell gauge theory [98–100, 145, 146], axions [147], and QED [148]. In particular, lattice examples using the Euclidean modified Villain formulation [149, 150] were provided in [98].[45] See [7] for a review.

Here we apply the half gauging construction to 1+1d lattice model on a tensor product Hilbert space. For notational simplicity we focus on the transverse-field Ising model, but our construction holds for more general Hamiltonians as we will discuss at the end of this subsection.[46]

We start with the transverse-field Ising model (B.3) at a generic coupling $g$ on a closed periodic chain of $L$ sites, with a $2^L$-dimensional Hilbert space. Later, we will restrict to the critical point $g = 1$. Half-gauging of the Ising lattice model on an open chain was recently discussed in [152].

We gauge the $\mathbb{Z}_2$ symmetry in the segment $1 \leq j \leq J$ on the closed chain with some $1 < J < L$. (In Section B.1, we reviewed the ordinary gauging of the $\mathbb{Z}_2$ global symmetry on the entire lattice.) We introduce $\mathbb{Z}_2$ gauge fields $\tilde{X}_{j-\frac{1}{2}}$ only on the links $(j-1, j)$ with $1 \leq j \leq J$ to find the Hamiltonian

$$H_{\mathrm{h.g.}}(g) = -\sum_{j=1}^{J}(g^{-1}Z_{j-1}\tilde{X}_{j-\frac{1}{2}}Z_j + gX_j) - \sum_{j=J+1}^{L}(g^{-1}Z_{j-1}Z_j + gX_j). \tag{B.18}$$

The enlarged Hilbert space is now $2^{L+J}$-dimensional. Next, we impose Gauss law as a projection on this enlarged Hilbert space:

$$G_j = \tilde{Z}_{j-\frac{1}{2}}X_j\tilde{Z}_{j+\frac{1}{2}} = 1, \quad j = 1, 2, \cdots, J-1. \tag{B.19}$$

---

[45]It would also be interesting to realize non-invertible symmetries in the Hamiltonian formalism of the modified Villain lattice model [52, 151].

[46]See [13] and Appendix C for an alternative construction of the defect Hamiltonian for $\mathcal{D}$ of the critical transverse-field Ising model from bosonization of the Majorana chain on odd number of sites.

Importantly, since we only have $J-1$ Gauss laws but $J$ qubits from the $\mathbb{Z}_2$ gauge fields, the projected Hilbert space is $2^{L+1}$-dimensional and has one more qubit than the initial Hilbert space. (This is to be contrasted with the ordinary gauging on the entire closed chain, reviewed in Section B.1, where the projected Hilbert space has the same dimension as the initial Hilbert space.) Some of the original local operators (e.g., $Z_1$) do not act within this $2^{L+1}$-dimensional Hilbert space because they do not commute with the Gauss law constraints.

Next, we follow the strategy we used above to find an effective description of the $2^{L+1}$-dimensional physical Hilbert space. In particular, we would like to find the operators acting in that subspace. They are the operators that commute with all the Gauss law constraints. These gauge-invariant operators are generated by the following list of local operators:

$$
\begin{aligned}
\widehat{X}_{j-\frac{1}{2}}, \widehat{Z}_{j-\frac{1}{2}}, & \qquad j = 1, \cdots J, \\
X_j, Z_j, & \qquad j = J, \cdots L.
\end{aligned}
\tag{B.20}
$$

where

$$
\begin{aligned}
\widehat{Z}_{j-\frac{1}{2}} &= \tilde{Z}_{j-\frac{1}{2}}, & j &= 1, 2, \cdots, J, \\
\widehat{X}_{\frac{1}{2}} &= \tilde{X}_{\frac{1}{2}} Z_1, \\
\widehat{X}_{j-\frac{1}{2}} &= Z_{j-1} \tilde{X}_{j-\frac{1}{2}} Z_j, & j &= 2, 3, \cdots, J-1, \\
\widehat{X}_{J-\frac{1}{2}} &= Z_{J-1} \tilde{X}_{J-\frac{1}{2}}.
\end{aligned}
\tag{B.21}
$$

These $L+1$ pairs of operators in (B.20) satisfy the standard algebra of Pauli matrices. Using the Gauss law, the Hamiltonian can be written entirely in terms of these new set of local operators:

$$
\begin{aligned}
H_{\text{h.g.}}(g) = & -g^{-1} Z_L \widehat{X}_{\frac{1}{2}} - \sum_{j=2}^{J-1} \left( g \widehat{Z}_{j-\frac{3}{2}} \widehat{Z}_{j-\frac{1}{2}} + g^{-1} \widehat{X}_{j-\frac{1}{2}} \right) \\
& - g \widehat{Z}_{J-\frac{3}{2}} \widehat{Z}_{J-\frac{1}{2}} - g^{-1} \widehat{X}_{J-\frac{1}{2}} Z_J - g X_J - \sum_{j=J+1}^{L} \left( g^{-1} Z_{j-1} Z_j + g X_j \right).
\end{aligned}
\tag{B.22}
$$

Finally, similar to (B.12), we relabel the hatted operators by a half translation and drop the hats, $\widehat{X}_{j-\frac{1}{2}} \to X_j, \widehat{Z}_{j-\frac{1}{2}} \to Z_j$ for $j = 1, 2, \cdots, J-1$. For the last pair of hatted operators, we rename $\widehat{X}_{J-\frac{1}{2}}, \widehat{Z}_{J-\frac{1}{2}}$ as $X_{(J-1,J)}, Z_{(J-1,J)}$. To simplify the expression, we further conjugate the Hamiltonian $H_{\text{h.g.}}(g)$ by the unitary operator $\mathsf{CZ}_{(J-1,J),J}$ to obtain $H_{\mathcal{D};\mathcal{D}^*}^{(L,1);(J-1,J)}(g)$:

$$
\begin{aligned}
H_{\mathcal{D};\mathcal{D}^*}^{(L,1);(J-1,J)}(g) = & -g^{-1} Z_L X_1 - \sum_{j=2}^{J-1} \left( g Z_{j-1} Z_j + g^{-1} X_j \right) - \left( g Z_{J-1} Z_{(J-1,J)} + g^{-1} X_{(J-1,J)} \right) \\
& - g Z_{(J-1,J)} X_J - \sum_{j=J+1}^{L} \left( g^{-1} Z_{j-1} Z_j + g X_j \right).
\end{aligned}
\tag{B.23}
$$

In this final expression, we see the Ising model with coupling $g$ in the segment $J < j \leq L$, and with coupling $g^{-1}$ in the segment $1 < j < J$. Around the link $(L, 1)$, we find a Kramers-Wannier *interface* between the high and low temperature phases of the Ising model, and we find the dual interface around the link $(J-1, J)$. We summarize these steps of half-gauging in Figure 4.

Something special happens at the critical point $g = 1$. The local terms in the two regions are now identical, and the interfaces become topological *defects* in a single system. More specifically, the Hamiltonian $H_{\mathcal{D};\mathcal{D}^*}^{(L,1);(J-1,J)}(g = 1)$ has a $\mathcal{D}$ defect (22) at one end of the half gauging segment near the link $(L, 1)$:

$$
\mathcal{D}: \quad \cdots - (Z_{L-1} Z_L + X_L) - Z_L X_1 - (Z_1 Z_2 + X_2) - \cdots.
\tag{B.24}
$$

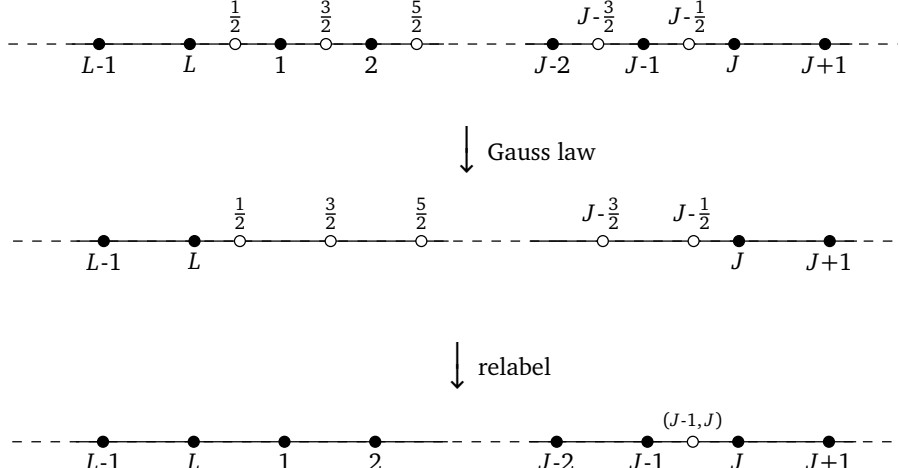

Figure 4: The Hilbert space in each step of the half-gauging between site 0 and site $J$ on a closed, periodic Ising chain with $L$ sites. In the first line, the black dots stand for the original qubits with Pauli operators $X_j, Z_j$ on each site $j = 1, 2, \cdots, L$. The white dots stand for the qubits on the links with Pauli operators $\tilde{X}_{j-\frac{1}{2}}, \tilde{Z}_{j-\frac{1}{2}}$ for the $\mathbb{Z}_2$ gauge fields. In the second line we impose the Gauss law constraints, and the local operators are now generated by $X_j, Z_j$ for $j = J, \cdots, L$ (black dots) and $\hat{X}_{j-\frac{1}{2}}, \hat{Z}_{j-\frac{1}{2}}$ for $j = 1, \cdots, J$ (white dots). In the third line we rename the qubits on the links to the sites by $\hat{X}_{j-\frac{1}{2}} \to X_j, \hat{Z}_{j-\frac{1}{2}} \to Z_j$ for $j = 1, 2, \cdots, J-1$ (black dots), and $\hat{X}_{J-\frac{1}{2}} \to X_{(J-1,J)}, \hat{Z}_{J-\frac{1}{2}} \to Z_{(J-1,J)}$ (white dot). In the end we find a duality defect $\mathcal{D}$ on the link $(L, 1)$ and its dual defect $\mathcal{D}^*$ on the link $(J-1, J)$. The latter defect involves an extra qubit labeled by the white dot.

At the other end, there is an additional site at the link $(J-1, J)$ with Pauli operators $X_{(J-1,J)}, Z_{(J-1,J)}$. This is precisely the dual of the non-invertible defect, $\mathcal{D}^* = \mathcal{D} \otimes \mathcal{T}^+$:

$$
\begin{aligned}
\mathcal{D}^*: \quad &\cdots - \left(Z_{J-2}Z_{J-1} + X_{J-1}\right) - \left(Z_{J-1}Z_{(J-1,J)} + X_{(J-1,J)}\right) \\
&- Z_{(J-1,J)}X_J - \left(Z_J Z_{J+1} + X_{J+1}\right) - \cdots .
\end{aligned}
\tag{B.25}
$$

See Appendix E.3 for more discussions on this dual defect. We conclude that gauging in a segment of the lattice yields the non-invertible duality defect $\mathcal{D}$ at one end and its dual $\mathcal{D}^*$ at the other end.

To demonstrate the generality of the half-gauging construction, we apply it to the deformations in (118) and (119) and derive the corresponding defect. Locally, the non-invertible duality defect $\mathcal{D}$ at the link $(L, 1)$ for the deformation (118) is

$$
\begin{aligned}
&\cdots + \lambda_1\left(X_{L-2}X_{L-1} + Z_{L-2}Z_L\right) \\
&+ \lambda_1\left(X_{L-1}X_L + Z_{L-1}X_1 + X_L Z_1 Z_2 + Z_L X_1 X_2 + Z_1 Z_3\right) \\
&+ \lambda_1\left(X_2 X_3 + Z_2 Z_4\right) + \cdots ,
\end{aligned}
\tag{B.26}
$$

where the terms represented by the ellipses are of the form $\lambda_1(X_j X_{j+1} + Z_j Z_{j+2})$. Similarly, the non-invertible defect $\mathcal{D}$ at the link $(L, 1)$ for the deformation (119) is locally

$$
\begin{aligned}
&\cdots + \lambda_2\left(X_{L-2}Z_{L-1}Z_L + Z_{L-2}Z_{L-1}X_L\right) \\
&+ \lambda_2\left(X_{L-1}Z_L X_1 + Z_{L-1}Z_L Z_1 Z_2 + X_L X_2 + Z_L X_1 Z_2 Z_3 + Z_1 Z_2 X_3\right) \\
&+ \lambda_2\left(X_2 Z_3 Z_4 + Z_2 Z_3 X_4\right) + \cdots .
\end{aligned}
\tag{B.27}
$$

One can check that the same unitary movement operator $U_{\mathcal{D}}^j = \mathsf{CZ}_{j+1,j}\mathsf{H}_j$ in (24) can be applied to move these $\mathcal{D}$ defects in the deformed Hamiltonians. These deformed Hamiltonians are also invariant under $\eta_{\mathcal{D}}$ and $\mathsf{D}_{\mathcal{D}}$ in (85).

To summarize, *the invariance under gauging the $\mathbb{Z}_2$ global symmetry implies the existence of a topological non-invertible defect $\mathcal{D}$.*

### B.3 The non-invertible operator $\mathsf{D}$ as gauging in the future

Just as in Appendix B.2 we described the defect $\mathsf{D}$ as gauging in part of space, here we will describe the symmetry operator $\mathsf{D}$ as gauging in part of time.

Closely related constructions have appeared in [14–16, 18, 26]. Here, we will emphasize the interpretation in terms of gauging and will insist on $\mathsf{D}$ being a map from the Hilbert space $\mathcal{H}$ to itself rather than to another Hilbert space.

The idea is to use the equivalence between the matter system with Hamiltonian terms like (B.4) and the effective description of the gauged system with Hamiltonian term like (B.9). We will describe the system as evolving in the past using the Hamiltonian $H(X,Z)$ acting on a $2^L$-dimensional Hilbert space $\mathcal{H}$. Then, the operator $\mathsf{D}$ acts in the present. In the future, we have the gauge theory, which we will describe using the effective variables $\widehat{X}$ and $\widehat{Z}$. We will shift the site indices by a half so that we can use the variables $X$ and $Z$ acting on *the same $2^L$-dimensional Hilbert space $\mathcal{H}$* with the same Hamiltonian $H(X,Z)$.

As explained in Appendix B.1, the invariance under $\mathsf{D}$ is the statement that the past and the future Hamiltonians are the same, except that they are expressed in terms of different variables. The operator $\mathsf{D}$ acts in the present and implements this change in the variables.

To make it more explicit, let us pick a basis of states in the past with diagonal $X_k$

$$
\begin{aligned}
&\bigotimes_j |x_j\rangle \in \mathcal{H}\,, \\
&X_k \bigotimes_j |x_j\rangle = x_k \bigotimes_j |x_j\rangle\,, \\
&x_j = \pm\,.
\end{aligned}
\tag{B.28}
$$

In the future, it is more convenient we pick a basis with diagonal $Z_k$

$$
\begin{aligned}
&\bigotimes_j |z_j\rangle \in \mathcal{H}\,, \\
&Z_k \bigotimes_j |z_j\rangle = (-1)^{z_k} \bigotimes_j |z_j\rangle\,, \\
&z_j = 0, 1\,.
\end{aligned}
\tag{B.29}
$$

(Recall our conventions, $X|\pm\rangle = \pm|\pm\rangle, Z|0\rangle = |0\rangle, Z|1\rangle = -|1\rangle$.)

The operator $\mathsf{D}$ is then a map from $\mathcal{H} \to \mathcal{H}$. To find this map, we should understand what the gauging in the future does. Before going to the effective theory with $\widehat{X}$ and $\widehat{Z}$ (which we identify with $X$ and $Z$ in the future, after shifting the indices), we have a $2^{2L}$-dimensional Hilbert space $\mathcal{H}_{\text{gauge}}$ with the operators $X, Z, \tilde{X},$ and $\tilde{Z}$. And the physical Hilbert space $\mathcal{H}$ is the Gauss-law-invariant subspace of $\mathcal{H}_{\text{gauge}}$. In order to follow the map, we identify the $2^{2L}$-dimensional Hilbert space $\mathcal{H}_{\text{gauge}}$ with the Hilbert space in the present.

Our operator $\mathsf{D}$ will then be a composition of maps

$$
\mathsf{D}: \quad \mathcal{H} \to \mathcal{H}_{\text{gauge}} \to \mathcal{H}\,.
\tag{B.30}
$$

We find it convenient to pick a basis in the present with diagonal $X_k$ and $\tilde{Z}_{k+\frac{1}{2}}$

$$
\bigotimes_j |x_j\rangle \bigotimes_j \left|\tilde{z}_{j+\frac{1}{2}}\right\rangle \in \mathcal{H}_{\text{gauge}},
$$

$$
X_k \bigotimes_j |x_j\rangle \bigotimes_j \left|\tilde{z}_{j+\frac{1}{2}}\right\rangle = x_k \bigotimes_j |x_j\rangle \bigotimes_j \left|\tilde{z}_{j+\frac{1}{2}}\right\rangle,
$$

$$
\tilde{Z}_{k+\frac{1}{2}} \bigotimes_j |x_j\rangle \bigotimes_j \left|\tilde{z}_{j+\frac{1}{2}}\right\rangle = (-1)^{\tilde{z}_{k+\frac{1}{2}}} \bigotimes_j |x_j\rangle \bigotimes_j \left|\tilde{z}_{j+\frac{1}{2}}\right\rangle.
$$

(B.31)

Then, we map a state in the past $\bigotimes_j |x_j\rangle$ as follows.
If $\eta = \prod_j x_j = -1$, we map it to zero, i.e., the operator D annihilates the state.
If $\eta = \prod_j x_j = +1$, we map

$$
\bigotimes_j |x_j\rangle \to \left( \bigotimes_j |x_j\rangle \bigotimes_j \left|\tilde{z}_{j+\frac{1}{2}}\right\rangle \right) + \left( \bigotimes_j |x_j\rangle \bigotimes_j \left|1 - \tilde{z}_{j+\frac{1}{2}}\right\rangle \right),
$$

(B.32)

where the values of $\tilde{z}_{j-\frac{1}{2}}$ are determined by

$$
(-1)^{\tilde{z}_{j-\frac{1}{2}}} x_j (-1)^{\tilde{z}_{j+\frac{1}{2}}} = 1.
$$

(B.33)

Let us make some comments about this map.

- The equations (B.33) determine $\{\tilde{z}_{j-\frac{1}{2}}\}$ in terms of $\{x_j\}$. This is the statement that the state in the present Hilbert space $\mathcal{H}_{\text{gauge}}$ satisfies Gauss law.

- The conditions (B.33) can be solved for $\{\tilde{z}_{j-\frac{1}{2}}\}$ because we need to solve them only when $\eta = \prod_j x_j = +1$.

- Actually, for every $\{x_j\}$, there are two solutions for $\{\tilde{z}_{j-\frac{1}{2}}\}$, differing by $\{\tilde{z}_{j-\frac{1}{2}}\} \to \{1 - \tilde{z}_{j-\frac{1}{2}}\}$. This fact does not affect the right hand side of (B.32).

- The resulting states in $\mathcal{H}_{\text{gauge}}$ are eigenstate of $\tilde{\eta} = \prod_j \tilde{X}_{j+\frac{1}{2}}$ with eigenvalue $+1$.

To summarize, the states in $\mathcal{H}$ with $\eta = -1$ are mapped to zero and the states with $\eta = +1$ are mapped to physical states in $\mathcal{H}_{\text{gauge}}$ with $\tilde{\eta} = +1$.

Next, we map the states in the present to states in the future

$$
\bigotimes_j |x_j\rangle \bigotimes_j \left|\tilde{z}_{j+\frac{1}{2}}\right\rangle \to \bigotimes_j \left|z_j = \tilde{z}_{j+\frac{1}{2}}\right\rangle.
$$

(B.34)

Therefore, our total map from the past to the future is

$$
\text{D}: \quad \bigotimes_j |x_j\rangle \to 0, \qquad\qquad\qquad \text{for} \quad \eta = \prod_j x_j = -1,
$$

$$
\text{D}: \quad \bigotimes_j |x_j\rangle \to \left( \bigotimes_j |x_j\rangle \bigotimes_j \left|\tilde{z}_{j+\frac{1}{2}}\right\rangle \right) + \left( \bigotimes_j |x_j\rangle \bigotimes_j \left|1 - \tilde{z}_{j+\frac{1}{2}}\right\rangle \right)
$$

$$
\to \bigotimes_j \left|z_j = \tilde{z}_{j+\frac{1}{2}}\right\rangle + \bigotimes_j \left|z_j = 1 - \tilde{z}_{j+\frac{1}{2}}\right\rangle, \qquad \text{for} \quad \eta = \prod_j x_j = +1,
$$

(B.35)

with $(-1)^{\tilde{z}_{j-\frac{1}{2}}} x_j (-1)^{\tilde{z}_{j+\frac{1}{2}}} = 1$. The normalization of this map can be checked by computing $D|+ + \ldots +\rangle = |00\ldots0\rangle + |11\ldots1\rangle$, which is consistent with (123) and $D^2|+ + \ldots +\rangle = 2|+ + \ldots +\rangle$. This map is related to the one in [14, 18, 26].[47]

When we commute operators through D, the operators on the right act in the past and the operators on the left act in the future. Let us consider some examples.

The past operator $X_j$ acts in the present like $\tilde{Z}_{j-\frac{1}{2}} \tilde{Z}_{j+\frac{1}{2}}$ and therefore, it acts in the future like $Z_{j-1} Z_j$. Hence,

$$\mathsf{D} X_j = Z_{j-1} Z_j \mathsf{D} . \tag{B.38}$$

Similarly, the $\mathbb{Z}_2$ invariant past operator $Z_j Z_{j'}$ acts in the present like $Z_j \tilde{X}_{j+\frac{1}{2}} \tilde{X}_{j+\frac{3}{2}} \cdots \tilde{X}_{j'-\frac{1}{2}} Z_{j'} = \widehat{X}_{j+\frac{1}{2}} \widehat{X}_{j+\frac{3}{2}} \cdots \widehat{X}_{j'-\frac{1}{2}}$ and therefore it acts in the future like $X_j X_{j+1} \cdots X_{j'-1}$. Hence,

$$\mathsf{D} Z_j Z_{j'} = X_j X_{j+1} \cdots X_{j'-1} \mathsf{D} . \tag{B.39}$$

(Note that since D projects on states with $\prod_j \tilde{X}_{j+\frac{1}{2}} = \prod X_j = +1$, it does not matter whether the factors of $\tilde{X}$ in the present are inserted from $j$ to $j'$ or the other way around.) In the special case $j' = j + 1$, this becomes

$$\mathsf{D} Z_j Z_{j+1} = X_j \mathsf{D} . \tag{B.40}$$

These are consistent with the action (5).

## B.4 The Hamiltonian is D-invariant iff it is invariant under gauging

We now assemble the results from this appendix to reach the statement that, $H\mathsf{D} = \mathsf{D}H$ if, and only if, the Hamiltonian H is invariant under gauging the $\mathbb{Z}_2$ symmetry on a closed periodic chain.

The only if part already follows from the discussion in Appendix B.1.

The if part follows from Appendix B.2. Assuming that the Hamiltonian is invariant under gauging the $\mathbb{Z}_2$ symmetry, we constructed the duality defect $\mathcal{D}$ by gauging in half of the space. Next, following Section 2.3.1, the defect $\mathcal{D}$ leads to a conserved non-invertible operator D. This completes the argument.

---

[47]In order to relate this D to the construction in [26], we need to make several changes. We should follow our maps from the future to the past rather than the other way around. We should also shift the final result in [26] from variables on the links to variables on the sites, so that we map the Hilbert space to itself. More importantly, we should swap the vertices/sites and the edges/links in the present so that the matter fields are on the sites, the gauge fields are on the links, and Gauss law is on the sites. Finally, we should change the overall normalization.

Specifically, we start with a $\mathbb{Z}_2$ invariant state in the future, say $\bigotimes_j |z_j\rangle + \bigotimes_j |1 - z_j\rangle$. It corresponds to the gauge part in the present

$$|\psi\rangle = \bigotimes_j \left| \tilde{z}_{j+\frac{1}{2}} = z_j \right\rangle + \bigotimes_j \left| \tilde{z}_{j+\frac{1}{2}} = 1 - z_j \right\rangle . \tag{B.36}$$

We add to it the matter part $\bigotimes_j |x_j\rangle$ such that Gauss law is satisfied. This leads to the state

$$\prod_i \mathsf{CZ}_{i-\frac{1}{2},i} \mathsf{CZ}_{i+\frac{1}{2},i} \bigotimes_j |+\rangle_j \otimes |\psi\rangle . \tag{B.37}$$

In the product over $i$, every site $j$ appears twice and it is acted upon only if the two links next to it have different values of $\tilde{z}_{j\pm\frac{1}{2}}$. In that case, its $x_j$ changes from $+$ to $-$. This guarantees that the whole state satisfies Gauss law. (Note that this happens an even number of times and therefore the resulting state has $\prod_j x_j = +1$.) Then, to proceed to the past, we need to remove the gauge part on the links $|\psi\rangle$. We do that by multiplying the state in the present by $2^{\frac{L-2}{2}} \bigotimes_j \langle+|_{j+\frac{1}{2}}$. The resulting state is our state in the past $\bigotimes_j |x_j\rangle$ in (B.28).

## C Relation to the Majorana chain

Here we review the construction of the non-invertible operator D of the transverse-field Ising model from the lattice translation of the Majorana chain [13]. We will relate the non-invertible operator D in (53) to the one derived in [13].

Consider a closed Majorana chain of $2L$ sites indexed by $\ell = 1, 2, \cdots, 2L$ with $\ell \sim \ell + 2L$. There is a Majorana fermion $\chi_\ell$ at every site $\ell$ obeying the algebra $\{\chi_\ell, \chi_{\ell'}\} = 2\delta_{\ell,\ell'}$. We focus on the following two Hamiltonians

$$H_\pm = \pm i\chi_1\chi_{2L} + i\sum_{\ell=1}^{2L-1} \chi_{\ell+1}\chi_\ell. \tag{C.1}$$

The minus sign in the first term $H_-$ represents a $(-1)^F$ defect.[48] We will focus on two invertible symmetries of $H_\pm$:

- Fermion parity $(-1)^F$. It is a $\mathbb{Z}_2$ global symmetry given by

  $$(-1)^F = i^L \chi_1\chi_2\cdots\chi_{2L}, \tag{C.2}$$

  which acts on the fermions as $(-1)^F : \chi_\ell \to -\chi_\ell$. The phase $i^L$ is chosen so that $[(-1)^F]^2 = 1$.

- Majorana translations $T_\pm$, which acts on the fermions as

  $$T_+\chi_\ell(T_+)^{-1} = \chi_{\ell+1}, \quad \forall\, \ell,$$
  $$T_-\chi_\ell(T_-)^{-1} = \begin{cases} \chi_{\ell+1}, & \ell = 1, 2, \cdots, 2L-1, \\ -\chi_1, & \ell = 2L. \end{cases} \tag{C.3}$$

  $T_+$ ($T_-$) is a symmetry of $H_+$ ($H_-$).[49]

Using (C.2) and (C.3), one immediately finds the algebra:

$$T_+(-1)^F = -T_+(-1)^F,$$
$$T_-(-1)^F = T_-(-1)^F. \tag{C.4}$$

The minus sign in the first line was first pointed out in [81,101,102]. In [13], it was interpreted as a mixed anomaly between the internal $(-1)^F$ symmetry and translation and was matched with the 't Hooft anomaly in the continuum Majorana CFT [153].

The explicit expressions for the Majorana lattice translation operators for $H_\pm$ are

$$T_+ = \frac{e^{2\pi i \frac{L-1}{8}}}{(\sqrt{2})^{2L-1}} \chi_1(1+\chi_1\chi_2)(1+\chi_2\chi_3)\cdots(1+\chi_{2L-1}\chi_{2L}),$$
$$T_- = \frac{e^{-2\pi i \frac{L}{8}}}{(\sqrt{2})^{2L-1}} (1-\chi_1\chi_2)(1-\chi_2\chi_3)\cdots(1-\chi_{2L-1}\chi_{2L}). \tag{C.5}$$

The overall phases are chosen so that $(T_+)^{2L} = 1$ and $(T_-)^{2L} = (-1)^F$. We can also deform the Hamiltonians $H_\pm$, while preserving the above two symmetries, but for simplicity we focus on the simplest nearest-neighbor terms above.

---

[48]Equivalently, for the theory with $H_-$, we can use the same Hamiltonian $H_+$ and impose the anti-periodic boundary conditions for the fermion operator. See footnote 5 for these two perspectives on the defect.

[49]In the continuum, the periodic and anti-periodic boundary conditions correspond to the Ramond-Ramond (RR) and Neveu-Schwarz-Neveu-Schwarz (NSNS) boundary condition for a massless Majorana fermion in the continuum. In [13], the translation operators $T_+$ and $T_-$ were denoted as $T_{\text{RR}}$ and $T_{\text{NSNS}}$, respectively.

The Jordan-Wigner transformation pairs up two Majorana sites into an Ising site:

$$\chi_{2j-1} = \left(\prod_{k=1}^{j-1}\sigma_k^x\right)\sigma_j^z, \quad \chi_{2j} = \left(\prod_{k=1}^{j-1}\sigma_k^x\right)\sigma_j^y, \tag{C.6}$$

with $j = 1, 2, \cdots, L$ labeling the Ising site. Here $\sigma_j^{x,y,z}$ are the Pauli matrices at site $j$. The Majorana Hamiltonians $H_\pm$ written in terms of the new Pauli operators are

$$H_\pm = -\sum_{j=1}^{L}\sigma_j^x - \sum_{j=1}^{L-1}\sigma_j^z\sigma_{j+1}^z \pm (-1)^F\sigma_L^z\sigma_1^z,$$
$$(-1)^F = \prod_{j=1}^{L}\sigma_j^x. \tag{C.7}$$

*Locally*, these are the Hamiltonians for the transverse-field Ising model (10) and the same system with a defect (12). However, the last term is non-local in terms of the Pauli operators because of the factor of $(-1)^F$. Indeed, the Majorana chain is not equivalent to the Ising model globally.

To obtain the Ising model *globally*, we follow the bosonization in the continuum, which can be understood as gauging $(-1)^F$. See, for example, [119, 128, 154–160] for reviews of bosonization in 1+1d continuum field theories, and Appendix I for related discussions. On the lattice, we proceed as follows:

- First, we double the Hilbert space $\widetilde{\mathcal{H}} = \mathcal{H}_- \oplus \mathcal{H}_+$, where each of $\mathcal{H}_\pm$ is a copy the $2^L$-dimensional Hilbert space for the Majorana Hamiltonian $H_\pm$. We can implement this doubling by adding an extra qubit and considering the Hamiltonian

$$\widetilde{H} = -i\widetilde{Z}\chi_1\chi_{2L} + i\sum_{\ell=1}^{2L-1}\chi_{\ell+1}\chi_\ell = \begin{pmatrix} H_- & 0 \\ 0 & H_+ \end{pmatrix}$$
$$= -\sum_{j=1}^{L}\sigma_j^x - \sum_{j=1}^{L-1}\sigma_j^z\sigma_{j+1}^z - \widetilde{Z}(-1)^F\sigma_L^z\sigma_1^z, \tag{C.8}$$

  where $\widetilde{Z} = \mathrm{diag}(I, -I)$ is the Pauli $Z$-operator acting on this qubit with $I$ being the $2^L \times 2^L$ identity matrix.

- Second, we project on states satisfying $\widetilde{Z}(-1)^F = 1$.

To get the transverse-field Ising Hamiltonian (10), we define

$$X_j = \sigma_j^x, \qquad X_{L+1} = \widetilde{X},$$
$$Z_j = \sigma_j^z\widetilde{X}, \qquad Z_{L+1} = \widetilde{Z}(-1)^F, \tag{C.9}$$

for $j = 1, \cdots, L$ to find

$$\widetilde{H} = -\sum_{j=1}^{L}X_j - \sum_{j=1}^{L-1}Z_jZ_{j+1} - Z_{L+1}Z_LZ_1, \tag{C.10}$$

and

$$T_+ = \frac{e^{2\pi i\frac{L-1}{8}}}{(\sqrt{2})^{2L-1}}(Z_1X_{L+1})(1-iX_1)(1-iZ_1Z_2)\cdots(1-iZ_{L-1}Z_L)(1-iX_L),$$
$$T_- = \frac{e^{-2\pi i\frac{L}{8}}}{(\sqrt{2})^{2L-1}}(1+iX_1)(1+iZ_1Z_2)\cdots(1+iZ_{L-1}Z_L)(1+iX_L). \tag{C.11}$$

Indeed, the Hamiltonian $\widetilde{H}$ projected onto $Z_{L+1} = 1$, or equivalently onto $\widetilde{Z}(-1)^F = 1$, becomes the untwisted Ising Hamiltonian (10). The $\mathbb{Z}_2$ symmetry of the Ising model is given by $\eta = \prod_{j=1}^{L} X_j = \prod_{j=1}^{L} \sigma_j^x = (-1)^F$.

What happens to the Majorana translation operator $T_\pm$ under the bosonization? The Ising lattice translation $T$ (which obeys $T^L = 1$) is related to the square of the Majorana translation:

$$T = \begin{pmatrix} T_-^2 & 0 \\ 0 & T_+^2 \end{pmatrix}\Big|_{\tilde{Z}(-1)^F=1}. \tag{C.12}$$

However, because of the minus sign in (C.4), the operator $\text{diag}(T_-, T_+)$ does not commute with the projection. Therefore, it does not act in the projected Hilbert space $\mathcal{H}$ for the Ising model. What survives the projection is the operator [13]

$$D = \begin{pmatrix} T_- & 0 \\ 0 & 0 \end{pmatrix}\Big|_{\tilde{Z}(-1)^F=1} = e^{-\frac{2\pi i}{8}} \mathsf{d}_1 \mathsf{d}_2 \cdots \mathsf{d}_{L-1} \frac{1+iX_L}{\sqrt{2}} \frac{1+\prod_{j=1}^{L} X_j}{2}, \tag{C.13}$$

where

$$\mathsf{d}_j = e^{-2\pi i/8} \frac{1+iX_j}{\sqrt{2}} \frac{1+iZ_{j+1}Z_j}{\sqrt{2}} = e^{-2\pi i/8}\left(\mathbb{1}_{j+1} \otimes \frac{1+iX_j}{2} + Z_{j+1} \otimes (iZ_j)\frac{1+iX_j}{2}\right). \tag{C.14}$$

$D$ is the non-invertible operator of the Ising model, which is the Majorana translation in the $\mathbb{Z}_2$ even sector $\eta = +1$ but 0 in the $\mathbb{Z}_2$ odd sector $\eta = -1$. Even though the second expression in terms of $X_j, Z_j$ in (C.13) is not manifestly invariant under the Ising lattice translation, the first expression in terms of $T_-$ makes it clear that $DT = TD$.

We now relate the expression (C.13) to the operator D (56). The movement operator can be written as

$$U_{\mathcal{D}}^j = \frac{1+iZ_{j+1}}{\sqrt{2}} \mathsf{d}_j^\dagger \frac{1-iZ_j}{\sqrt{2}}. \tag{C.15}$$

Using the MPO presentation (56), we find that the unitary factors $\frac{1-iZ_j}{\sqrt{2}}$ cancel in the expression for D and

$$\begin{aligned} \mathsf{D} &= e^{2\pi i/8} \frac{1-iX_L}{2} \mathsf{d}_{L-1}^\dagger \cdots \mathsf{d}_2^\dagger \mathsf{d}_1^\dagger + e^{2\pi i/8}(-iZ_L)\frac{1-iX_L}{2} \mathsf{d}_{L-1}^\dagger \cdots \mathsf{d}_2^\dagger \mathsf{d}_1^\dagger Z_1 \\ &= e^{2\pi i/8} \frac{1+\prod_{j=1}^{L} X_j}{\sqrt{2}} \frac{1-iX_L}{\sqrt{2}} \mathsf{d}_{L-1}^\dagger \cdots \mathsf{d}_2^\dagger \mathsf{d}_1^\dagger, \end{aligned} \tag{C.16}$$

which is related to the operator in (C.13) of [13] as

$$\mathsf{D} = \sqrt{2} D^\dagger. \tag{C.17}$$

**The operator $\mathsf{D}_\eta$**

To obtain the $\mathbb{Z}_2$-twisted Ising Hamiltonian (12), we project the extended Hamiltonian $\widetilde{H}$ onto $Z_{L+1} = -1$, or equivalently onto $\widetilde{Z}(-1)^F = -1$. The $\mathbb{Z}_2$ symmetry of the $\mathbb{Z}_2$-twisted Ising model is given by $\eta = \prod_{j=1}^{L} X_j = \prod_{j=1}^{L} \sigma_j^x = (-1)^F = -\tilde{Z}$.

We now relate the symmetry operators $\mathsf{D}_\eta$ in the $\mathbb{Z}_2$ twisted Ising model to the Majorana lattice translations operators. We first write the translation symmetry operator

$$\widetilde{T} = \begin{pmatrix} T_- & 0 \\ 0 & T_+ \end{pmatrix} = \frac{1+\widetilde{Z}}{2} T_- + \frac{1-\widetilde{Z}}{2} T_+, \tag{C.18}$$

in terms of the Pauli operators. Using (C.11), which implies

$$T_+ = T_-(e^{\frac{2\pi i}{8}}X_{L+1}Z_L) = (e^{\frac{-2\pi i}{8}}X_{L+1}Z_1\eta)T_-,\qquad(C.19)$$

and (C.9), we find

$$\widetilde{T} = \frac{1+Z_{L+1}\eta}{2}T_- + \frac{1-Z_{L+1}\eta}{2}(e^{\frac{-2\pi i}{8}}X_{L+1}Z_1\eta)T_-.\qquad(C.20)$$

We now relate $\widetilde{T}$ to the operator $\mathsf{D}_{1\oplus\eta}$. Note that both of these operators commute with the extended Hamiltonian $\widetilde{H}$.

The crucial relations are

$$\begin{aligned}(T_-)^\dagger &= e^{2\pi i\frac{\eta-1}{8}}\frac{\mathsf{D}+\mathsf{D}_\eta}{\sqrt{2}} = \frac{\mathsf{D}-i\mathsf{D}_\eta}{\sqrt{2}},\\[4pt](T_+)^\dagger &= e^{2\pi i\frac{\eta}{8}}\frac{\mathsf{D}_{\eta\to1}+\mathsf{D}_{1\to\eta}}{\sqrt{2}}X_{L+1}.\end{aligned}\qquad(C.21)$$

(See Appendix F.3 for the definition of $\mathsf{D}_{\eta\to1}$ and $\mathsf{D}_{1\to\eta}$.) This relations can be verified using equations (57), (69), and (F.26). Using (C.21) we find[50]

$$\mathsf{D}_{1\oplus\eta} = e^{\frac{2\pi i}{8}(1-\eta)}\begin{pmatrix} T_- & 0 \\ 0 & e^{\frac{2\pi i}{8}}T_+ \end{pmatrix}^\dagger.\qquad(C.22)$$

This leads to the expressions of $\mathsf{D}$ and $\mathsf{D}_\eta$ in terms of the Pauli operators:

$$\mathsf{D} = \frac{1+\eta}{\sqrt{2}}(T_-)^\dagger,\quad\text{and}\quad \mathsf{D}_\eta = i\frac{1-\eta}{\sqrt{2}}(T_-)^\dagger,\qquad(C.23)$$

where $T_-$ is given in (C.11). It is important that $\mathsf{D}$ and $\mathsf{D}_\eta$ act in the $2^L$-dimensional Hilbert space of the Ising model and the $\mathbb{Z}_2$-twisted Ising model, rather than in the larger Hilbert space $\widetilde{\mathcal{H}}$.

**The operator $\mathsf{D}_{\mathcal{D}}$**

We now discuss the operator $\mathsf{D}_{\mathcal{D}}$ in the fermionic theory. Since in the fermionic theory, $\mathsf{D}$ corresponds to translation, the defect $\mathcal{D}$ corresponds to removing a site from the system. Therefore, the bosonic system with a $\mathcal{D}$ defect corresponds to the Majorana chains with an odd number of sites [13].

We use the same Hilbert space $\widetilde{\mathcal{H}}$ and study the following Hamiltonian

$$\begin{aligned}\widetilde{H}_{\text{odd}} &= -i\widetilde{Z}\chi_2\chi_{2L} + i\sum_{\ell=2}^{2L-1}\chi_{\ell+1}\chi_\ell = \begin{pmatrix} H_{\mathsf{G}} & 0 \\ 0 & H \end{pmatrix}\\[4pt]&= -\sum_{j=2}^{L}(\sigma_{j-1}^z\sigma_j^z + \sigma_j^x) - \widetilde{Z}(-1)^F\sigma_L^z\sigma_1^y.\end{aligned}\qquad(C.24)$$

Note that the above Hamiltonian does not involve $\chi_1$ and $\widetilde{Z} = \begin{pmatrix} 1 & 0 \\ 0 & -1 \end{pmatrix}$ commutes with the Majorana variables. Here $(-1)^F = \prod_j\sigma_j^x$ as before.

---

[50]Note that the $2\times2$ matrix presentation used here corresponds to diagonalizing the operator $\widetilde{Z} = Z_{L+1}\eta$. This is different from the $2\times2$ matrix presentation used in (F.26), which corresponds to diagonalizing the operator $Z_{L+1}$.

The lattice translation symmetry of this Hamiltonian can be taken to be

$$\widetilde{T}_{\text{odd}} = \begin{pmatrix} T_{\mathsf{G}} & 0 \\ 0 & T \end{pmatrix}, \tag{C.25}$$

where

$$
\begin{aligned}
T_{\mathsf{G}} &= \frac{e^{2\pi i \frac{2L-1}{16}}}{2^{L-1}} (1 + \chi_1 \chi_2)(1 + \chi_2 \chi_3) \cdots (1 + \chi_{2L-1} \chi_{2L}), \\
T &= \frac{e^{-2\pi i \frac{2L-1}{16}}}{2^{L-1}} (1 - \chi_1 \chi_2)(1 - \chi_2 \chi_3) \cdots (1 - \chi_{2L-1} \chi_{2L}).
\end{aligned} \tag{C.26}
$$

The overall phases are chosen so that $T^{2L-1} = e^{-\frac{2\pi i}{16}}$ and $(T_{\mathsf{G}})^{2L-1} = e^{\frac{2\pi i}{16}}$.[51] The fermion parity symmetry $(-1)^F$ is still defined by (C.2). The Majorana translation operators act on the fermions $\chi_2, \ldots, \chi_{2L}$ as

$$
\begin{aligned}
T \chi_\ell T^{-1} &= \begin{cases} \chi_{\ell+1}, & \ell = 2, \cdots, 2L-1, \\ \chi_2, & \ell = 2L, \end{cases} \\
T_{\mathsf{G}} \chi_\ell (T_{\mathsf{G}})^{-1} &= \begin{cases} -\chi_{\ell+1}, & \ell = 2, \cdots, 2L-1, \\ \chi_2, & \ell = 2L. \end{cases}
\end{aligned} \tag{C.27}
$$

To find the relation with the Ising chain, we use the same Jordan-Wigner transformation as in (C.6), but instead of (C.9) we define

$$
\begin{aligned}
X_1 &= \sigma_1^y (-1)^F \widetilde{X} \widetilde{Z}, & X_j &= \sigma_j^x, & X_{L+1} &= \sigma_1^z, \\
Z_1 &= \sigma_1^z \widetilde{X}, & Z_j &= \sigma_j^z \widetilde{X}, & Z_{L+1} &= \widetilde{Z}(-1)^F,
\end{aligned} \tag{C.28}
$$

for $j = 2, \cdots, L$. Using these new bosonic variables we find

$$\widetilde{H}_{\text{odd}} = -\sum_{j=2}^{L} (Z_{j-1} Z_j + X_j) - Z_L X_1, \tag{C.29}$$

and

$$
\begin{aligned}
T_{\mathsf{G}} &= \frac{e^{2\pi i \frac{2L-1}{16}}}{2^{L-1}} (1 - i Z_1 Z_2) \cdots (1 - i Z_{L-1} Z_L)(1 - i X_L), \\
T &= \frac{e^{-2\pi i \frac{2L-1}{16}}}{2^{L-1}} (1 + i Z_1 Z_2) \cdots (1 + i Z_{L-1} Z_L)(1 + i X_L).
\end{aligned} \tag{C.30}
$$

We see that the Hamiltonian $\widetilde{H}_{\text{odd}}$ becomes the duality-twisted Hamiltonian (22). The $\mathbb{Z}_2$ symmetry of the duality-twisted Ising chain is given by

$$\eta_{\mathcal{D}} = Z_1 \prod_{j=1}^{L} X_j = -i \widetilde{Z}. \tag{C.31}$$

Using the crucial relations

$$
\begin{aligned}
T^\dagger &= e^{-\frac{2\pi i}{16}} \frac{1 + i Z_1 Z_L}{\sqrt{2}} \frac{1 - Z_1 Z_L \eta_{\mathcal{D}}}{\sqrt{2}} \mathsf{D}_{\mathcal{D}}, \\
T_{\mathsf{G}} &= T e^{\frac{2\pi i}{8}} Z_1 Z_L,
\end{aligned} \tag{C.32}
$$

---

[51]These phases are chosen to match with the continuum symmetry operators. In particular, as stated in [13], we find $T = (-1)^{F_L} e^{\frac{2\pi i P}{2L-1}}$ on the low-lying states – here $P$ is the continuum momentum operator and $(-1)^{F_L}$ is the chiral fermion parity symmetry.

we find the relation between $D_{\mathcal{D}}$ and the Majorana translations as

$$
\begin{aligned}
D_{\mathcal{D}} &= e^{\frac{2\pi}{16}\eta_{\mathcal{D}}} \left( \frac{1+i\eta_{\mathcal{D}}}{2} T_{\mathsf{G}} + \frac{1-i\eta_{\mathcal{D}}}{2} T \right)^{\dagger} \\
&= e^{\frac{2\pi}{16}\eta_{\mathcal{D}}} \left( \begin{array}{cc} T_{\mathsf{G}} & 0 \\ 0 & T \end{array} \right)^{\dagger}.
\end{aligned}
\tag{C.33}
$$

Taking the square of this equation we find $D_{\mathcal{D}}^2 = e^{\frac{2\pi}{8}\eta_{\mathcal{D}}}(\widetilde{T}_{\text{odd}})^{-2}$. Comparing this with (87), we conclude that

$$
\widetilde{T}_{\text{odd}}^2 = T_{\mathcal{D}}.
\tag{C.34}
$$

# D  Sequential quantum circuit and the non-invertible symmetry

Here we relate our non-invertible operator $D$ to the sequential quantum circuit $\mathcal{U}$ of [25], which was based on earlier works in [23, 24]. We work on a closed periodic chain with $L$ sites, with $j = 1, 2, \cdots, L$ and $j \sim j + L$. The sequential quantum circuit is defined as

$$
\mathcal{U} = \frac{1-iZ_1 Z_L}{\sqrt{2}} \frac{1-iX_L}{\sqrt{2}} \frac{1-iZ_L Z_{L-1}}{\sqrt{2}} \cdots \frac{1-iX_3}{\sqrt{2}} \frac{1-iZ_3 Z_2}{\sqrt{2}} \frac{1-iX_2}{\sqrt{2}} \frac{1-iZ_2 Z_1}{\sqrt{2}}.
\tag{D.1}
$$

From the Majorana presentation of the non-invertible operator (C.13), it is straightforward to show that

$$
D = e^{\frac{2\pi i L}{8}} \frac{1+\eta}{\sqrt{2}} \mathcal{U}.
\tag{D.2}
$$

Let us compare these two operators. The non-invertible operator $D$ implements the Kramers-Wannier transformation at every site (63):

$$
DX_j = Z_{j-1}Z_j D, \quad DZ_{j-1}Z_j = X_{j-1}D, \quad \forall j.
\tag{D.3}
$$

In contrast, $\mathcal{U}$ acts on the $\mathbb{Z}_2$-even local operators in the same way almost everywhere except for one site/link:

$$
\begin{aligned}
\mathcal{U}X_1\mathcal{U}^{-1} &= \left( \prod_{j=1}^{L} X_j \right) Z_L Z_1, \\
\mathcal{U}X_j\mathcal{U}^{-1} &= Z_{j-1}Z_j, \quad j \neq 1, \\
\mathcal{U}Z_{j-1}Z_j\mathcal{U}^{-1} &= X_{j-1}, \quad j \neq 2, \\
\mathcal{U}Z_1 Z_2\mathcal{U}^{-1} &= X_2 X_3 \cdots X_L.
\end{aligned}
\tag{D.4}
$$

The non-invertible operator $D$ is translationally invariant (i.e., $TD = DT$) and commutes with the critical Ising Hamiltonian (10) (i.e., $HD = DH$), but it is non-invertible. In contrast, the sequential quantum circuit $\mathcal{U}$ is unitary (and in particular invertible), but it is not translationally invariant and does not commute with the Hamiltonian. Finally, both $D$ and $\mathcal{U}$ swap the product state $|++...+\rangle$ with the GHZ state.

# E  Translation symmetry and its defects

As we discussed in Section 1.2, adding translation $T$ to the symmetry group cannot be incorporated simply in terms of a fusion category. In particular, while defects of an internal symmetry correspond to localized changes in the Hamiltonian without changing the Hilbert space, this

is not true for translation defects. Instead, we can think of translation defects as changing the Hilbert space; $\mathcal{T}^+$ adds a site and $\mathcal{T}^-$ removes a site.

In this section we will examine such defects and their properties. In Section E.1, we will construct the defects. In Section E.2 we will see how they are related to the translation operator. And in Section E.3, we will study their fusion.

## E.1 Translation defect

Let us start with the translation defect $\mathcal{T}^-$. It corresponds to removing a site. Removing site number $L$, we consider the defect Hamiltonian and Hilbert space

$$H_{\mathcal{T}^-}^L = -(Z_{L-1}Z_1 + X_1) - \sum_{j=2}^{L-1}(Z_{j-1}Z_j + X_j), \qquad \mathcal{H}_{\mathcal{T}^-}^L = \mathcal{H}_1 \otimes \mathcal{H}_2 \otimes \cdots \otimes \mathcal{H}_{L-1}. \qquad \text{(E.1)}$$

Note that this Hamiltonian is the standard one for $L-1$ sites.[52] The superscript in the Hamiltonian (E.1) denotes the fact that we removed site number $L$ and we interpret it to mean that we have the original problem with $L$ sites with the defect $\mathcal{T}^-$ at position $L$. Diagrammatically, we represent it as

$$\cdots \underset{L-2}{\bullet} \ \underset{L-1}{\bullet} \ \Big| \ \underset{L}{\bullet} \ \underset{1}{\bullet} \ \underset{2}{\bullet} \cdots \ = \ \cdots \underset{L-2}{\bullet} \ \underset{L-1}{\bullet} \ \underset{1}{\bullet} \ \underset{2}{\bullet} \cdots \ . \qquad \text{(E.2)}$$
$$\mathcal{T}^-$$

As it stands, the system described by (E.1) is manifestly translation invariant. Following [53], we would like to present it in a language similar to the one we used for defects of internal symmetries.

In order to do that, we go back to the original Hilbert space with $L$ sites $\mathcal{H} = \mathcal{H}_1 \otimes \mathcal{H}_2 \otimes \cdots \otimes \mathcal{H}_L$ and consider a subspace $\mathcal{H}_{\mathcal{T}^-}^J$ where site $J$ is missing. Then, translation in the original problem $T$ acts in the large Hilbert space $\mathcal{H}$ by mapping $\mathcal{H}_j \to \mathcal{H}_{j+1}$ (with $j \sim j+L$). This action is not meaningful in the smaller Hilbert space $\mathcal{H}_{\mathcal{T}^-}^J$. Therefore, as with all our defects, we should correct it by adding a movement operator.

In the presence of the defect, the original translation operator $T$ is corrected to be

$$T_{\mathcal{T}^-} = \mathsf{S}_{J,J+1}T, \qquad \text{(E.3)}$$

which acts on the original Hilbert space $\mathcal{H}$ and commutes with $H_{\mathcal{T}^-}^J \otimes \mathbb{1}_J$. Here $\mathsf{S}_{J,J+1}$ is the swap operator acting as (A.4).

Motivated by (E.3), we would like to interpret $\mathsf{S}_{J,J+1}$ as a movement operator. However, it acts in the large Hilbert space $\mathcal{H}$. Therefore, we define the movement operator as the restriction of the swap operator $\mathsf{S}_{j+1,j}$ to the domain $\mathcal{H}_{\mathcal{T}^-}^j$ and codomain $\mathcal{H}_{\mathcal{T}^-}^{j+1}$. Explicitly,

$$U_{\mathcal{T}^-}^{J,J+1} = \left(U_{\mathcal{T}^-}^{J+1,J}\right)^{-1} = \left(|0\rangle_{J+1} \otimes \langle 0|_J + |1\rangle_{J+1} \otimes \langle 1|_J\right) \bigotimes_{j \neq J,J+1} \mathbb{1}_j,$$
$$U_{\mathcal{T}^-}^{J+1,J} : \mathcal{H}_{\mathcal{T}^-}^J \to \mathcal{H}_{\mathcal{T}^-}^{J+1}. \qquad \text{(E.4)}$$

It acts as

$$U_{\mathcal{T}^-}^{J+1,J} : \quad \begin{array}{l} X_{J+1} \mapsto X_J, \\ Z_{J+1} \mapsto Z_J. \end{array} \qquad \text{(E.5)}$$

---

[52]Removing the degrees of freedom at site $L$ can be implemented energetically. We can keep the Hilbert space unchanged and instead add the term $Z_L$ with a very large negative coefficient to the defect Hamiltonian (E.1) to freeze the spin at site $L$ to $|0\rangle_L$ and effectively remove it from the system.

From this equation, we can easily check that

$$H^{J+1}_{\mathcal{T}^-} = U^{J+1,J}_{\mathcal{T}^-} H^J_{\mathcal{T}^-} U^{J,J+1}_{\mathcal{T}^-}.$$ (E.6)

The defect $\mathcal{T}^+$ corresponds to adding a site. We construct it by adding a qubit $\mathcal{H}_{(L,1)} = \mathbb{C}^2$ on link $(L,1)$, such that the defect Hilbert space is $\mathcal{H}^{(L,1)}_{\mathcal{T}^+} = \mathcal{H}_{(L,1)} \otimes \mathcal{H}_1 \otimes \cdots \otimes \mathcal{H}_L$. The defect Hamiltonian is

$$H^{(L,1)}_{\mathcal{T}^+} = -(Z_L Z_{(L,1)} + X_{(L,1)}) - (Z_{(L,1)} Z_1 + X_1) - \sum_{j=2}^{L}(Z_{j-1} Z_j + X_j),$$ (E.7)

where $X_{(L,1)}$ and $Z_{(L,1)}$ acts on the added qubit $\mathcal{H}_{(L,1)}$ on link $(L,1)$. Diagrammatically, we write

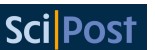 (E.8)

This defect is topological since the unitary operator[53]

$$U^J_{\mathcal{T}^+}: \quad \begin{aligned} X_J &\mapsto X_{(J,J+1)}, \quad X_{(J-1,J)} \mapsto X_J, \\ Z_J &\mapsto Z_{(J,J+1)}, \quad Z_{(J-1,J)} \mapsto Z_J, \end{aligned}$$

$$U^J_{\mathcal{T}^+}: \mathcal{H}^{(J-1,J)}_{\mathcal{T}^+} \to \mathcal{H}^{(J,J+1)}_{\mathcal{T}^+},$$ (E.9)

moves the defect by conjugation:

$$H^{(J,J+1)}_{\mathcal{T}^+} = U^J_{\mathcal{T}^+} H^{(J-1,J)}_{\mathcal{T}^+} \left(U^J_{\mathcal{T}^+}\right)^{-1}.$$ (E.10)

## E.2  Translation operator

To relate the topological defects $\mathcal{T}^\pm$ to the translation symmetry operators, we note that moving the defects through a region in space acts on that region as lattice translation. Specifically, consider the unitary operator

$$U^{J_2,J_2-1}_{\mathcal{T}^-} U^{J_2-1,J_2-2}_{\mathcal{T}^-} \cdots U^{J_1+2,J_1+1}_{\mathcal{T}^-} U^{J_1+1,J_1}_{\mathcal{T}^-}: \; H^{J_1}_{\mathcal{T}^-} \mapsto H^{J_2}_{\mathcal{T}^-},$$ (E.11)

which moves the defect $\mathcal{T}^-$ from site $J_1$ to site $J_2$ for $J_1 < J_2$. This unitary operator acts as $T^{-1}$ on any local operator $O_j$ that is supported between sites $J_1$ and $J_2$:

$$U^{J_2,J_2-1}_{\mathcal{T}^-} U^{J_2-1,J_2-2}_{\mathcal{T}^-} \cdots U^{J_1+2,J_1+1}_{\mathcal{T}^-} U^{J_1+1,J_1}_{\mathcal{T}^-}: \; O_j \mapsto O_{j-1}, \quad \text{for} \quad J_1 < j < J_2.$$ (E.12)

Moving the defect $\mathcal{T}^+$ from link $(J_1-1,J_1)$ to $(J_2,J_2+1)$ is implemented by

$$U^{J_2}_{\mathcal{T}^+} \cdots U^{J_1+1}_{\mathcal{T}^+} U^{J_1}_{\mathcal{T}^+}: \; O_j \mapsto O_{j+1}, \quad \text{for} \quad J_1 < j < J_2,$$ (E.13)

which indeed acts as the lattice translation symmetry $T$.

To establish this connection further, we will now use the defect $\mathcal{T}^-$ to construct the lattice translation operator $T^{-1}$. As in section 2.3.1, we construct the translation symmetry operator $T^{-1}$ by first creating a pair of defects $\mathcal{T}^+$ and $\mathcal{T}^-$, moving one of them around the chain, and then fusing them. These moves are implemented by unitary operators. We will show that the product of these unitary operators is the lattice translation operator.

---

[53]The unitary operator $U^J_{\mathcal{T}^+}$ is the restriction of $\mathsf{S}_{J,(J-1,J)}\mathsf{S}_{(J,J+1),J}$ to $\mathcal{H}^{(J-1,J)}_{\mathcal{T}^+}$.

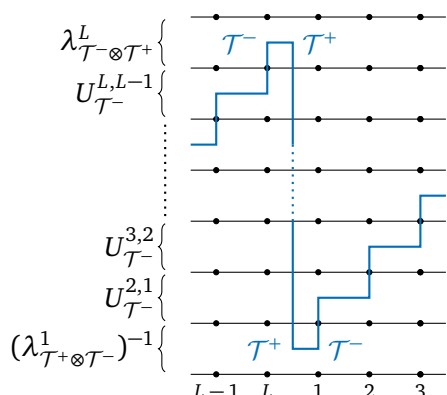

Figure 5: Construction of the lattice translation symmetry operator $T^{-1}$ from the topological defect $\mathcal{T}^-$. The diagram represent $T^{-1}$ as a sequence of unitary operators implementing pair creation of $\mathcal{T}^\pm$ defects, moving $\mathcal{T}^-$ around the chain and finally the fusion of $\mathcal{T}^-$ with $\mathcal{T}^+$.

Explicitly, creating a pair of defects is implemented by the splitting operator (the inverse of the fusion operator)

$$(\lambda^1_{\mathcal{T}^+\otimes\mathcal{T}^-})^{-1} = |0\rangle_{(L,1)}\langle 0|_1 + |1\rangle_{(L,1)}\langle 1|_1 = \quad , \tag{E.14}$$

where $(\lambda^1_{\mathcal{T}^+\otimes\mathcal{T}^-})^{-1}H\lambda^1_{\mathcal{T}^+\otimes\mathcal{T}^-} = H^{(L,1);1}_{\mathcal{T}^+;\mathcal{T}^-}$. Note that this operator does not change the size of the Hilbert space. In fact, it is even unitary. The sequence of unitary operators $U^{L,L-1}_{\mathcal{T}^-}\cdots U^{2,1}_{\mathcal{T}^-}$ moves the $\mathcal{T}^-$ defect from site 1 to site $L$. Finally, we fuse the two defects using the fusion operator

$$\lambda^L_{\mathcal{T}^-\otimes\mathcal{T}^+} = |0\rangle_L\langle 0|_{(L,1)} + |1\rangle_L\langle 1|_{(L,1)} = \quad , \tag{E.15}$$

where $\lambda^L_{\mathcal{T}^-\otimes\mathcal{T}^+}(H^{L;(L,1)}_{\mathcal{T}^-;\mathcal{T}^+})(\lambda^L_{\mathcal{T}^-\otimes\mathcal{T}^+})^{-1} = H$.

The product of all these unitary operators is indeed the lattice translation symmetry

$$\begin{aligned} T^{-1} &= \lambda^L_{\mathcal{T}^-\otimes\mathcal{T}^+}\, U^{L,L-1}_{\mathcal{T}^-}\cdots U^{3,2}_{\mathcal{T}^-}U^{2,1}_{\mathcal{T}^-}(\lambda^1_{\mathcal{T}^+\otimes\mathcal{T}^-})^{-1} \\ &= |0\rangle_L\left(U^{L,L-1}_{\mathcal{T}^-}\cdots U^{2,1}_{\mathcal{T}^-}\right)\langle 0|_1 + |1\rangle_L\left(U^{L,L-1}_{\mathcal{T}^-}\cdots U^{2,1}_{\mathcal{T}^-}\right)\langle 1|_1, \end{aligned} \tag{E.16}$$

which is similar to equation (49). See Figure 5 for a diagrammatic expression.

Finally, noting that

$$U^{j+1,j}_{\mathcal{T}^-} = \begin{pmatrix} \langle 0|_{j+1} & \langle 1|_{j+1} \end{pmatrix}\begin{pmatrix} |0\rangle_j \\ |1\rangle_j \end{pmatrix} = |0\rangle_j\langle 0|_{j+1} + |1\rangle_j\langle 1|_{j+1}, \tag{E.17}$$

we can rewrite the second line of (E.16) as the following MPO

$$T^{-1} = \mathrm{Tr}\left(\mathbb{U}^L_{\mathcal{T}^-}\mathbb{U}^{L-1}_{\mathcal{T}^-}\cdots\mathbb{U}^1_{\mathcal{T}^-}\right), \quad \text{where} \quad \mathbb{U}^j_{\mathcal{T}^-} = \begin{pmatrix} |0\rangle\langle 0|_j & |0\rangle\langle 1|_j \\ |1\rangle\langle 0|_j & |1\rangle\langle 1|_j \end{pmatrix}. \tag{E.18}$$

### E.3 Fusion involving the translation defects

Here we elaborate on some of the fusion rules involving the lattice translation defects.

We start with the fusion rule $\mathcal{T}^-\eta = \eta\otimes\mathcal{T}^- = \mathcal{T}^-\otimes\eta$ in Section 2.2.2. Consider the defect Hamiltonian with an $\eta$ defect at link $(L-1,L)$ and a $\mathcal{T}^-$ defect at site 1:

$$H^{(L-1,L);1}_{\eta;\mathcal{T}^-} = -(-Z_{L-1}Z_L + X_L) - (Z_L Z_2 + X_2) - \sum_{j=3}^{L-1}(Z_{j-1}Z_j + X_j). \tag{E.19}$$

Note that the Hilbert space is $\mathcal{H}_2 \otimes \mathcal{H}_3 \otimes \cdots \otimes \mathcal{H}_L$ with site 1 removed because of $\mathcal{T}^-$.

Next, we fuse the $\eta$ defect with the $\mathcal{T}^-$ defect using the fusion operator $\lambda^L_{\eta\otimes\mathcal{T}^-} = X_L$ to obtain the Hamiltonian for the composite defect $\mathcal{T}^-\eta$ in (36):

$$H^1_{\mathcal{T}^-\eta} = \lambda^L_{\eta\otimes\mathcal{T}^-} H^{(L-1,L);1}_{\eta;\mathcal{T}^-}(\lambda^L_{\eta\otimes\mathcal{T}^-})^{-1} = -(-Z_L Z_2 + X_2) - \sum_{j=3}^{L}(Z_{j-1}Z_j + X_j). \tag{E.20}$$

We interpret this equation as the fusion rule $\eta\mathcal{T}^- = \eta \otimes \mathcal{T}^-$, which can be diagrammatically represented as

$$\lambda^L_{\eta\otimes\mathcal{T}^-} = X_L \quad = \qquad \text{.} \tag{E.21}$$

The other fusion rule $\mathcal{T}^-\eta = \mathcal{T}^-\otimes\eta$ can be implemented similarly by another fusion operator.

Next, we discuss the dual $\mathcal{D}^*$ of the non-invertible defect $\mathcal{D}$. The dual defect $\mathcal{D}^*$ is obtained by starting with the Hamiltonian with a $\mathcal{D}$ defect at the link $(L-1,L)$ and a $\mathcal{T}^+$ defect at the link $(L,1)$:

$$H^{(L-1,L);(L,1)}_{\mathcal{D};\mathcal{T}^+} = -Z_{L-1}X_L - (Z_L Z_{(L,1)} + X_{(L,1)}) - (Z_{(L,1)}Z_1 + X_1) - \sum_{j=2}^{L-1}(Z_{j-1}Z_j + X_j), \tag{E.22}$$

which acts in the Hilbert space $\mathcal{H}_{(L,1)} \otimes \mathcal{H}_1 \otimes \cdots \otimes \mathcal{H}_L$ with $X_{(L,1)}, Z_{(L,1)}$ acting on the added qubit from $\mathcal{T}^+$. To fuse these two defects, we apply the fusion operator $\lambda^L_{\mathcal{D}\otimes\mathcal{T}^+} = \mathsf{CZ}_{L,(L,1)}\mathsf{H}_L$:

$$\begin{aligned}
H^{(L,1)}_{\mathcal{D}^*} &= \lambda^L_{\mathcal{D}\otimes\mathcal{T}^+} H^{(L-1,L);(L,1)}_{\mathcal{D};\mathcal{T}^+}(\lambda^1_{\mathcal{D}\otimes\mathcal{T}^+})^{-1} \\
&= -(Z_{L-1}Z_L + X_L) - Z_L X_{(L,1)} - (Z_{(L,1)}Z_1 + X_1) - \sum_{j=2}^{L-1}(Z_{j-1}Z_j + X_j).
\end{aligned} \tag{E.23}$$

We interpret this unitary transformation as the fusion rule $\mathcal{D}^* = \mathcal{D} \otimes \mathcal{T}^+$, which can be diagrammatically represented as

$$\lambda^L_{\mathcal{D}\otimes\mathcal{T}^+} = \mathsf{CZ}_{L,(L,1)}\mathsf{H}_L \quad = \qquad \text{.} \tag{E.24}$$

The fusion rule $\mathcal{D}^* = \mathcal{T}^+ \otimes \mathcal{D}$ can be implemented similarly by another fusion operator.

Finally, we demonstrate the fusion rule $\mathcal{D} \otimes \mathcal{D}^* = 1 \oplus \eta$. We start with a $\mathcal{D}$ defect at the link $(L-1, L)$ and the dual defect $\mathcal{D}^*$ at the link $(L, 1)$:

$$H_{\mathcal{D};\mathcal{D}^*}^{(L-1,L);(L,1)} = -Z_{L-1}X_L - Z_L X_{(L,1)} - (Z_{(L,1)}Z_1 + X_1) - \sum_{j=2}^{L-1}(Z_{j-1}Z_j + X_j), \qquad \text{(E.25)}$$

which acts in the Hilbert space $\mathcal{H}_{(L,1)} \otimes \mathcal{H}_1 \otimes \cdots \mathcal{H}_L$. The fusion operator is $\lambda_{\mathcal{D}\otimes\mathcal{D}^*}^L = \mathsf{CNOT}_{L,(L,1)}\mathsf{H}_L$:

$$
\begin{aligned}
H_{1\oplus\eta}^{(L,1)} &= \lambda_{\mathcal{D}\otimes\mathcal{D}^*}^L H_{\mathcal{D};\mathcal{D}^*}^{(L-1,L);(L,1)} \left(\lambda_{\mathcal{D}\otimes\mathcal{D}^*}^L\right)^{-1} \\
&= -Z_{(L,1)}Z_L Z_1 - X_1 - \sum_{j=2}^{L}(Z_{j-1}Z_j + X_j),
\end{aligned}
\qquad \text{(E.26)}
$$

which corresponds to inserting the non-simple defect $1 \oplus \eta$ on the link $(L, 1)$. We diagrammatically represent this fusion as

$$\lambda_{\mathcal{D}\otimes\mathcal{D}^*}^L = \mathsf{CNOT}_{L,(L,1)}\mathsf{H}_L \quad = \qquad\qquad\qquad . \qquad\qquad \text{(E.27)}$$

The fusion $\mathcal{D}^* \otimes \mathcal{D} = 1 \oplus \eta$ can be implemented similarly using another fusion operator.

# F  More on the operator algebra

In this appendix, we calculate the algebra between various symmetry operators using their MPO presentations.

## F.1  Preliminaries

### Matrix product operators (MPOs)

To begin, let us summarize our conventions for MPOs.

A Matrix Product Operator [19, 20] is constructed out of a tensor, such as $\mathbb{U}_{\mathcal{A}}^j$, which we represent as a $d \times d$ operator-valued matrix. More precisely, $(\mathbb{U}_{\mathcal{A}}^j)_{aa'}$ is an operator acting on site $j$ and the indices $a, a'$ are called the virtual, or auxiliary, indices. The size of this matrix, $d$, is called the bond dimension of the MPO. We mostly consider MPOs with bond dimension $d = 2$, therefore we take $a, a' = 0, 1$. The product of two MPOs have bond dimension $d = 4$, and we represent them by $(\mathbb{U}_{\mathcal{A}\times\mathcal{B}}^j)_{ab,a'b'}$ where $a, b, a', b' = 0, 1$,

Some of the operators take the following form. Given a tensor $\mathbb{U}_{\mathcal{A}}^j$ (of bond dimension 2) and a choice of boundary condition $\mathbb{A}$, we construct the following MPO that acts on a chain with $L$ sites:

$$
\begin{aligned}
\mathsf{A} &= \mathrm{Tr}\left(\mathbb{A}\, \mathbb{U}_{\mathcal{A}}^L \mathbb{U}_{\mathcal{A}}^{L-1} \cdots \mathbb{U}_{\mathcal{A}}^1\right) \\
&= \sum_{a_0, a_1, \ldots, a_L = 0,1} \mathbb{A}_{a_0 a_L} \left(\mathbb{U}_{\mathcal{A}}^L\right)_{a_L a_{L-1}} \left(\mathbb{U}_{\mathcal{A}}^{L-1}\right)_{a_{L-1}a_{L-2}} \cdots \left(\mathbb{U}_{\mathcal{A}}^2\right)_{a_2 a_1} \left(\mathbb{U}_{\mathcal{A}}^1\right)_{a_1 a_0}.
\end{aligned}
\qquad \text{(F.1)}
$$

Imposing periodic boundary condition corresponds to taking $\mathbb{A} = \mathbb{1}$.

**MPO presentations of $\eta$, D, and $T^{-1}$**

The MPO presentation of the symmetry operators with periodic boundary condition are

$$\eta = X_L \cdots X_1, \quad \mathsf{D} = \mathrm{Tr}\left(\mathbb{U}_{\mathcal{D}}^L \mathbb{U}_{\mathcal{D}}^{L-1} \cdots \mathbb{U}_{\mathcal{D}}^1\right), \quad T^{-1} = \mathrm{Tr}\left(\mathbb{U}_{\mathcal{T}^-}^L \mathbb{U}_{\mathcal{T}^-}^{L-1} \cdots \mathbb{U}_{\mathcal{T}^-}^1\right), \qquad \text{(F.2)}$$

where

$$\left(\mathbb{U}_{\mathcal{D}}^j\right)_{aa'} = \left|(-1)^{a+a'}\right\rangle\!\left\langle a'\right|_j, \quad \text{and} \quad \left(\mathbb{U}_{\mathcal{T}^-}^j\right)_{aa'} = |a\rangle\!\left\langle a'\right|_j. \qquad \text{(F.3)}$$

As usual, the virtual indices are denoted by $a, b, a', b' \in \{0, 1\}$. See [16, 17, 127, 139–143] for discussions of MPOs and their relations to fusion category.

**General method**

Let us first sketch the general strategy to compute the fusion algebra. Consider the MPOs $\mathsf{A} = \mathrm{Tr}\left(\mathbb{U}_{\mathcal{A}}^L \mathbb{U}_{\mathcal{A}}^{L-1} \cdots \mathbb{U}_{\mathcal{A}}^1\right)$ and $\mathsf{B} = \mathrm{Tr}\left(\mathbb{U}_{\mathcal{B}}^L \mathbb{U}_{\mathcal{B}}^{L-1} \cdots \mathbb{U}_{\mathcal{B}}^1\right)$. Their product is given by

$$\mathsf{A} \times \mathsf{B} = \mathrm{Tr}\left(\mathbb{U}_{\mathcal{A}\times\mathcal{B}}^L \mathbb{U}_{\mathcal{A}\times\mathcal{B}}^{L-1} \cdots \mathbb{U}_{\mathcal{A}\times\mathcal{B}}^2 \mathbb{U}_{\mathcal{A}\times\mathcal{B}}^1\right), \qquad \text{(F.4)}$$

where

$$\left(\mathbb{U}_{\mathcal{A}\times\mathcal{B}}^j\right)_{ab,a'b'} = \left(\mathbb{U}_{\mathcal{A}}^j\right)_{aa'}\left(\mathbb{U}_{\mathcal{B}}^j\right)_{bb'}. \qquad \text{(F.5)}$$

To establish a fusion relation of the form $\mathsf{A} \times \mathsf{B} = \mathsf{C}_1 + \mathsf{C}_2$, we need to transform $\mathbb{U}_{\mathcal{A}\times\mathcal{B}}^j$ into a block diagonal form where the blocks are $\mathbb{U}_{\mathcal{C}_1}^j$ and $\mathbb{U}_{\mathcal{C}_2}^j$. More precisely, we need to find a similarity transformation $(\mathbb{S})_{ab,a'b'}$, acting on the auxiliary degrees of freedom, such that

$$\mathbb{U}_{\mathcal{A}\times\mathcal{B}}^j = \mathbb{S}\begin{pmatrix} \mathbb{U}_{\mathcal{C}_1}^j & 0 \\ 0 & \mathbb{U}_{\mathcal{C}_2}^j \end{pmatrix}\mathbb{S}^{-1}. \qquad \text{(F.6)}$$

Such a relation clearly implies $\mathsf{A} \times \mathsf{B} = \mathsf{C}_1 + \mathsf{C}_2$.

**Defining properties of the MPOs**

It will be useful to have a set of defining relations for the tensors $\mathbb{U}_{\mathcal{D}}^j$ and $\mathbb{U}_{\mathcal{T}^-}^j$. These relation are

$$\begin{aligned} X_j \mathbb{U}_{\mathcal{D}}^j &= \mathbb{Z}\, \mathbb{U}_{\mathcal{D}}^j\, \mathbb{Z}, & \mathbb{U}_{\mathcal{D}}^j X_j &= \mathbb{X}\, \mathbb{U}_{\mathcal{D}}^j\, \mathbb{X}, \\ Z_j \mathbb{U}_{\mathcal{D}}^j &= \mathbb{X}\, \mathbb{U}_{\mathcal{D}}^j, & \mathbb{U}_{\mathcal{D}}^j Z_j &= \mathbb{U}_{\mathcal{D}}^j\, \mathbb{Z}, \end{aligned} \qquad \text{(F.7)}$$

and

$$\begin{aligned} X_j \mathbb{U}_{\mathcal{T}^-}^j &= \mathbb{X}\, \mathbb{U}_{\mathcal{T}^-}^j, & \mathbb{U}_{\mathcal{T}^-}^j X_j &= \mathbb{U}_{\mathcal{T}^-}^j\, \mathbb{X}, \\ Z_j \mathbb{U}_{\mathcal{T}^-}^j &= \mathbb{Z}\, \mathbb{U}_{\mathcal{T}^-}^j, & \mathbb{U}_{\mathcal{T}^-}^j Z_j &= \mathbb{U}_{\mathcal{T}^-}^j\, \mathbb{Z}, \end{aligned} \qquad \text{(F.8)}$$

which determine the tensors $\mathbb{U}_{\mathcal{D}}^j$ and $\mathbb{U}_{\mathcal{T}^-}^j$ up to an overall normalization.

## F.2 $\mathsf{D}^2$

Here we find the operator version of the defect fusion relation $\mathcal{D} \otimes \mathcal{D} = \mathcal{T}^- \oplus \mathcal{T}^-\eta$. Recall the MPO presentations

$$T^{-1} = \mathrm{Tr}\left(\mathbb{U}_{\mathcal{T}^-}^L \mathbb{U}_{\mathcal{T}^-}^{L-1} \cdots \mathbb{U}_{\mathcal{T}^-}^1\right), \quad \text{and} \quad T^{-1}\eta = \mathrm{Tr}\left(\mathbb{U}_{\mathcal{T}^-\eta}^L \mathbb{U}_{\mathcal{T}^-\eta}^{L-1} \cdots \mathbb{U}_{\mathcal{T}^-\eta}^1\right), \qquad \text{(F.9)}$$

where

$$\left(\mathbb{U}_{\mathcal{T}^-}^j\right)_{aa'} = |a\rangle\!\left\langle a'\right|_j, \quad \text{and} \quad \left(\mathbb{U}_{\mathcal{T}^-\eta}^j\right)_{aa'} = |a+1\rangle\!\left\langle a'\right|_j. \qquad \text{(F.10)}$$

To compute the product of D with itself, it will be useful to use an alternative MPO presentation. We write the movement operator (54) as

$$U_{\mathcal{D}}^j = \begin{pmatrix} 1 & Z_{j+1} \end{pmatrix} \begin{pmatrix} \frac{1+Z_j}{2}\mathsf{H}_j \\ \frac{1-Z_j}{2}\mathsf{H}_j \end{pmatrix} = \frac{1+Z_j}{2}\mathsf{H}_j + Z_{j+1}\frac{1-Z_j}{2}\mathsf{H}_j \,, \tag{F.11}$$

which leads to the MPO tensor

$$\mathbb{U}'^j_{\mathcal{D}} = \begin{pmatrix} \frac{1+Z_j}{2}\mathsf{H}_j \\ \frac{1-Z_j}{2}\mathsf{H}_j \end{pmatrix} \begin{pmatrix} 1 & Z_j \end{pmatrix} = \begin{pmatrix} |0\rangle\langle+|_j & |0\rangle\langle-|_j \\ |1\rangle\langle-|_j & |1\rangle\langle+|_j \end{pmatrix}, \qquad \left(\mathbb{U}'^j_{\mathcal{D}}\right)_{aa'} = |a\rangle\langle(-1)^{a+a'}|_j \,. \tag{F.12}$$

The tensors $\mathbb{U}^j_{\mathcal{D}}$ of (F.3) and $\mathbb{U}'^j_{\mathcal{D}}$ are related by a similarity transformation that acts on the auxiliary degrees of freedom. More precisely, they are related by the Hadamard matrix; namely,

$$\mathbb{U}'^j_{\mathcal{D}} = \mathbb{H}\mathbb{U}^j_{\mathcal{D}}\mathbb{H}, \quad \text{where} \quad (\mathbb{H})_{aa'} = (-1)^{aa'}/\sqrt{2}. \tag{F.13}$$

Therefore, they are equivalent MPOs and lead to the same operator

$$\mathsf{D} = \text{Tr}\left(\mathbb{U}^L_{\mathcal{D}}\mathbb{U}^{L-1}_{\mathcal{D}}\cdots\mathbb{U}^1_{\mathcal{D}}\right) = \text{Tr}\left(\mathbb{U}'^L_{\mathcal{D}}\mathbb{U}'^{L-1}_{\mathcal{D}}\cdots\mathbb{U}'^1_{\mathcal{D}}\right).$$

Using both MPO presentations, we find

$$\mathsf{D}^2 = \text{Tr}\left(\mathbb{U}'^L_{\mathcal{D}\times\mathcal{D}}\,\mathbb{U}'^{L-1}_{\mathcal{D}\times\mathcal{D}}\cdots\mathbb{U}'^2_{\mathcal{D}\times\mathcal{D}}\,\mathbb{U}'^1_{\mathcal{D}\times\mathcal{D}}\right), \tag{F.14}$$

where

$$\left(\mathbb{U}'^j_{\mathcal{D}\times\mathcal{D}}\right)_{ab,a'b'} = \left(\mathbb{U}'^j_{\mathcal{D}}\right)_{aa'}\left(\mathbb{U}^j_{\mathcal{D}}\right)_{bb'} = \left(\delta_{a+a',b+b' \pmod 2}\right)|a\rangle\langle b'|_j \,, \tag{F.15}$$

uses the two different presentations (F.3) and (F.12). Applying the

$$(\mathbb{CNOT})_{ab,a'b'} = \delta_{b,b'}\delta_{a,a'+b' \pmod 2}$$

gate we find

$$\left(\mathbb{CNOT}\,\mathbb{U}'^j_{\mathcal{D}\times\mathcal{D}}\,\mathbb{CNOT}\right)_{ab,a'b'} = \delta_{a,a'}|b+a\rangle\langle b'|_j = \begin{pmatrix} |b\rangle\langle b'|_j & 0 \\ 0 & |b+1\rangle\langle b'|_j \end{pmatrix}_{aa'}, \tag{F.16}$$

which indeed gives

$$\mathbb{CNOT}\,\mathbb{U}'^j_{\mathcal{D}\times\mathcal{D}}\,\mathbb{CNOT} = \begin{pmatrix} \mathbb{U}^j_{\mathcal{T}^-} & 0 \\ 0 & \mathbb{U}^j_{\mathcal{T}^-\eta} \end{pmatrix}. \tag{F.17}$$

Thus we have established the fusion relation

$$\mathsf{D}^2 = T^{-1}(1+\eta). \tag{F.18}$$

Instead of using $\mathbb{U}'^j_{\mathcal{D}\times\mathcal{D}}$, we can use the original MPO presentation $\mathbb{U}^j_{\mathcal{D}\times\mathcal{D}}$ and find the similarity transformation $\mathbb{S}$ satisfying

$$\mathbb{S}\,\mathbb{U}^j_{\mathcal{D}\times\mathcal{D}}\,\mathbb{S}^{-1} = \begin{pmatrix} \mathbb{U}^j_{\mathcal{T}^-} & 0 \\ 0 & \mathbb{U}^j_{\mathcal{T}^-\eta} \end{pmatrix}. \tag{F.19}$$

This similarity transformation is given by $\mathbb{S} = \mathbb{CNOT}(\mathbb{H}\otimes\mathbb{1}) = (\mathbb{H}\otimes\mathbb{1})\mathbb{CZ}$, which follows from $\mathbb{U}'^j_{\mathcal{D}\times\mathcal{D}} = (\mathbb{H}\otimes\mathbb{1})\mathbb{U}^j_{\mathcal{D}\times\mathcal{D}}(\mathbb{H}\otimes\mathbb{1})$ and (F.17). More explicitly, we have

$$\begin{aligned} \mathbb{S}_{ab,a'b'} &= ((\mathbb{H}\otimes\mathbb{1})\mathbb{CZ})_{ab,a'b'} = \delta_{b,b'}\frac{(-1)^{(a+b)a'}}{\sqrt{2}}, \\ \mathbb{S}^{-1}_{ab,a'b'} &= (\mathbb{CZ}(\mathbb{H}\otimes\mathbb{1}))_{ab,a'b'} = \delta_{b,b'}\frac{(-1)^{a(a'+b')}}{\sqrt{2}}. \end{aligned} \tag{F.20}$$

### F.3   $D^2_{1\oplus\eta}$

Here we compute the operator product $D^2_{1\oplus\eta}$. As we will see below, this operator fusion leads to the relation between $D^\dagger$ and $D$, and similarly $D^\dagger_\eta$ and $D_\eta$. We begin with the MPO presentation of the operator $D_{1\oplus\eta} : \mathcal{H} \oplus \mathcal{H} \to \mathcal{H} \oplus \mathcal{H}$, defined in (49), and its matrix elements.

**MPO presentation of $D_{1\oplus\eta}$**

Taking various matrix elements of $D_{1\oplus\eta} H^{(L,1)}_{1\oplus\eta} = H^{(L,1)}_{1\oplus\eta} D_{1\oplus\eta}$, leads to symmetry operators

$$
\begin{aligned}
D\,H &= H\,D\,, & \text{for} && D &= \sqrt{2}\,\langle 0|\,D_{1\oplus\eta}\,|0\rangle_{L+1}\,, \\
D_\eta\,H^{(L,1)}_\eta &= H^{(L,1)}_\eta\,D_\eta\,, & \text{for} && D_\eta &= \sqrt{2}\,\langle 1|\,D_{1\oplus\eta}\,|1\rangle_{L+1}\,, \\
D_{1\to\eta}\,H &= H^{(L,1)}_\eta\,D_{1\to\eta}\,, & \text{for} && D_{1\to\eta} &= \sqrt{2}\,\langle 1|\,D_{1\oplus\eta}\,|0\rangle_{L+1}\,, \\
D_{\eta\to1}\,H^{(L,1)}_\eta &= H\,D_{\eta\to1}\,, & \text{for} && D_{\eta\to1} &= \sqrt{2}\,\langle 0|\,D_{1\oplus\eta}\,|1\rangle_{L+1}\,.
\end{aligned}
\tag{F.21}
$$

In the $2 \times 2$ matrix presentation of the Pauli operator $Z_{L+1} = Z_{(L,1)}$, we have

$$
H^{(L,1)}_{1\oplus\eta} = \begin{pmatrix} H & 0 \\ 0 & H^{(L,1)}_\eta \end{pmatrix}, \quad \text{and} \quad D_{1\oplus\eta} = \frac{1}{\sqrt{2}} \begin{pmatrix} D & D_{\eta\to1} \\ D_{1\to\eta} & D_\eta \end{pmatrix}.
\tag{F.22}
$$

We now provide an MPO presentation of the operator $D_{1\oplus\eta}$ given in (50):

$$
\begin{aligned}
D_{1\oplus\eta} &= H_{L+1}\, U^L_{\mathcal{D}} U^{L-1}_{\mathcal{D}} \cdots U^1_{\mathcal{D}}\, H_{L+1}\, U^{L+1}_{\mathcal{D}} \\
&= H_{L+1} \left( \tfrac{1+Z_{L+1}}{2} \quad \tfrac{1-Z_{L+1}}{2} \right) \mathbb{U}^L_{\mathcal{D}} \cdots \mathbb{U}^1_{\mathcal{D}}\, H_{L+1} \begin{pmatrix} H_{L+1} \\ Z_{L+1} H_{L+1} \end{pmatrix} \\
&= \mathrm{Tr}\left( \mathbb{D}^{L+1}_{1\oplus\eta}\, \mathbb{U}^L_{\mathcal{D}} \cdots \mathbb{U}^1_{\mathcal{D}} \right),
\end{aligned}
\tag{F.23}
$$

where

$$
\mathbb{D}^{L+1}_{1\oplus\eta} = \begin{pmatrix} H_{L+1} \tfrac{1+Z_{L+1}}{2} & H_{L+1} \tfrac{1-Z_{L+1}}{2} \\ H_{L+1} \tfrac{1+Z_{L+1}}{2} X_{L+1} & H_{L+1} \tfrac{1-Z_{L+1}}{2} X_{L+1} \end{pmatrix} = \begin{pmatrix} |+\rangle\langle 0|_{L+1} & |-\rangle\langle 1|_{L+1} \\ |+\rangle\langle 1|_{L+1} & |-\rangle\langle 0|_{L+1} \end{pmatrix}.
\tag{F.24}
$$

This leads to

$$
\begin{aligned}
D &= \mathrm{Tr}\left( \mathbb{1}\, \mathbb{U}^L_{\mathcal{D}} \cdots \mathbb{U}^1_{\mathcal{D}} \right), & \text{for} && \mathbb{1} &= \sqrt{2}\,\langle 0|\mathbb{D}^{L+1}_{1\oplus\eta}|0\rangle_{L+1} = \begin{pmatrix} 1 & 0 \\ 0 & 1 \end{pmatrix}, \\
D_\eta &= \mathrm{Tr}\left( \mathbb{XZ}\, \mathbb{U}^L_{\mathcal{D}} \cdots \mathbb{U}^1_{\mathcal{D}} \right), & \text{for} && \mathbb{XZ} &= \sqrt{2}\,\langle 1|\mathbb{D}^{L+1}_{1\oplus\eta}|1\rangle_{L+1} = \begin{pmatrix} 0 & -1 \\ 1 & 0 \end{pmatrix}, \\
D_{\eta\to1} &= \mathrm{Tr}\left( \mathbb{X}\, \mathbb{U}^L_{\mathcal{D}} \cdots \mathbb{U}^1_{\mathcal{D}} \right), & \text{for} && \mathbb{X} &= \sqrt{2}\,\langle 0|\mathbb{D}^{L+1}_{1\oplus\eta}|1\rangle_{L+1} = \begin{pmatrix} 0 & 1 \\ 1 & 0 \end{pmatrix}, \\
D_{1\to\eta} &= \mathrm{Tr}\left( \mathbb{Z}\, \mathbb{U}^L_{\mathcal{D}} \cdots \mathbb{U}^1_{\mathcal{D}} \right), & \text{for} && \mathbb{Z} &= \sqrt{2}\,\langle 1|\mathbb{D}^{L+1}_{1\oplus\eta}|0\rangle_{L+1} = \begin{pmatrix} 1 & 0 \\ 0 & -1 \end{pmatrix}.
\end{aligned}
\tag{F.25}
$$

Using the second line of (F.7), we conclude

$$
D_{1\oplus\eta} = \frac{1}{\sqrt{2}} \begin{pmatrix} D & D_{\eta\to1} \\ D_{1\to\eta} & D_\eta \end{pmatrix} = \frac{1}{\sqrt{2}} \begin{pmatrix} D & Z_L D \\ D Z_1 & -Z_L D Z_1 \end{pmatrix}.
\tag{F.26}
$$

**The fusion**

Using the MPO presentation of $D_{1\oplus\eta}$, we find

$$D_{1\oplus\eta}^2 = \mathrm{Tr}\Big( (\mathbb{D}^2)_{1\oplus\eta}^{L+1}\, \mathbb{U}_{\mathcal{D}\times\mathcal{D}}^L\, \mathbb{U}_{\mathcal{D}\times\mathcal{D}}^{L-1} \cdots \mathbb{U}_{\mathcal{D}\times\mathcal{D}}^2\, \mathbb{U}_{\mathcal{D}\times\mathcal{D}}^1 \Big), \tag{F.27}$$

where

$$\Big(\mathbb{D}_{1\oplus\eta}^{L+1}\Big)_{aa'} = \big|(-1)^{a'}\big\rangle\big\langle a+a'\big|_{L+1} \quad \Rightarrow \quad \Big((\mathbb{D}^2)_{1\oplus\eta}^{L+1}\Big)_{ab,a'b'} = \frac{(-1)^{(a+a')b'}}{\sqrt{2}}\big|(-1)^{a'}\big\rangle\big\langle b+b'\big|_{L+1}. \tag{F.28}$$

To simplify the operator product, we use the similarity transformation (F.20) to find

$$\begin{aligned}
\Big(\mathbb{S}\,(\mathbb{D}^2)_{1\oplus\eta}^{L+1}\,\mathbb{S}^{-1}\Big)_{ab,a'b'} &= \sum_{a'',a'''=0,1} \frac{(-1)^{(a+b)a''+(a''+a''')b'+a'''(a'+b')}}{2\sqrt{2}}\big|(-1)^{a'''}\big\rangle\big\langle b+b'\big|_{L+1} \\
&= \sum_{a''=0,1} \frac{(-1)^{a''(a+b+b')}}{2}\big|a'\big\rangle\big\langle b+b'\big|_{L+1} = \big(\delta_{a,b+b' \ (\mathrm{mod}\ 2)}\big)\big|a'\big\rangle\big\langle a\big|_{L+1} \\
&= \begin{pmatrix} \delta_{b,b'}\,|0\rangle\langle 0|_{L+1} & \delta_{b,b'}\,|1\rangle\langle 0|_{L+1} \\ \delta_{1-b,b'}\,|0\rangle\langle 1|_{L+1} & \delta_{1-b,b'}\,|1\rangle\langle 1|_{L+1} \end{pmatrix}_{aa'}.
\end{aligned} \tag{F.29}$$

Using the MPO presentation (F.27) and the relations (F.19) and (F.29), we find

$$\begin{aligned}
D_{1\oplus\eta}^2 &= |0\rangle\langle 0|_{L+1}\mathrm{Tr}\big(\mathbb{U}_{\mathcal{T}-}^L\mathbb{U}_{\mathcal{T}-}^{L-1}\cdots\mathbb{U}_{\mathcal{T}-}^1\big) + |1\rangle\langle 1|_{L+1}\mathrm{Tr}\big(\mathbb{X}\,\mathbb{U}_{\mathcal{T}-\eta}^L\mathbb{U}_{\mathcal{T}-\eta}^{L-1}\cdots\mathbb{U}_{\mathcal{T}-\eta}^1\big) \\
&= |0\rangle\langle 0|_{L+1}T^{-1} + |1\rangle\langle 1|_{L+1}T_\eta^{-1}\eta,
\end{aligned} \tag{F.30}$$

where

$$T_\eta^{-1} = T^{-1}X_1 = X_L T^{-1}. \tag{F.31}$$

In terms of the $2 \times 2$ presentation (F.26), (F.30) becomes

$$D_{1\oplus\eta}^2 = \begin{pmatrix} T^{-1} & 0 \\ 0 & T_\eta^{-1}\eta \end{pmatrix}. \tag{F.32}$$

Since $D_{1\oplus\eta}$ is a unitary operator, we find

$$\begin{aligned}
D_{1\oplus\eta}^\dagger &= \frac{1}{\sqrt{2}}\begin{pmatrix} D^\dagger & D_{1\to\eta}^\dagger \\ D_{\eta\to 1}^\dagger & D_\eta^\dagger \end{pmatrix} = \frac{1}{\sqrt{2}}\begin{pmatrix} D & D_{\eta\to 1} \\ D_{1\to\eta} & D_\eta \end{pmatrix}\begin{pmatrix} T & 0 \\ 0 & T_\eta\eta \end{pmatrix} \\
&= \frac{1}{\sqrt{2}}\begin{pmatrix} T & 0 \\ 0 & T_\eta\eta \end{pmatrix}\begin{pmatrix} D & D_{\eta\to 1} \\ D_{1\to\eta} & D_\eta \end{pmatrix}.
\end{aligned} \tag{F.33}$$

Thus we find

$$D^\dagger = DT = TD, \quad \text{and} \quad D_\eta^\dagger = -D_\eta T_\eta = -T_\eta D_\eta, \tag{F.34}$$

where above we have used the fusion relation $D_\eta\eta = \eta D_\eta = -D_\eta$.

## F.4 $D_{\mathcal{D}}^2$

We begin with the MPO presentations of the symmetry operators $\eta_{\mathcal{D}}$, $T_{\mathcal{D}}^{-1}$ and $D_{\mathcal{D}}$ that commute with the defect Hamiltonian $H_{\mathcal{D}}^{(L,1)}$ given in (22).

The $\mathcal{D}$-twisted $\mathbb{Z}_2$ symmetry operator is denoted by

$$\eta_{\mathcal{D}} = X_L X_{L-1}\cdots X_2 (Z_1 X_1). \tag{F.35}$$

It is straightforward to verify that it commutes with the defect Hamiltonian (22). Following the discussion in footnote 24, the $\mathcal{D}$-twisted lattice translation is given by $T_{\mathcal{D}}^{-1} = T^{-1} U_{\mathcal{D}}^1$. Writing the movement operator (24) as

$$U_{\mathcal{D}}^j = |0\rangle\langle+|_j + Z_{j+1}|1\rangle\langle-|_j,\tag{F.36}$$

and using the MPO presentation of the translation operator we get

$$\begin{aligned}
T_{\mathcal{D}}^{-1} &= \text{Tr}\left(\mathbb{U}_{\mathcal{T}^-}^L\mathbb{U}_{\mathcal{T}^-}^{L-1}\cdots\mathbb{U}_{\mathcal{T}^-}^3\mathbb{U}_{\mathcal{T}^-}^2\begin{pmatrix}|0\rangle\langle0|_1 & |0\rangle\langle1|_1 \\[4pt] |1\rangle\langle0|_1 & |1\rangle\langle1|_1\end{pmatrix}\right)U_{\mathcal{D}}^1 \\[8pt]
&= \text{Tr}\left(\mathbb{U}_{\mathcal{T}^-}^L\mathbb{U}_{\mathcal{T}^-}^{L-1}\cdots\mathbb{U}_{\mathcal{T}^-}^3\begin{pmatrix}|0\rangle\langle0|_2 & |0\rangle\langle1|_2 \\[4pt] |1\rangle\langle0|_2 & |1\rangle\langle1|_2\end{pmatrix}\begin{pmatrix}|0\rangle\langle+|_1 & Z_2|0\rangle\langle-|_1 \\[4pt] |1\rangle\langle+|_1 & Z_2|1\rangle\langle-|_1\end{pmatrix}\right) \\[8pt]
&= \text{Tr}\left(\mathbb{U}_{\mathcal{T}^-}^L\mathbb{U}_{\mathcal{T}^-}^{L-1}\cdots\mathbb{U}_{\mathcal{T}^-}^3\mathbb{U}_{\mathcal{T}^-}^2\begin{pmatrix}|0\rangle\langle+|_1 & |0\rangle\langle-|_1 \\[4pt] |1\rangle\langle+|_1 & -|1\rangle\langle-|_1\end{pmatrix}\right).
\end{aligned}\tag{F.37}$$

In summary we find

$$T_{\mathcal{D}}^{-1} = \text{Tr}\left(\mathbb{U}_{\mathcal{T}^-}^L\cdots\mathbb{U}_{\mathcal{T}^-}^2(\mathbb{T}^{-1})_{\mathcal{D}}^1\right),\quad\text{where}\quad\left((\mathbb{T}^{-1})_{\mathcal{D}}^1\right)_{aa'} = (-1)^{aa'}|a\rangle\langle(-1)^{a'}|_1.\tag{F.38}$$

Finally, the MPO presentation of $D_{\mathcal{D}}$, as given in equations (71) and (72), is

$$D_{\mathcal{D}} = \text{Tr}\left(\mathbb{U}_{\mathcal{D}}^L\cdots\mathbb{U}_{\mathcal{D}}^2\mathbb{D}_{\mathcal{D}}^1\right),\quad\text{where}\quad\left(\mathbb{D}_{\mathcal{D}}^1\right)_{aa'} = (-1)^{aa'}|a'\rangle\langle(-1)^{a'}|_1.\tag{F.39}$$

Using this expression we find

$$D_{\mathcal{D}}^2 = \text{Tr}\left(\mathbb{U}_{\mathcal{D}\times\mathcal{D}}^L\cdots\mathbb{U}_{\mathcal{D}\times\mathcal{D}}^2(\mathbb{D}^2)_{\mathcal{D}}^1\right),\tag{F.40}$$

where

$$\left((\mathbb{D}^2)_{\mathcal{D}}^1\right)_{ab,a'b'} = \left(\mathbb{D}_{\mathcal{D}}^1\right)_{aa'}\left(\mathbb{D}_{\mathcal{D}}^1\right)_{bb'} = \frac{(-1)^{aa'+bb'+a'b'}}{\sqrt{2}}|a'\rangle\langle(-1)^{b'}|_1.\tag{F.41}$$

Using the similarity transformation (F.20), we get

$$\begin{aligned}
\left(\mathbb{S}(\mathbb{D}^2)_{\mathcal{D}}^1\mathbb{S}^{-1}\right)_{ab,a'b'} &= \sum_{a'',a'''=0,1}\frac{(-1)^{(a+b)a''+a''a'''+bb'+a'''b'+a'''(a'+b')}}{2\sqrt{2}}|a'''\rangle\langle(-1)^{b'}|_1 \\[6pt]
&= \sum_{a'''=0,1}\left(\delta_{a''',a+b\ (\text{mod }2)}\right)\frac{(-1)^{bb'+a'''a'}}{\sqrt{2}}|a'''\rangle\langle(-1)^{b'}|_1 \\[6pt]
&= \frac{(-1)^{bb'+(a+b)a'}}{\sqrt{2}}|a+b\rangle\langle(-1)^{b'}|_1 \\[6pt]
&= \frac{1}{\sqrt{2}}(Z_1)^{a'}(X_1)^a\left((\mathbb{T}^{-1})_{\mathcal{D}}^1\right)_{bb'},
\end{aligned}\tag{F.42}$$

where in the last line we have used the expression for $(\mathbb{T}^{-1})_{\mathcal{D}}^1$ given in (F.38). Using (F.19), we further simplify the operator product to

$$\begin{aligned}
D_{\mathcal{D}}^2 &= \frac{1}{\sqrt{2}}\text{Tr}\left(\mathbb{U}_{\mathcal{T}^-}^L\cdots\mathbb{U}_{\mathcal{T}^-}^2(\mathbb{T}^{-1})_{\mathcal{D}}^1\right) + \frac{1}{\sqrt{2}}\text{Tr}\left(\mathbb{U}_{\mathcal{T}^-\eta}^L\cdots\mathbb{U}_{\mathcal{T}^-\eta}^2(Z_1X_1)(\mathbb{T}^{-1})_{\mathcal{D}}^1\right) \\[6pt]
&= \frac{1+X_L\cdots X_2(Z_1X_1)}{\sqrt{2}}\text{Tr}\left(\mathbb{U}_{\mathcal{T}^-}^L\cdots\mathbb{U}_{\mathcal{T}^-}^2(\mathbb{T}^{-1})_{\mathcal{D}}^1\right) \\[6pt]
&= \frac{1}{\sqrt{2}}(1+\eta_{\mathcal{D}})T_{\mathcal{D}}^{-1},
\end{aligned}\tag{F.43}$$

where we have used the relation $\mathbb{U}_{\mathcal{T}^-\eta}^j = X_j\mathbb{U}_{\mathcal{T}^-}^j$; see equation (F.10).

# G   Tambara-Yamagami fusion categories

In this appendix, we review some aspects of the TY($\mathbb{Z}_2, \epsilon$) fusion categories. We present the F-symbols and derive the operator algebras in the twisted Hilbert spaces. Mathematically, the operator algebras on (and more generally, between) different twisted Hilbert spaces are described by the tube algebra [124, 161, 162]. Finally, we compare these operator algebras with their lattice counterparts and comment on the ambiguity of the FS indicator on the lattice. Our discussion of the TY($\mathbb{Z}_2, \epsilon$) fusion categories follows [12, 14, 119, 162] closely.

## G.1   F-symbols of TY($\mathbb{Z}_2, \epsilon$)

The two TY fusion categories TY($\mathbb{Z}_2, \epsilon = \pm$) share the same fusion algebra

$$\mathcal{N}^2 = 1 + \eta, \qquad \mathcal{N}\eta = \eta\mathcal{N} = \mathcal{N}, \qquad \eta^2 = 1. \tag{G.1}$$

The FS indicators $\epsilon$ enter the following two F-moves:[54]

$$\tag{G.2}$$

Here the red and blue lines stand for non-invertible line $\mathcal{N}$ and the invertible $\mathbb{Z}_2$ line $\eta$, respectively. The other F-symbols are (see Table 1 of [114]):

$$\tag{G.3}$$

---

[54]For readers who are more familiar with MTCs [34, 113], the two fusion categories TY($\mathbb{Z}_2, \epsilon$) are obtained from the MTCs associated with the 2+1d $Spin(\nu)_1$ Chern-Simons theory (also known as Kitaev's 16-fold way [114]) with odd $\nu$ by forgetting the braiding structure (i.e., the complex phase in the R-symbols). The $Spin(\nu)_1$ MTCs with $\nu = 1, 7, 9, 15$ mod 16 share the same F-symbols corresponding to TY($\mathbb{Z}_2, +$), while the $\nu = 3, 5, 11, 13$ mod 16 ones give rise to TY($\mathbb{Z}_2, -$). This is correlated with the fact that for these values of $\nu$, the spinor of $Spin(\nu)$ is real and pseudoreal respectively.

## G.2 Operator algebra in the $\mathbb{Z}_2$-twisted Hilbert space

In the $\mathbb{Z}_2$-twisted Hilbert space, we define two operators as:

$$\eta_\eta = \quad = \quad , \qquad \mathcal{N}_\eta = \quad . \tag{G.4}$$

The black square represents a cylinder with the two vertical sides identified. The vertical direction stands for time, while the horizontal direction stands for space, which is a circle. The vertical blue line represents the $\mathbb{Z}_2$ twist at a fixed position on the spatial circle. For the $\mathbb{Z}_2$ operator $\eta_\eta$ in the $\mathbb{Z}_2$-twisted Hilbert space, the two configurations are identical because of the trivial F-move in the upper right corner in (G.3). This follows from the fact that this $\mathbb{Z}_2$ symmetry is free of 't Hooft anomalies [117].

Since the F-move involving just the $\mathbb{Z}_2$ line is trivial, we have $\eta_\eta^2 = 1$. Next, by applying the F-moves in (G.2), we compute $\eta_\eta \times \mathcal{N}_\eta$:

$$\eta_\eta \times \mathcal{N}_\eta = \quad = - \quad = - \quad . \tag{G.5}$$

The product $\mathcal{N}_\eta \times \eta_\eta$ can be computed similarly and we find $\eta_\eta \times \mathcal{N}_\eta = \mathcal{N}_\eta \times \eta_\eta = -\mathcal{N}_\eta$. Using the F-moves in (G.2) and (G.3), we compute $\mathcal{N}_\eta \times \mathcal{N}_\eta$ [119]:

$$\mathcal{N}_\eta \times \mathcal{N}_\eta = \quad = - \quad = - \frac{\epsilon}{\sqrt{2}} \quad + \frac{\epsilon}{\sqrt{2}} \quad$$

$$= -\epsilon \quad + \epsilon \quad . \tag{G.6}$$

We summarize the operator algebra in the $\mathbb{Z}_2$-twisted Hilbert space [119]:

$$\begin{aligned} \eta_\eta^2 &= 1, \\ \eta_\eta \times \mathcal{N}_\eta &= \mathcal{N}_\eta \times \eta_\eta = -\mathcal{N}_\eta, \\ \mathcal{N}_\eta \times \mathcal{N}_\eta &= -\epsilon(1 - \eta_\eta). \end{aligned} \tag{G.7}$$

The lattice counterpart of this algebra is in (79).

### G.3   Operator algebra in the duality-twisted Hilbert space

In the duality-twisted Hilbert space, we define two operators as

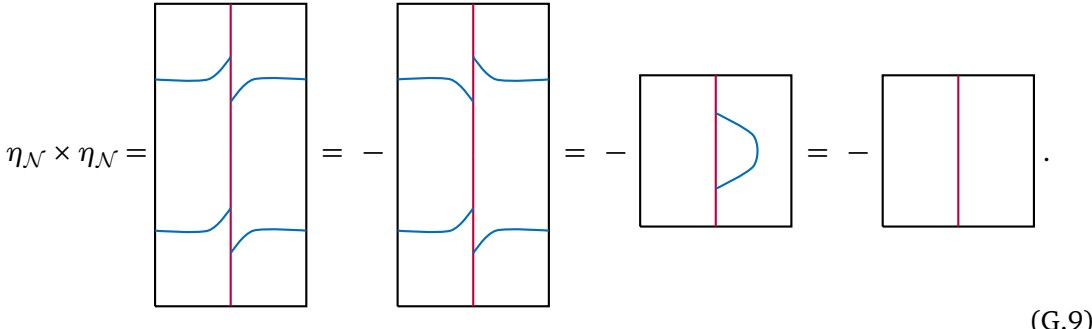

$$\eta_{\mathcal{N}} = \quad , \qquad \mathcal{N}_{\mathcal{N}} = \quad . \tag{G.8}$$

We show $\eta_{\mathcal{N}}^2 = -1$ as follows:

$$\eta_{\mathcal{N}} \times \eta_{\mathcal{N}} = \quad = - \quad = - \quad = - \quad . \tag{G.9}$$

Next, by applying the F-moves in (G.2), we compute $\mathcal{N}_{\mathcal{N}} \times \mathcal{N}_{\mathcal{N}}$:

$$\mathcal{N}_{\mathcal{N}} \times \mathcal{N}_{\mathcal{N}} = \quad = \frac{\epsilon}{\sqrt{2}} \quad + \frac{\epsilon}{\sqrt{2}} \quad \tag{G.10}$$

$$= \frac{\epsilon}{\sqrt{2}} \quad + \frac{\epsilon}{\sqrt{2}} \quad .$$

We conclude that [12, 14],

$$\begin{aligned} \eta_{\mathcal{N}}^2 &= -1, \\ \mathcal{N}_{\mathcal{N}} \times \mathcal{N}_{\mathcal{N}} &= \frac{\epsilon}{\sqrt{2}} (1 + \eta_{\mathcal{N}}). \end{aligned} \tag{G.11}$$

Note that $\eta_{\mathcal{N}} \times \mathcal{N}_{\mathcal{N}} = \mathcal{N}_{\mathcal{N}} \times \eta_{\mathcal{N}}$ gives another independent operator. The lattice counterpart of this algebra is in (87).

### G.4   Projective phases in the action of the non-invertible symmetry and parity

In Section 2.5 we found that the lattice symmetry algebra involving parity/time-reversal in the presence of a non-invertible defect is realized projectively. Here we derive the corresponding algebra for the $\mathbb{Z}_2$ operator $\eta_{\mathcal{N}}$ and parity $\mathcal{P}_{\mathcal{N}}$ in the $\mathcal{N}$-twisted Hilbert space in the continuum.

We compute $\mathcal{P}_{\mathcal{N}} \eta_{\mathcal{N}} \mathcal{P}_{\mathcal{N}}^{-1}$ as follows:

$$\mathcal{P}_{\mathcal{N}} \eta_{\mathcal{N}} \mathcal{P}_{\mathcal{N}}^{-1} = \quad\quad = - \quad\quad = -\eta_{\mathcal{N}}, \tag{G.12}$$

where in the second equality we have used the F-move in (G.3). This provides a more general derivation of the projective algebra in (103) for the Ising CFT.[55]

# H Lattice interpolation between $\mathrm{TY}(\mathbb{Z}_2, \epsilon = \pm)$

In Section 3.2, we emphasized that our lattice symmetry does not have a FS indicator, while the continuum symmetry does. In this Appendix, we demonstrate how the FS indicator of the continuum theory can depend on the choice of the lattice Hamiltonian. We will present a continuous family of lattice Hamiltonians enjoying the same lattice symmetry. The symmetry of the continuum theory of some of them is $\mathrm{TY}(\mathbb{Z}_2, +)$, while for the others, it is $\mathrm{TY}(\mathbb{Z}_2, -)$.

Our example will make use of the following fact about the two $\mathrm{TY}(\mathbb{Z}_2, \epsilon)$ categories [12]. Consider a general CFT with a fusion category symmetry $\mathrm{TY}(\mathbb{Z}_2, \epsilon) \boxtimes \mathrm{Vec}_{\mathbb{Z}_2}^\omega$, with $\omega \in H^3(\mathbb{Z}_2, U(1))$ being the nontrivial element. Here, $\mathrm{Vec}_{\mathbb{Z}_2}^\omega$ is the fusion category describing an anomalous, invertible $\mathbb{Z}_2$ symmetry, whose generator we denote by $\mathcal{C}$. In this big category, one can consider a subcategory generated by $1, \eta, \mathcal{N}' = \mathcal{N} \otimes \mathcal{C}$. One can show that these three simple lines generate a $\mathrm{TY}(\mathbb{Z}_2, -\epsilon)$ fusion category, with the opposite FS indicator compared to the one generated by $1, \eta, \mathcal{N}$. That is, the stacking with an anomalous $\mathbb{Z}_2$ line flips the sign of the FS indicator of a non-invertible line. As an example, which will be used below, consider a tensor product of the Ising fusing category $\mathrm{TY}(\mathbb{Z}_2, \epsilon = +1)$ and the fusion category of $SU(2)_1$. The tensor product includes the fusion category of $SU(2)_2$, which is $\mathrm{TY}(\mathbb{Z}_2, \epsilon = -1)$.

We now proceed with our lattice discussion. We consider first the Heisenberg chain with $2L$ sites

$$H_{\mathrm{Heisenberg}}(t) = (1-t)H_{\mathrm{trivial}} + tH_{\mathrm{XXX}},$$

$$H_{\mathrm{trivial}} = \sum_{j=1}^{2L} X_j,$$

$$H_{\mathrm{XXX}} = \sum_{j=1}^{2L}(X_j X_{j+1} + Y_j Y_{j+1} + Z_j Z_{j+1}). \tag{H.1}$$

For $t = 1$, this is the $SO(3)$ invariant Heisenberg chain without an external magnetic field. In the continuum limit, it leads to the $SU(2)_1$ conformal field theory and the lattice translation symmetry $T_{\mathrm{Heisenberg}}$ leads to an emanant internal $\mathbb{Z}_2^{\mathcal{C}}$ symmetry generated by $\mathcal{C}$ [46,52]. This $\mathbb{Z}_2^{\mathcal{C}}$ symmetry has a self-anomaly (see, for example, [117,164]). That anomaly has a lattice precursor near the continuum limit [52], but it is not an exact property of the lattice system. To see that, consider the background magnetic field deformation in (H.1). It breaks the $SO(3)$ symmetry, but preserves the $\mathbb{Z}_2$ symmetry that we are interested in. Now, it is clear that the long-distance behavior can be different and in particular, for $t = 0$, it is trivial and does not have the emanant symmetry with its self-anomaly.

Next, we tensor this model with our Ising chain with $L$ sites and Hamiltonian $H$ in (10).

---

[55]See [163] for more discussions of mixed anomalies between fusion categories and time-reversal/parity.

This means that the total Hamiltonian is[56]

$$H_{\text{Total}} = H \otimes 1 + 1 \otimes H_{\text{Heisenberg}}(t), \tag{H.2}$$

and consider the symmetry operator

$$\mathsf{D}_{\text{Total}} = \mathsf{D} \otimes T_{\text{Heisenberg}}^{-1}. \tag{H.3}$$

This operator commutes with the total Hamiltonian $H_{\text{Total}}$ and it satisfies

$$
\begin{aligned}
\mathsf{D}_{\text{Total}}^2 &= (1 + \eta_{\text{Total}}) T_{\text{Total}}^{-1}, \\
T_{\text{Total}} &= T \otimes T_{\text{Heisenberg}}^2, \\
\eta_{\text{Total}} &= \eta \otimes 1,
\end{aligned}
\tag{H.4}
$$

which is the same algebra as the symmetry algebra of the Ising model without the Heisenberg factor. (Recall that the number of sites in the added Heisenberg chain is $2L$ and number of sites of the Ising chain is $L$, such that $T_{\text{Total}}^L = 1$.)

For $t = 0$, the low energy phase of the combined system is described the Ising CFT and $\mathsf{D}_{\text{Total}}$ flows to the non-invertible symmetry $\mathcal{N}$ of $\text{TY}(\mathbb{Z}_2, +)$ with $\epsilon = +1$. For $t = 1$, the low energy phase is described by a tensor product of two CFTs, Ising and $SU(2)_1$, and the operator $\mathsf{D}_{\text{Total}}$ becomes the non-invertible symmetry $\mathcal{N}' = \mathcal{N} \otimes \mathcal{C}$. As discussed at the beginning of this appendix, the FS indicator for $\mathcal{N}'$ is $\epsilon = -1$, and $1, \eta, \mathcal{N}'$ generate the $\text{TY}(\mathbb{Z}_2, -)$ fusion category.

In conclusion, this lattice model is invariant under our lattice symmetry operators. As we vary its parameters, it has (at least) two phases both with the continuum symmetry $\text{TY}(\mathbb{Z}_2, \epsilon)$. In one of them, the FS indicator $\epsilon$ is $+1$ and in the other, it is $-1$.

# I Non-invertible symmetry of superconformal minimal models and their deformations

## I.1 The supersymmetric Ginzburg-Landau model

### I.1.1 The model and its symmetries

Following [165], we consider a Ginzburg-Landau description of the unitary supersymmetric minimal models. This is a continuum 1+1d theory with $(1, 1)$ supersymmetry, based on a real superfield

$$\Phi(\theta_L, \theta_R) = \varphi + \theta_L \psi_R + \theta_R \psi_L + i \theta_L \theta_R F, \tag{I.1}$$

and the Lagrangian

$$\int d^2\theta \left( \frac{1}{2} D_L \Phi D_R \Phi + W(\Phi) \right). \tag{I.2}$$

This theory has a standard $(-1)^F$ symmetry under which $\theta_L$ and $\theta_R$ change sign. When the superpotential $W(\Phi)$ is even, there is also a $\mathbb{Z}_2$ global symmetry acting as $\Phi(\theta_L, \theta_R) \to -\Phi(\theta_L, \theta_R)$.

We are interested in the case of odd $W(\Phi)$. Then, the theory has a $(-1)^{F_L}$ symmetry acting as

$$(-1)^{F_L} \quad : \quad \Phi(\theta_L, \theta_R) \to -\Phi(-\theta_L, \theta_R). \tag{I.3}$$

Clearly, it is an R-symmetry. It is known to be related to Kramers-Wannier duality [165].

---

[56] Here $\otimes$ stands for the standard tensor product of two linear maps on two vector spaces. This is not to be confused with the fusion operation for the defects.

As in [153], when the fermions are periodic, i.e., in the RR theory, we have an anomaly

$$(-1)^{F_L}(-1)^F = -(-1)^F(-1)^{F_L}.\tag{I.4}$$

This anomaly is at the root of the non-invertible symmetry of the bosonic theory, which we will discuss soon.

Most of the literature about this theory focuses on its gapless phases. In particular, for $W(\Phi) \propto \Phi^m$, the model flows to the $(1,1)$ superconformal minimal model with central charge $c = \frac{3}{2} - \frac{12}{m(m+2)}$. We will consider its gapped phase. The simplest such case with odd $W(\Phi)$ has

$$W(\Phi) = h\left(\Phi - \frac{1}{3}\Phi^3\right),\tag{I.5}$$

and the corresponding potential is

$$V = h^2(\varphi - 1)^2(\varphi + 1)^2.\tag{I.6}$$

### I.1.2 Ignoring the fermions

For large $h$, we can use a semiclassical picture. Let us first ignore the fermions. The bosonic potential has two minima $\varphi = \pm 1$ and correspondingly, the system has two approximate ground states $|\varphi = \pm 1\rangle$. Instantons mix them and lead to two energy eigenstates

$$|\pm\rangle = \frac{1}{\sqrt{2}}\Big(|\varphi = +1\rangle \pm |\varphi = -1\rangle\Big),\tag{I.7}$$

which are $(-1)^{F_L}$ eigenstates

$$(-1)^{F_L}|\pm\rangle = \pm|\pm\rangle.\tag{I.8}$$

The true ground state is $|+\rangle$ and the state $|-\rangle$ has energy of order $e^{-aL}$ with some positive constant $a$, where $L$ is the spatial circumference of space.

### I.1.3 The fermionic field theory

Next, we add the fermions. Now, $\varphi$ and $\psi_L\psi_R$ can mix because they are both $(-1)^F$ even and $(-1)^{F_L}$ odd. Indeed, the component Lagrangian includes a term proportional to $\varphi\psi_L\psi_R$. This coupling means that the fermions are massive at the two states $|\varphi = \pm 1\rangle$ and one might want to ignore them.

When the fermions are anti-periodic, i.e., in the NSNS theory, this reasoning is correct and we can simply ignore the fermions. Therefore, the lowest energy states are $|\pm\rangle_{NSNS}$ of (I.7), where we added the subscript $NSNS$ to show that these are states in the NSNS theory. They are eigenstates of $(-1)^F$ and $(-1)^{F_L}$

$$\begin{aligned}
(-1)^F|\pm\rangle_{NSNS} &= +|\pm\rangle_{NSNS},\\
(-1)^{F_L}|\pm\rangle_{NSNS} &= \pm|\pm\rangle_{NSNS}.
\end{aligned}\tag{I.9}$$

As in the problem without the fermions, the true ground state is $|+\rangle_{NSNS}$ and the state $|-\rangle_{NSNS}$ has slightly higher energy.

The situation is more interesting when the fermions are periodic, i.e., in the RR theory. Now, the instanton that interpolates between the two states $|\varphi = \pm 1\rangle$ has a fermion zero mode. Therefore, it does not split the degeneracy between them. Instead, we end up with two degenerate states. We can take them to be $|\varphi = \pm 1\rangle_{RR}$, where the subscript $RR$ denotes that they are the ground states in the RR theory. These states satisfy

$$\begin{aligned}
(-1)^F|\varphi = \pm 1\rangle_{RR} &= \pm|\varphi = \pm 1\rangle_{RR},\\
(-1)^{F_L}|\varphi = \pm 1\rangle_{RR} &= |\varphi = \mp 1\rangle_{RR}.
\end{aligned}\tag{I.10}$$

This is consistent with the algebra (I.4), which does not have one-dimensional representations. (Of course, we could have taken the basis $|\pm\rangle_{RR}$.)

Clearly, the two states $|\varphi = \pm 1\rangle_{RR}$ are related by the action of supersymmetry. Therefore, this model has vanishing Witten index (the contribution of these two states to the index cancel each other because they have opposite $(-1)^F$) and supersymmetry is spontaneously broken.

### I.1.4 Performing the GSO projection

So far, we discussed the Ginzburg-Landau model as a fermionic quantum field theory, which depends on a choice of the spin structure. Let us consider the corresponding bosonic quantum field theory (which does not require a choice of the spin structure) obtained by performing the GSO projection.

In particular, as emphasized in [128], we should distinguish between two different gapless theories corresponding to $W(\Phi) \propto \Phi^3$. Its fermionic version is the first $(1, 1)$ superconformal minimal model and its bosonized version is the tricritical Ising CFT.

Following the standard bosonization procedure, reviewed in [119, 128, 154–160], we combine the NSNS and the RR Hilbert spaces and assign a quantum $\mathbb{Z}_2^\eta$ symmetry $\eta = +1$ to the NSNS sector and $\eta = -1$ to the RR sector. Then, we project on $(-1)^F = +1$.

After bosonization, the deformation $\Phi$ of the pure $\Phi^3$ superpotential is mapped to the subleading thermal deformation $\varepsilon'$ of the tricritical Ising CFT discussed in Section 4.2 and denoted by $\lambda$ at the tricritical point in Figure 1.

Let us determine the low-lying states of this bosonic theory in its gapped phase. From the NSNS sector, we get the states $|\pm\rangle_{NSNS}$ with $\eta = +1$ and $(-1)^{F_L} = \pm 1$. From the RR sector, we have a single state $|\varphi = +1\rangle_{RR}$ with $\eta = -1$ and no well-defined $(-1)^{F_L}$. (Alternatively, we could project in the RR sector on $(-1)^F = -1$ and end up with $|\varphi = -1\rangle_{RR}$ also with $\eta = -1$.)

This picture fits with our general story. As in [154, 157, 160], we identify $\mathcal{N} = \sqrt{2}(-1)^{F_L}$ in the $\eta = +1$ sector and $\mathcal{N} = 0$ in the $\eta = -1$ sector. They satisfy (G.1)

$$\eta^2 = 1, \qquad \eta\mathcal{N} = \mathcal{N}\eta = \mathcal{N}, \qquad \mathcal{N}^2 = 1 + \eta, \tag{I.11}$$

and we end up with three low-lying states:

$$
\begin{aligned}
&|+\rangle_{NSNS}, &&\text{with} && \eta = +1, && \mathcal{N} = +\sqrt{2}, \\
&|-\rangle_{NSNS}, &&\text{with} && \eta = +1, && \mathcal{N} = -\sqrt{2}, \\
&|\varphi = +1\rangle_{RR}, &&\text{with} && \eta = -1, && \mathcal{N} = 0.
\end{aligned}
\tag{I.12}
$$

These three states are nearly degenerate – their energy differences are of order $e^{-aL}$. This energy difference between the two states from the NSNS sector is clear from the discussion above. In order to see why the state from the RR sector is also close to these, we need to study the large $L$ theory and examine the action of the local operators there. We will do it now.

## I.2 Spontaneous breaking of (non-)invertible symmetries in infinite volume

Let us discuss the infinite volume system and follow the steps above. Ignoring the fermions, we have the two low-lying states $|\pm\rangle$ of (I.7). In the infinite volume limit, the Hilbert space splits into two distinct superselection sectors where $\varphi$ is diagonal. The ground states in these sectors are $|\varphi = \pm 1\rangle$ and $\varphi = \text{diag}(+1, -1)$. We see that the $(-1)^{F_L}$ symmetry is spontaneously broken.

Adding the fermions, the same conclusion applies to the fermionic Ginzburg-Landau field theory. The $(-1)^{F_L}$ symmetry is spontaneously broken and the superselection sectors are $|\varphi = \pm 1\rangle_{NSNS}$ in the NSNS theory, and $|\varphi = \pm 1\rangle_{RR}$ in the RR theory. Note that unlike the

NSNS theory, where the two states have $(-1)^F = 1$, in the RR theory these two states have $(-1)^F = \pm 1$.

The RR theory, but not the NSNS theory, is supersymmetric. In finite volume, this supersymmetry was spontaneously broken. In the infinite volume limit, it is restored and the ground states have vanishing energy. One way to see that is to note that the spectrum is gapped and does not include a massless Goldstino.

Finally, we turn to the GSO-projected, bosonic field theory in infinite volume. Here, the three low-lying states of the finite-volume theory (I.12) lead to three separate superselection sectors with ground states

$$\frac{1}{\sqrt{2}}\big(|\varphi = +1\rangle_{NSNS} \pm |\varphi = +1\rangle_{RR}\big), \qquad |\varphi = -1\rangle_{NSNS}. \tag{I.13}$$

One way to pick this basis is to diagonalize the local operators of the theory. The operator $\varphi$ and the spin field $\sigma$, which maps NSNS$\longleftrightarrow$RR, act as

$$\begin{aligned} \varphi &= \mathrm{diag}(+1, +1, -1), \\ \sigma &= \mathrm{diag}(+1, -1, 0), \end{aligned} \tag{I.14}$$

where the last 0 in $\sigma$ follows from the fact that there is no corresponding RR state.

This shows that in the infinite volume bosonic theory, the $\mathbb{Z}_2^{\eta}$ symmetry is spontaneously broken in the first two superselection sectors and it is unbroken in the third. $\mathcal{N}$ does not have a simple action in any of them and we interpret it to mean that it is spontaneously broken. More precisely, the symmetry operators $\eta$ and $\mathcal{N}$ do not exist in the infinite volume theory.

## I.3 Relation to the 1+1d Ising TQFT

Let us return to the finite-volume theory, focus on its three low-lying states (I.12), and try to write an effective theory describing them. As we reviewed in Section 4.2, it is standard to ignore their energy differences and then the low-energy theory is a 1+1d topological theory with three states.

Since we neglect the energy differences, we can use either the basis (I.12), where the line operators $(1, \mathcal{N}, \eta)$ are diagonal: $\mathcal{N} = \mathrm{diag}(+\sqrt{2}, -\sqrt{2}, 0)$, $\eta = \mathrm{diag}(+1, +1, -1)$, or the basis (I.13), where the point operators $(1, \varphi, \sigma)$ are diagonal: $\varphi = \mathrm{diag}(+1, +1, -1)$, $\sigma = \mathrm{diag}(+1, -1, 0)$ and the line operators are

$$\eta = \begin{pmatrix} 0 & 1 & 0 \\ 1 & 0 & 0 \\ 0 & 0 & 1 \end{pmatrix}, \quad \text{and} \quad \mathcal{N} = \begin{pmatrix} 0 & 0 & 1 \\ 0 & 0 & 1 \\ 1 & 1 & 0 \end{pmatrix}. \tag{I.15}$$

We emphasize that the operators (I.15) make sense in the finite-volume theory, but they are meaningless in the infinite-volume theory, where the symmetry operators $\eta$ and $\mathcal{N}$ do not exist.

This TQFT is well studied. In this context, it is convenient to express the local operators 1, $\varphi$, $\sigma$ as linear combinations of idempotent operators $\varepsilon_1, \varepsilon_2, \varepsilon_3$ that satisfy the simple projection operator algebra [166, 167]

$$\varepsilon_i \varepsilon_j = \delta_{ij} \varepsilon_i. \tag{I.16}$$

Specifically, consider the operators [60, 168]

$$\begin{aligned} 1 &= \varepsilon_1 + \varepsilon_2 + \varepsilon_3, \\ v_\varphi &= \varphi = \varepsilon_1 + \varepsilon_2 - \varepsilon_3, \\ v_\sigma &= \sqrt{2}\sigma = \sqrt{2}\varepsilon_1 - \sqrt{2}\varepsilon_2. \end{aligned} \tag{I.17}$$

The projection operators $\varepsilon_i$ project onto the normalized states $|i\rangle$:

$$
\begin{aligned}
|1\rangle &= \varepsilon_1 |\text{HH}\rangle\,, \\
|2\rangle &= \varepsilon_2 |\text{HH}\rangle\,, \\
|3\rangle &= \frac{1}{\sqrt{2}} \varepsilon_3 |\text{HH}\rangle\,,
\end{aligned}
\tag{I.18}
$$

where $|\text{HH}\rangle$ is the "Hartle-Hawking" state corresponding to the identity operator. In our conventions, we have

$$
\begin{aligned}
|\text{HH}\rangle &= |1\rangle + |2\rangle + \sqrt{2}|3\rangle\,, \\
\left|v_\varphi\right\rangle &= v_\varphi |\text{HH}\rangle = |1\rangle + |2\rangle - \sqrt{2}|3\rangle\,, \\
|v_\sigma\rangle &= v_\sigma |\text{HH}\rangle = \sqrt{2}|1\rangle - \sqrt{2}|2\rangle\,,
\end{aligned}
\tag{I.19}
$$

where $\langle \text{HH}|\text{HH}\rangle = \langle v_\varphi | v_\varphi \rangle = \langle v_\sigma | v_\sigma \rangle = 4$. (We use $v_\varphi$ and $v_\sigma$ both for the operators and the label of the states.) They correspond to the states $|I\rangle$ in Section 4.2.

The basis of states $|i\rangle$ is natural from the point of view of the state/operator correspondence of the operators $\varepsilon_i$. Furthermore, the local operators $1$, $\varphi$, $\sigma$ are diagonal in this basis (I.14). Finally, as above, in this basis of states, the symmetry line operators are given by (I.15). These matrices satisfy the fusion rule (I.11), and thus form a Nonnegative Integer-valued Matrix representation (NIM-rep) of the algebra.

So far, we discussed the TQFT. As mentioned in [60], the states (I.18), where the local operators $1$, $\varphi$, $\sigma$ are diagonal, correspond to the superselection sectors of the infinite-volume theory. This leads to the conclusion about spontaneous symmetry breaking we discussed above.

## I.4 Generalizations

Clearly, this discussion is easily generalized to a more complicated odd superpotentials $W(\Phi)$ and therefore it applies to all the gapped states obtained by appropriate supersymmetric deformations of the odd members of the supersymmetric discrete series.

Similarly, we can deform this model by adding a supersymmetry-breaking, but $(-1)^F$ and $(-1)^{F_L}$-preserving deformations, like $\varphi^2$, without a qualitative change in our conclusions.

## J Constraints on renormalization group flows from non-invertible symmetries

Here we review a statement proven in Section 7.1 of [12] for renormalization group flows from a 1+1d CFT with a unique ground state in finite volume.[57]

*Consider a 1+1d CFT deformed by a relevant operator preserving a non-invertible topological line $\mathcal{L}$ with quantum dimension $\langle \mathcal{L} \rangle$.[58] Suppose $\langle \mathcal{L} \rangle \notin \mathbb{Z}_{\geq 0}$, then the low-energy phase cannot be gapped with a non-degenerate ground state.*

This constraint on the renormalization group flow was interpreted as a generalized 't Hooft anomaly for the non-invertible symmetry.[59] Recall that the quantum dimension $\langle \mathcal{L} \rangle$ of a topological line $\mathcal{L}$ in a CFT with a unique ground state $|1\rangle$ is defined as $\mathcal{L}|1\rangle = \langle \mathcal{L} \rangle |1\rangle$. It is known that $\langle \mathcal{L} \rangle = 1$, if and only if, the topological line $\mathcal{L}$ is associated with an invertible symmetry

---

[57]The standard definition of a CFT assumes that it has a unique ground state. Here we would like to exclude more general situations where the theory is gapless and has several ground states, e.g., a tensor product of a CFT and a TQFT.

[58]Note that the discussion here is about the quantum dimension of the continuum theory, rather than the lattice quantum dimension of the lattice model we discussed in section 3.4.

[59]Anomalies in non-invertible global symmetries were discussed in [12, 32, 60, 61, 116, 169–171].

(see [12] for a physics argument). We stress that some topological lines with integral quantum dimensions also forbid a trivially gapped phase. For example, an invertible line associated with a finite, invertible global symmetry with an 't Hooft anomaly has quantum dimension 1, but it is incompatible with a trivially gapped phase.

We prove the assertion above by contradiction. We assume that the low-energy phase is trivially gapped with a non-degenerate ground state. In the deep IR where all the massive degrees of freedom have been integrated out, the continuum description of this trivially gapped phase is a trivial 1+1d TQFT with a unique ground state. Its (untwisted) Hilbert space $\mathcal{H}$ on a circle is one-dimensional, i.e., $\dim \mathcal{H} = 1$. (See [167,168], for reviews of 1+1d TQFTs.) In this trivial TQFT, consider the torus partition function with a non-invertible operator $\mathcal{L}$ inserted at a fixed time:

$$Z^{\mathcal{L}} = \mathrm{Tr}_{\mathcal{H}}[\mathcal{L}] = \langle \mathcal{L} \rangle \,. \tag{J.1}$$

Next, consider the partition function over the $\mathcal{L}$-twisted Hilbert space, where $\mathcal{L}$ is now a defect inserted at a fixed point in space:

$$Z_{\mathcal{L}} = \mathrm{Tr}_{\mathcal{H}_{\mathcal{L}}}[1] = \dim \mathcal{H}_{\mathcal{L}} \,. \tag{J.2}$$

The two partition functions $Z^{\mathcal{L}}$ and $Z_{\mathcal{L}}$ are related by a modular transformation exchanging time with space:

$$Z^{\mathcal{L}} = Z_{\mathcal{L}} \,. \tag{J.3}$$

Hence,[60]

$$\langle \mathcal{L} \rangle = \dim \mathcal{H}_{\mathcal{L}} \,. \tag{J.4}$$

While a priori we do not know much about the twisted Hilbert space $\mathcal{H}_{\mathcal{L}}$, its dimension is necessarily a non-negative integer. We therefore have reached a contradiction if $\langle \mathcal{L} \rangle \notin \mathbb{Z}_{\geq 0}$. This completes the proof.

A renormalization group flow preserving a non-invertible symmetry with $\langle \mathcal{L} \rangle \notin \mathbb{Z}_{\geq 0}$ can flow in the low energy to either a gapless phase, or a gapped phase with multiple ground states described by a 1+1d TQFT. In the latter case, the non-invertible symmetry acts nontrivially on the ground states, which means that the symmetry is spontaneously broken.

See [12, 31, 32, 60, 126, 163, 168, 172–174], for more discussions of non-invertible symmetries and their spontaneous breaking in gapped phases.

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
