# Peer review of "Non-invertible symmetries and LSM-type constraints on a tensor product Hilbert space"

_SciPost Physics, doi:SciPost Phys. 16, 154 (2024)_

## Round 1 · Referee Report · Anonymous (Referee 1) · 2024-4-28

Report

This article investigated non-invertible symmetry associated with the Kramers-Wannier transformation on a tensor product Hilbert space of qubits on a periodic 1d chain. The authors pointed out that the lattice symmetry does not form a fusion category and that the Frobenius–Schur indicator of the continuum fusion category is not meaningful on the lattice. One of the main results is contained in Section 4.2 where an LSM-type constraint based on the non-invertible lattice symmetry was proposed. This article contains several important results, with remarkable clarity of exposition. The referee therefore recommends that the manuscript should be published in SciPost. Nevertheless, prior to publication, the authors are asked to address the following comments and questions.

Requested changes

  1. The authors demonstrated that the LSM-type constraint discussed in Section 4.2 applied to the critical Ising Hamiltonian and deformations that preserve the non-invertible symmetry. The examples provided in the articles are Ising and tricritical Ising CFTs. The referee would like to ask the author to clarify whether there are any other CFTs in the continuum that can be described by such Hamiltonians. The referee also asks the authors to provide more examples if possible. (The referee is aware that the minimal models in the continuum were discussed in Appendix I. However, even the discussion there seemed to put emphasis on the tricritical Ising CFT.)
  2. A similar comment applies to Figure 1. Can this or a similar phase diagram describe the flows between other CFTs?  How about the other flows discussed in Section 7.2 of [arXiv:1802.04445]?

Recommendation

Publish (surpasses expectations and criteria for this Journal; among top 10%)

---

## Round 1 · Referee Report · Anonymous (Referee 2) · 2024-5-6

Strengths

  1. Thorough and clear discussions on the non-invertible Kramers-Wannier duality defect and the corresponding symmetry operator on a tensor product Hilbert space of qubits.
  2. Careful comparison between the Kramers-Wannier symmetry on the lattice and that in the continuum, with an emphasis on the role of the lattice translation.
  3. LSM-type constraints on 1+1d lattice models with Kramers-Wannier symmetry.

Report

This paper describes various aspects of the Kramers-Wannier symmetry of 1+1d lattice models of qubits. As a prototypical example, the authors mainly focus on the Kramers-Wannier symmetry of the critical Ising model in 1+1d. However, many results derived in the paper also apply to lattice models with more general Hamiltonians as long as the Kramers-Wannier symmetry is preserved.

One of the important observations in the paper is that the fusion rules of the Kramers-Wannier duality defect on the lattice involve lattice translation, which plays a key role in understanding the difference between the Kramers-Wannier symmetry on the lattice and that in the continuum. In particular, the mixing with the lattice translation makes it impossible to define the Frobenius-Schur indicator of the duality defect on the lattice, which is in contrast to the situation in the continuum. Relatedly, the same Kramers-Wannier symmetry on the lattice can lead to different fusion category symmetries in the continuum. Indeed, the authors provided a one-parameter family of lattice models with the Kramers-Wannier symmetry that in the continuum interpolates between the two $\mathbb{Z}_2$ Tambara-Yamagami categories with different Frobenius-Schur indicators.

Another interesting result is a no-go argument for a realization of the Kramers-Wannier symmetry without involving lattice translation on a tensor product Hilbert space. The argument exploits a new definition of the quantum dimension on the lattice, which differs from the quantum dimension in the continuum. The authors also showed LSM-type constraints, which state that any finite range Hamiltonian with the Kramers-Wannier symmetry on a tensor product Hilbert space has to be either (1) gapless or (2) gapped with the ground state degeneracy being a multiple of 3. This statement is the lattice analogue of the fact that any (relativistic) continuum field theory with a $\mathbb{Z}_2$ Tambara-Yamagami symmetry flows to either (1) a CFT or (2) a TQFT with the number of vacua being a multiple of 3.

Along the way, the authors formulated the movement and fusion of Kramers-Wannier duality defects on the lattice by using unitary operators. This formulation allows one to derive the explicit MPO presentations of the Kramers-Wannier symmetry operators in the presence of various defects. The algebras of these symmetry operators are compared carefully with their counterparts in the continuum.

The above results clarify many important subtleties about the relation between topological defects on the lattice and those in the continuum. As such, this paper would be of great importance in the study of both lattice models and continuum field theories. Thus, I recommend this paper for publication in SciPost Physics.

Requested changes

Please see below for my questions and comments.

  1. p.20, below eq. (2.31), a topological defect is called simple if there is no local operator that commutes with the defect Hamiltonian. Is this definition specific to the model considered here? If this definition applies to more general models, is the identity defect of a commuting projector model non-simple because there are local operators that commute with the Hamiltonian?

  2. p.33, Section 2.5, could you clarify in what sense the projective algebras (2.84)-(2.86) are regarded as mixed anomalies? For example, are these projective algebras related to obstructions to gauging or a non-degenerate gapped phase?

  3. p.93, Section G.4 (related to the above question), an equation similar to eq. (G.12) would also hold for a $\mathbb{Z}_2 \times \mathbb{Z}_2$ Tambara-Yamagami symmetry with a specific choice of a bicharacter. Does this mean that there is a mixed anomaly between the $\mathbb{Z}_2 \times \mathbb{Z}_2$ Tambara-Yamagami symmetry and the parity/time-reversal? Does this further imply that a gapped phase with this symmetry must have degenerate ground states even if the Tambara-Yamagami symmetry is non-anomalous?

I would also like to point out the following small typos.

  1. p.22, eq. (2.39), "$(Z_{j-1} Z_{j})$" $\rightarrow$ "$(Z_{j-1} Z_{j} + X_j)$"

  2. p.28, in the title of Section 2.3.4, "operator $\mathsf{D}_{\mathcal{D}}$ operator" $\rightarrow$ "operator $\mathsf{D}_{\mathcal{D}}$"

  3. p.41, the second paragraph of Section 3.3, "we can defines a lattice counterpart" $\rightarrow$ "we can define a lattice counterpart"

  4. p.77, eq. (E.9), "$X_j \mapsto X_{(J, J+1)}$" $\rightarrow$ "$X_J \mapsto X_{(J, J+1)}$"

  5. p.78, the subscripts in eq. (E.17), "1" and "2" $\rightarrow$ "$j$" and "$j+1$"

Recommendation

Publish (surpasses expectations and criteria for this Journal; among top 10%)

---

## Round 2 · Referee Report · Anonymous (Referee 3) · 2024-5-21

Report

The authors have addressed the referees' comments as well as corrected the minor typos. I recommend this article for publication.

Recommendation

Publish (surpasses expectations and criteria for this Journal; among top 10%)

---

## Round 2 · Referee Report · Anonymous (Referee 4) · 2024-5-22

Report

The authors have addressed all the questions and comments raised. Thus, I recommend this paper for publication.

Recommendation

Publish (surpasses expectations and criteria for this Journal; among top 10%)

---

## Round 2 · Author Response

We thank the two referees for their detailed and helpful comments on our manuscript. We have addressed them in the new version of the draft. We have also fixed the typos pointed out by the referees and a number of other minor mistakes. See below a summary of the more significant changes.

Report 2:

  1. The definition of simple is not specific to the model that we considered here. Indeed, for a commuting projector Hamiltonian, the identity defect is not simple since the total symmetry of the model is very large and contains local operators that commute with the Hamiltonian. Relatedly, in 1+1d continuum QFTs, symmetries implemented by local operators correspond to multifusion categories. In these cases, the identity defect is not simple.

  2. We revised the discussion on the "anomaly" involving parity/time-reversal. We only mention that it is reminiscent of anomalies, and we do not claim that these projective phases correspond to obstruction to gauging or additional LSM-type constraints.

  3. Same as above.

Report 1:

We have only explored the phase diagram of lattice systems where the continuum limit gives the Ising and the tricritical Ising CFTs. It would indeed be interesting to consider more general lattice models and continuum field theories with non-invertible symmetries. We note that there have been some recent papers studying non-invertible symmetries in the 3-state Potts lattice model and CFT.

---

## Round 2 · List of Changes

1. We have revised the discussion on the "anomaly" involving parity/time-reversal throughout the manuscript.

2. We expanded the discussion in section 3.4.

3. Various minor typos fixed and references added.

---

## Editorial Decision

published